# A study of competitions in different fields through graphs under bipolar picture fuzzy environment

**Waheed Ahmad Khan**[1]*, **Hajra Begum**[1], **Trung Tuan Nguyen**[2],
**Minh Hoan Pham**[2], **Hai Van Pham**[3]*

**1** Division of Science and Technology, Department of Mathematics, University of Education Lahore, Attock Campus, Attock, Pakistan, **2** College of Technology, National Economics University, Hanoi, Vietnam, **3** School of Information and Communication Technology, Hanoi University of Science and Technology, Hanoi, Vietnam

☺ These authors contributed equally to this work.
* sirwak2003@yahoo.com; haipv@soict.hust.edu.vn

**Data availability statement:** The data are available from https://doi.org/10.6084/m9.figshare.29879801.

## Abstract

Recent developments in the theory of fuzzy graphs have led to many extensions for modeling real-world problems involving uncertainty. Among these, competition graphs are crucial for representing competitive and ecological systems. In this study, the notion of bipolar picture fuzzy competition graphs, along with several generalizations including bipolar picture fuzzy k-competition graphs, p-competition graphs, and m-competition graphs are introduced. Several characteristics of these newly established graphs are investigated. We also explore their structural properties and apply the models to real-world competitive scenarios using computational frameworks. A comparative study is conducted to demonstrate the improved efficiency of the proposed models over existing approaches.

## 1 Introduction

### 1.1 Background

Graphs are helpful in studying relationships between pairs of objects. Numerous applications of graphs in many fields such as medical sciences [1], computer science [4,5], artificial intelligence [6], chemistry [7] etc. have been explored. If a graph does not contain a loop, then it is called simple and is denoted as $G = (V, E)$, where $V$ represents the vertices (nodes) set and $E$ represents the set of edges (paths). A path between nodes $u, v \in V$ is represented by $(u, v)$. Similarly, directed graphs (or digraphs), denoted by $\vec{G} = (V, \vec{E})$ are such that each edge $(u, v) \in \vec{E}$ is a directed edge from $u$ to $v$. For further details about graph theory, one may consult the works of Diestel [8] and Bondy and Murthy [9]. Food chains in ecology have been elegantly modeled through digraphs, where nodes denote species and paths depict relationships like predation. We can demonstrate any food chain through a directed graph $\vec{G} = (V, \vec{E})$. Considering nodes $u, v, w \in V$ as species and an edge $(u, w) \in \vec{E}$ indicates that species $u$ hunts species $w$. If both species $u$ and $v$ prey on species $w$, they are said to compete for $w$. Due to this, Cohen [10] introduced the idea of competition graphs (CGs) associated with

**Funding:** The work is funded by National Economics University, Hanoi, Vietnam.

**Competing interests:** The authors have declared that no competing interests exist.

ecosystems. The CGs $C(\vec{G}) = (V, R)$ of $\vec{G}$ is an undirected graph, where nodes $u$ and $v$ in $V$ are connected by a path $(u, v) \in R$. The generalization of competition graphs (CGs) was discussed in [11]. After Cohen's introduction of CGs, several types these graphs were explored in [12–15]. The p-competition graphs of digraphs was introduced in [14,15]. Similarly, the m-step competition graph of a digraph was addressed in [16].

## 1.2 Need of fuzzy generalizations

In all of the above representations the graphs considered were crisp graphs. However, these competition graphs do not adequately address all types of competitions present in real-world problems. Traditional CGs focus on common prey and related species and hence these CGs do not consider the degree of dependence on a common prey relative to other species. Therefore, there was a need to represent these concepts in a more flexible framework. To address such circumstances, the notion of fuzzy competition graphs (FCGs) was introduced in the literature. Sometimes it depends on the competition power of the competitors, and the number of competitors may vary with respect to time. Taking these factors into account, fuzzy k-competition graphs (Fk-CGs) and fuzzy p-competition graphs (Fp-CGs) were described in [17]. These graphs help to describe the different kinds of competitions in the real-time scenarios. However, if there exists a species $w \in V$ such that $(u, w) \in \vec{E}$ and $(v, w) \in \vec{E}$ with $u \neq v$, then such new competitive situations may render between species and their prey are often inadequate. For instance, in food networks, species exhibit different properties like veg, non-veg, powerless, strong, etc while food can be delegated stomach-related, indigestive, delectable and so on. To address such situations i.e., for an indirect competition between species, the concept of m-step fuzzy CGs (m-SCGs) was introduced in [18]. Al-shehri and Akram [19] developed the notion of bipolar fuzzy competition graphs (BFCGs) which provided more precise the degrees of competitions. Many new terms related to BFCGs and their significant uses were explored in [20]. Similarly, some algorithms for computing the strength of competition in BFCGs were introduced in [21]. The notion of intuitionistic fuzzy competition graphs (IFCGs) was described in [22]. Bipolar intuitionistic fuzzy competition graphs (BIFCGs), an extension of IFCGs was explored in [23]. Likewise, another generalization of IFCGs termed picture fuzzy competition graphs (PFCGs) was presented in [24]. Some applications of PFCGs in the field of medicine were explored in [25]. Subsequently, we present further extension of PFCGs named BPFCGs along with its several types. Overall, we extend the theory of FCGs by introducing several significant concepts in the domain of BPFCGs.

The abbreviations used throughout this article are listed in Table 1.

## 1.3 Literature review

To address uncertainties and vagueness in everyday scenarios, Zadeh [26] introduced fuzzy sets (FSs) in 1965. A fuzzy set, represented by $A = \{x, \mu_A(x) | x \in X\}$ assigns a membership value to each element from $[0, 1]$. This degree of membership (DMS) allowed researchers to discuss the characteristics of elements in a set of data containing uncertainties, more precisely. Due to the flexible nature of FSs, many generalizations of FSs have been introduced. The first generalization of FSs termed interval-valued fuzzy sets (IVFSs) was also introduced by Zadeh [27]. Atanassov [28] introduced the notion of Intuitionistic fuzzy sets (IFSs) to handle uncertainties in everyday scenarios. In IFSs, both the degrees of membership and non-membership were allocated to each entity. However, the term neutrality was still missing in the theory of IFSs which has its own importance in daily life problems such as democratic elections and so on. To manipulate such circumstances, Cuong and Kreinovich [29] presented the idea of picture fuzzy sets (PFSs). PFSs includes an additional component the neutrality, which enhances

**Table 1**. Abbreviations used in this study.

| Terminologies | Notations | Terminologies | Notations |
|---|---|---|---|
| Fuzzy Sets | FSs | Bipolar Intuitionistic Fuzzy Sets | BIFSs |
| Fuzzy Graphs | FGs | Bipolar Intuitionistic Fuzzy Graphs | BIFGs |
| Intuitionistic Fuzzy Sets | IFSs | Bipolar Picture Fuzzy Sets | BPFSs |
| Intuitionistic Fuzzy Graphs | IFGs | Bipolar Picture Fuzzy Graphs | BPFGs |
| Picture Fuzzy Sets | PFSs | Intuitionistic Fuzzy Directed Graphs | IFDGs |
| Picture Fuzzy Graphs | PFGs | Fuzzy Competition Graphs | FCGs |
| Biploar Fuzzy Sets | BFSs | Bipolar Fuzzy Competition Graphs | BFCGs |
| Bipolar Fuzzy Graphs | BFGs | Bipolar Intuitionistic Fuzzy Directed Graphs | BIFdGs |
| Membership | MS | Bipolar Fuzzy Directed Graphs | BFDGs |
| Degree of Membership | DMS | Competition Graphs | CGs |
| Degree of Non-Membership | DNMS | Degree of Restrain Membership | DRMS |
| Fuzzy in-neighbourhood | FIN | Intuitionistic Fuzzy Competition Graphs | IFCGs |
| Fuzzy Directed Graph | FDG | Picture Fuzzy Competition Graphs | PFCGs |
| Fuzzy Out-neighbourhood | FON | Bipolar Intuitionistic Fuzzy Competition Graphs | BIFCGs |
| Bipolar Picture Fuzzy Path | BPFP | Bipolar Picture Fuzzy Competition Graphs | BPFCGs |

its capability to deal uncertain environments, adequately. Numerous applications of PFSs in different fields were explored [30–32]. Moreover, in many daily life problems human judgments typically involve both the positive and negative aspects of events. Reflecting this perspective, bipolar fuzzy sets (BFSs) [33] was introduced by W.-R.Zhang. BFSs was described as $A^* = \{x, \xi_A(x), \psi_A(x) | x \in A\}$ where $\xi_A(x) : A \to [0, 1]$ and $\psi_A(x) : A \to [-1, 0]$, it further extended the domain of FSs. Here $\xi_A(x)$ measures the DM in the interval $[0, 1]$ and DNMS $\psi_A(x)$ in the interval $[-1, 0]$. Moreover, to extend the capabilities of IFSs by incorporating both positive and negative membership degrees, the term bipolar intuitionistic fuzzy sets (BIFSs) was introduced in [34]. Similarly, Khan et al. [35] extended the term PFSs as bipolar picture fuzzy sets (BPFSs). BPFSs is the most extended form of FSs as depicted in Table 2 while Table 3 highlights the intrinsic superiority of BPFSs.

On the other hand, fuzzy graphs (FGs) theory was initially proposed by Rosenfeld [36], based on Kauffman's ideas [37], in 1973. Thus FGs were first initiated by Rosenfeld [36] based on fuzzy relations (FRs). Afterward, several new terms in the theory of FGs were addressed in [38]. The term interval-valued fuzzy graphs (IVFGs), an extension of FGs, was introduced in [39]. Similarly, the notion of bipolar fuzzy graphs (BFGs) was initiated in 2011 [40]. The notion of intuitionistic fuzzy graphs (IFGs) was first presented in [41]. Paravathi and Karunambigai redefined the concepts of IFGs in [42]. Some new operations on IFGs were described in [43]. Several new concepts in IFGs were explained in [44] and nth type IFGs was introduced by Davvaz et al. [45]. Some novel concepts of intuitionistic fuzzy directed graphs and its applications are investigated in [46]. Ezhilmaran and Sankar [34] combined the idea of

**Table 2**. Generalisation of FSs.

| Authors | References | Introduced Terms |
|---|---|---|
| Zadeh | [26] | Fuzzy Sets |
| W.-R.Zhang | [33] | Bipolar Fuzzy Sets |
| Atanassov | [28] | Intuitionistic Fuzzy Sets |
| Ezhilmaran and K Sankar | [34] | Bipolar Intuitionistic Fuzzy Sets |
| Cuong | [29] | Picture Fuzzy Sets |
| Khan et al. | [35] | Bipolar Picture Fuzzy Sets |

**Table 3**. Intrinsic properties of some fuzzy sets.

| Model | Membership | Non-Membership | Restrain Membership | Bipolar Support |
|---|---|---|---|---|
| FS | Yes | No | No | No |
| IFS | Yes | Yes | No | No |
| PFS | Yes | Yes | Yes | No |
| BFS | Yes | No | No | Yes |
| BIFS | Yes | Yes | No | Yes |
| BPFS | Yes | Yes | Yes | Yes |

bipolarity with IFSs and introduced the term bipolar intuitionistic fuzzy graphs (BIFGs). The generalization of IFGs termed picture fuzzy graphs (PFGs) was initiated in [47]. Subsequently, picture fuzzy directed hypergraphs with applications towards decision-making and managing hazardous chemicals are investigated in [48] Currently, various generalizations of PFGs have been established and numerous concepts from classical graph theory have been explored within them [49,50,52–57]. Recently, Khan et al. [58] presented the notion of bipolar picture fuzzy graphs (BPFGs), the most extended form of FGs (see Table 1 and Fig 1). Moreover, BPFGs extends PFGs by incorporating positive and negative membership, non-membership and neutral membership values, respectively. Hence BPFGs have greater capacity to manage the uncertain circumstances. Additionally, the significance of study in the framework of BPFGs is depicted through Fig 1.

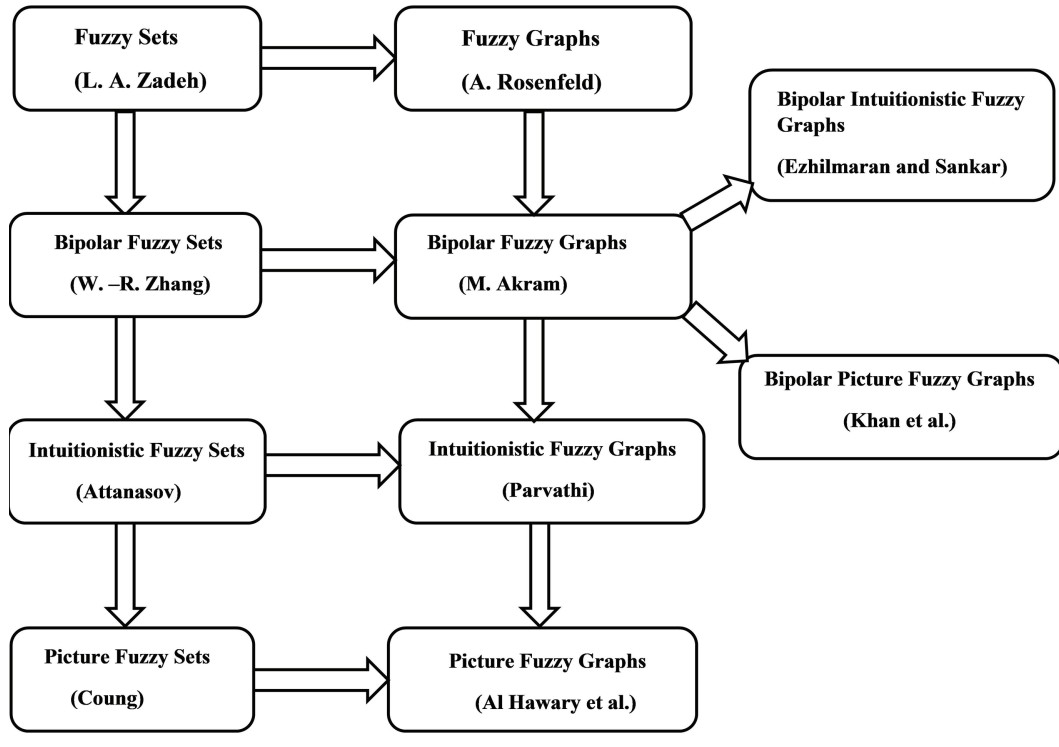

**Fig 1. A view to generalisation both the FSs and FGs.**

## 1.4 Contributions

The main objective of the present study is to introduce a new class of competition graphs named bipolar picture fuzzy digraphs (BPFDGs) – a comprehensive generalization of all prior models. Since the number of competitors may vary sometimes, and keeping these in view, the terms bipolar picture fuzzy k-CG (BPFk-CG), bipolar picture fuzzy p-CG (BPPFp-CG) and bipolar picture fuzzy m-step competition graphs (m-SBPFCG) are also proposed. These graphs appropriately describe various competitions in real-world problems. Applications of these graphs in several fields are also provided. A comparative analysis demonstrating improved decision quality over existing fuzzy graph models is also presented.

## 1.5 Motivations and novelty

Numerous fuzzy set generalizations, such as IFSs, PFSs, BFSs, and their hybrids, have been developed over time. Though they do not fully address contradictory, uncertain, and incomplete information in decision-making situations, each does address ambiguity to some degree. By including degrees of membership, non-membership, and refusal in addition to both positive and negative information poles, bipolar picture fuzzy sets (BPFSs) overcome this limitation. Although some forms of uncertainty are handled by current fuzzy and competition graph models, real-world competition including conflicting or ambiguous assessments is not well modeled by them. By using a six-valued representation, BPFCGs overcome this gap and make it possible to describe supporting, conflicting, and ambiguous interactions all at once. In order to simulate competitive relationships under such multifaceted uncertainty, we expand on this by introducing BPFCGs. In addition to incorporating the advantages of current competition graph models, BPFCGs offer a more comprehensive semantic framework that facilitates accurate and detailed representations. Empirical applications in ecological species interaction modeling and smartphone industry competition show how this approach theoretically and practically expands on the current competition graph theory. The novelty of our study is summarized as follows.

1. A novel structure of BPFCGs and its generalizations are introduced.
2. A comprehensive theoretical framework with new properties are explored.
3. Applications from daily life problems are explored (e.g., smartphone market and ecological systems).

## 1.6 Organization of this study

This article consists of eight sections. Sect 2 contains a useful review of a material relevant to FSs, FGs etc. Sect 3 is divided into three subsections, we first provide a comprehensive discussion on BPFCGs. We then introduce the concepts of BPFk-CGs in Sect 3.1. In Sect 3.2, the notion of BPFp-CGs is introduced. Afterward, a study on m-SBPFCGs is conducted in Sect 3.3. Similarly, Sect 4 comprises four subsections. In Sect 4.1, the analysis of competition among several cell phones companies through BPFCGs is provided. In Sect 4.2, airline system is investigated through BPFk-CGs. In Sect 4.3, an application of BPFp-CGs in networking systems is provided. Application of BPFm-step competition graphs in ecosystem is explored in Sect 4.4. Moreover, a computational framework for the respective applications is also established. In Sect 5, we provide a comparative study demonstrating that the proposed models are more effective than existing ones. Moreover, a real-life scenario is also addressed in Sect 6. Some limitations of our proposed models based on BPFCGs are also highlighted in Sect 7. Finally, Sect 8 contains concluding remarks and outlines potential future directions of this study.

## 2 Preliminaries

In this section, we provide a comprehensive review of the literature related to FSs, FGs, and some of their extensions, which will be useful for understanding the subsequent sections. For the basics of classical graph theory readers may consult [8].

**Definition 1.** [26] FSs on $U$ is defined as

$$S = \{p, \xi_S(p) : p \in U\}$$

where $\xi_S(p)$ is a DMS function defined on $U$ and $\xi_S(p) \in [0, 1]$. Geometrical representation of FSs is given in Fig 2.

A generalization of FSs termed IFSs can be described as.

**Definition 2.** [28] IFSs $S$ on $U$ is given by

$$S = \{(p, \xi_S(p), \chi_S(p)) : p \in U\}$$

where $\xi_S(p)$ is DMS and $\chi_S(p)$ is a DNMS of $p \in U$, defined as $\xi_S(p), \chi_S(x) : U \to [0, 1]$ with

$$0 \leq \xi_S(p) + \chi_S(p) \leq 1$$

for each $p \in U$. Geometrically description of IFSs is given in Fig 3.

A direct extension of IFSs, named PFSs, is defined as.

**Definition 3.** [29] PFSs $S$ on $U$ can be described as

$$S = \{(p, \xi_S(p), \psi_S(p), \chi_S(p)) : p \in U\}$$

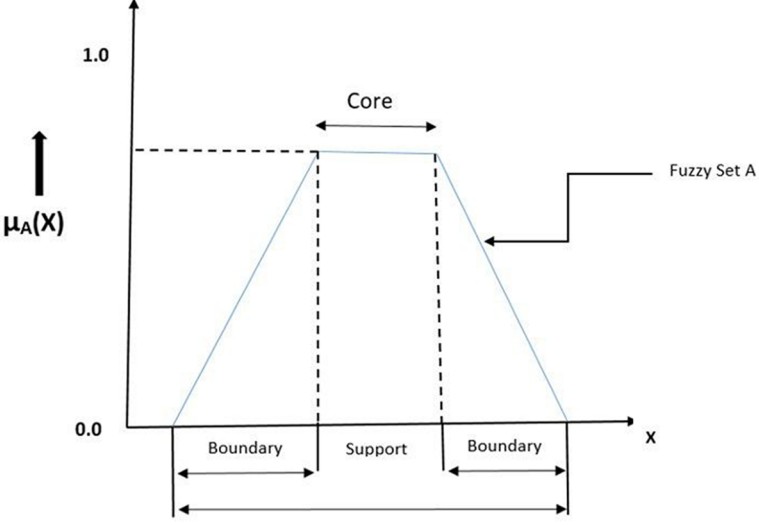

**Fig 2. Visual representation of FS.**

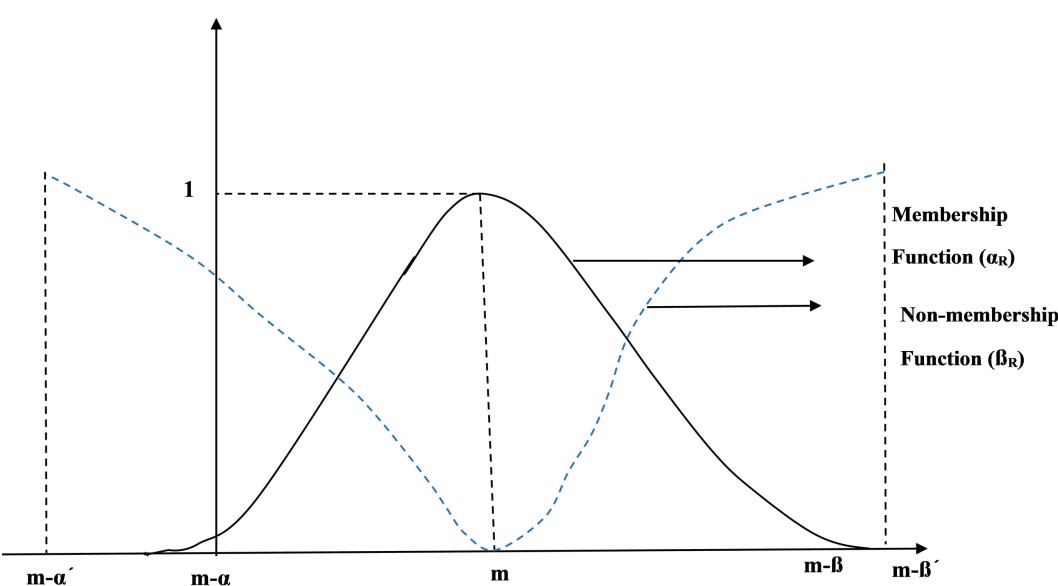

**Fig 3. Visual representation of IFS.**

where $\xi_S(p)$ is DMS, $\psi_S(p)$ is DRMS and $\chi_S(p)$ is DNMS of $p \in U$ are given by $\xi_S(p), \psi_S(p),$ $\chi_S(p) : U \rightarrow [0,1]$ satisfying

$$0 \leq \xi_S(p) + \psi_S(p) + \chi_S(p) \leq 1$$

for all $p \in U$. PFSs is depicted in Fig 4.

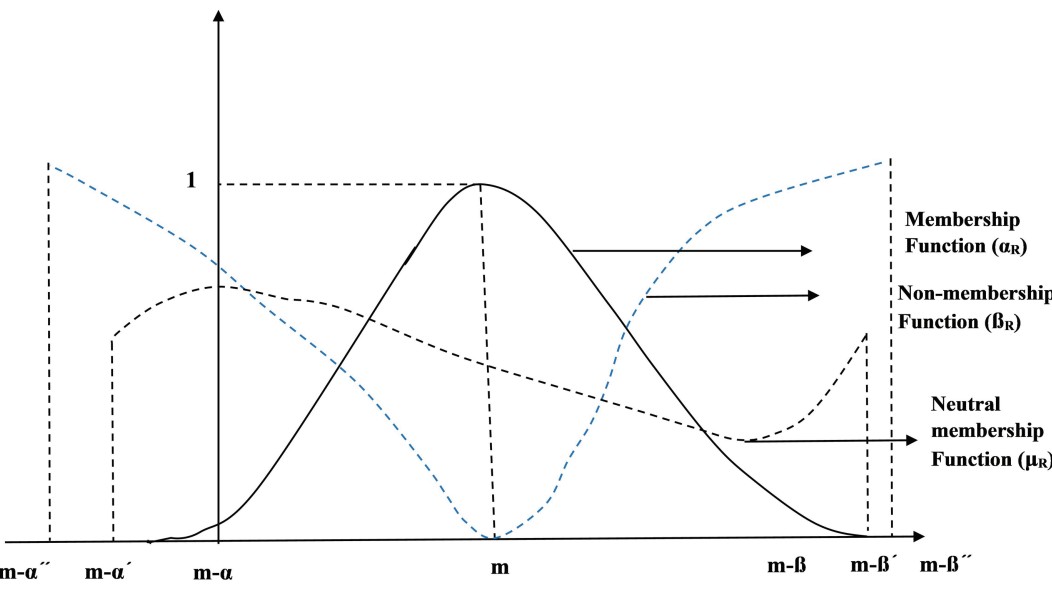

**Fig 4. Visual representation of PFS.**

Hereafter, we provide several further extensions of FSs, such as BFSs, BIFSs and BPFSs.

**Definition 4.** [40] BFSs $S$ on $U$ is given by

$$\{p, \xi_S^+(p), \xi_S^-(p) : p \in U\}$$

where $\xi_S^+(p)$ is positive DMS and $\xi_S^-(p)$ is negative DMS of $p \in U$, defined as $\xi^+ : U \to [0,1]$, $\xi^- : U \to [-1,0]$.

**Definition 5.** [34] BIFSs $S$ on $U$ given by

$$S = \{(p, \xi_S^+(p), \xi_S^-(p), \chi_S^+(p), \chi_S^-(p)) : p \in U\}$$

where $\xi_S^+(p)$ is positive DMS, $\xi_S^-(p)$ is negative DMS, $\chi_S^+(p)$ is positive DNMS and $\chi_S^-(p)$ is negative DNMS of $p \in U$ defined as $\xi_S^+(p) : U \to [0,1]$, $\xi_S^-(p) : U \to [-1,0]$, $\chi_S^+(p) : U \to [0,1]$ and $\chi_S^-(p) : U \to [-1,0]$, with

$$0 \le \xi_S^+(p) + \chi_S^+(p) \le 1$$
$$-1 \le \xi_S^-(p) + \chi_S^-(p) \le 0$$

for all $p \in U$.

**Definition 6.** [35] BPFSs $S$ on $U$ is defined as

$$S = \{(p, \xi_S^+(p), \xi_S^-(p), \psi_S^+(p), \psi_S^-(p), \chi_S^+(p), \chi_S^-(p)) : p \in U\}$$

where $\xi_S^+(p)$ is positive DMS, $\xi_S^-(p)$ is negative DMS, $\psi_S^+(p)$ is positive DRMS while $\psi_S^-(p)$ is negative DRMS and $\chi_S^+(p)$ is positive DNMS and $\chi_S^-(p)$ is negative DNMS of $p \in U$ given by $\xi_S^+(p) : U \to [0,1]$, $\psi_S^+(p) : U \to [0,1]$, $\chi_S^+(p) : U \to [0,1]$ and $\xi_S^-(p) : U \to [-1,0]$, $\psi_S^-(p) : U \to [-1,0]$, $\chi_S^-(p) : U \to [-1,0]$, with

$$0 \le \xi_S^+ + \psi_S^+ + \psi_S^+ + \chi_S^+ \le 1$$
$$-1 \le \xi_S^- + \psi_S^- + \psi_S^- + \chi_S^+ \le 0$$

$\forall p \in U$.

Likewise, we present the several extended forms of FGs from the existing literature.

**Definition 7.** [40] BFGs on a crisp graph $G = (V, E)$ is a collection $\tilde{G} = (R, S)$, where $R = \{\xi_R^+, \xi_R^-\}$ be a BFS on $V$ and $S = \{\xi_S^+, \xi_S^-\}$ be a BFR on $E$ such that

$$\xi_S^+(p,q) \le \{\xi_R^+(p) \wedge \xi_R^+(q)\}$$
$$\xi_S^-(p,q) \le \{\xi_R^-(p) \vee \xi_R^-(q)\}$$

for all $p, q$ in $E$.

**Definition 8.** [34] BIFGs on a crisp graph $G = (V, E)$ is a collection $\tilde{G} = (R, S)$, where $R = \{\xi_R^+, \xi_R^-, \chi_R^+, \chi_R^-\}$ is a BIFS on $V$ and $S = \{\xi_S^+, \xi_S^- \chi_S^+, \chi_S^-\}$ is a BIFR on $E$, satisfying

$$\xi_S^+(p,q) \le \{\xi_R^+(p) \wedge \xi_R^+(q)\}$$
$$\xi_S^-(p,q) \le \{\xi_R^-(p) \vee \xi_R^-(q)\}$$
$$\chi_S^+(p,q) \le \{\chi_R^+(p) \vee \chi_R^+(q)\}$$
$$\chi_S^-(p,q) \le \{\chi_R^-(p) \wedge \chi_R^-(yq)\}$$

for all $p, q \in E$.

**Definition 9.** [58] BPFGs on a crisp graph $G = (V, E)$ is a collection $\tilde{G} = (R, S)$, where $R = \{\xi_R^+, \xi_R^-, \psi_R^+, \psi_R^-, \chi_R^+, \chi_R^-\}$ is a BPFS on $V$ and $S = \{\xi_S^+, \xi_S^- \chi_S^+, \chi_S^-, \psi_S^+, \psi_S^-\}$ is a BPFR on $E$ in such a way that

$$\xi_S^+(p,q) \le \{\xi_R^+(p) \wedge \xi_R^+(q)\}$$
$$\xi_S^-(p,q) \le \{\xi_R^-(p) \vee \xi_R^-(q)\}$$
$$\psi_S^+(p,q) \le \{\psi_R^+(p) \wedge \psi_R^+(q)\}$$
$$\psi_S^-(p,q) \le \{\psi_R^-(p) \vee \psi_R^-(q)\}$$
$$\chi_S^+(x,q) \le \{\chi_R^+(p) \vee \chi_R^+(q)\}$$
$$\chi_S^-(p,q) \le \{\chi_R^-(p) \wedge \chi_R^-(q)\}$$

for all $x, q \in E$.

**Example 10.** Graph shown in Fig 5 is a BPFG.

**Definition 11.** [51] A pair $\vec{G} = (\xi_V, \vec{\xi_E})$ is a FDG on $G = (V, E)$, where $\xi_V$ is a function of MS for vertex set and $\vec{\xi_E}$ is a MS function of directed edges defined as $\xi_V : U \to [0,1]$ and $\vec{\xi_E} : U \times U \to [0,1]$. With

$$\vec{\xi_E}(p,q) \le \xi_V(p) \wedge \xi_V(q)$$

$\forall p, q \in U$. The graph shown in Fig 6 is a FDG.

**Definition 12.** [10] FON for a vertex $p$ in $\vec{G}$ is $N^o(p) = \{U_p^o, \xi_p^o\}$, where $U_p^o = \{q : \xi_B \overrightarrow{(p,q)} > 0\}$ and $\xi_x^o : U_p^o \to [0,1]$ are defined as $\xi_p^o(q) = \xi_B(\overrightarrow{p,q})$. Similarly, FIN of a vertex $p$ in $\vec{G}$ is $N^i(p) = \{U_P^i, \xi_p^i\}$, where $U_p^i = \{q : \xi_B\overrightarrow{(p,q)} > 0\}$ and $\xi_p^i : Y_p^i \to [0,1]$ are defined as $\xi_p^i(q) = \xi_B(\overrightarrow{p,q})$.

**Definition 13.** [10] Let $\vec{G} = (V, \xi_R)$ be a FDG. The FCG for $\vec{G}$ is expressed as $C(\vec{G})$ which is an undirected FG $G = (V, \xi_R, \xi_T)$ having similar set of vertices and $T$ be a set of edges in $C(\vec{G})$ in such a way that there will be an edge between two vertices $p$ and $q$ $(p, q \in V)$ in $C(\vec{G})$ if and only if $N^o(p) \cap N^o(q) \ne \varnothing$. The DMS of the edge $(p, q)$ in $C(\vec{G})$ can be determined by $\xi_S(p,q) = (\xi_R(p) \wedge \xi_R(q))h(E(N_E^o(p)nN_E^o(q)))$, where $E(N_E^o(p)nN_E^o(q))$ is the arrangement of edges occurrence to the vertices of $N_E^o(p)nN_E^o(q)$.

**Example 14.** Let us consider a FDG $\vec{G}$ given in Fig 7(a). The DMS of vertices are $\{a(0.3), b(0.8), c(0.9), d(0.4), (e0.6), f(0.2), g(0.1)\}$. The set of edges with DMS be $\xi_Y \overrightarrow{(a,d)} = 0.5$, $\xi_Y \overrightarrow{(b,a)} = 0.6$, $\xi_Y \overrightarrow{(b,c)} = 0.45$, $\xi_Y \overrightarrow{(d,c)} = 0.7$, $\xi_Y \overrightarrow{(d,e)} = 0.8$, $\xi_Y \overrightarrow{(f,a)} = 0.25$, $\xi_Y \overrightarrow{(f,e)} = 0.36$, $\xi_Y$

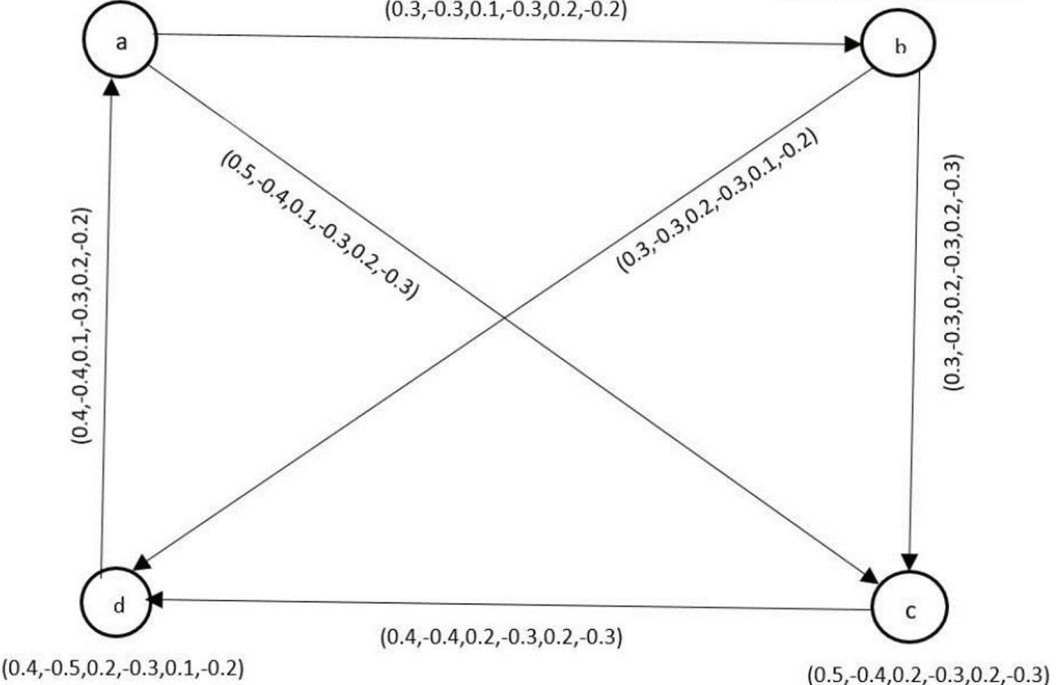

**Fig 5. Bipolar picture fuzzy graph.**

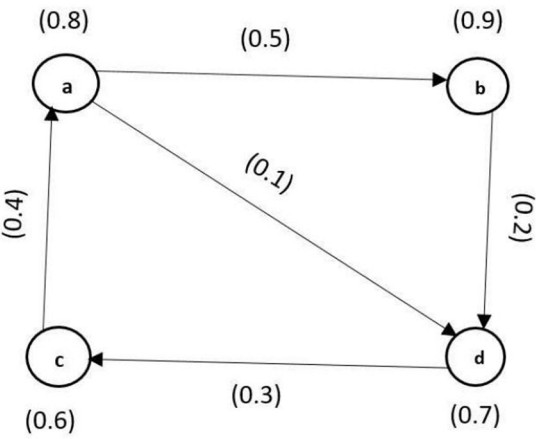

**Fig 6. Fuzzy directed graph.**

$\overrightarrow{(f,g)}$ = 0.9. Observe that $a$ is a common neighbourhood of $b$ and $f$, so vertices $b$ and $f$ will be connected by an undirected edge in $C(\vec{G})$. Essentially, the vertices $b$ and $d$ are contenders and $d$ and $f$ are contenders. In this way $(b,d)$ and $(d,f)$ are two different edges of $C(\vec{G})$. The DMS of the edge $(b,d)$ is $\xi_X(b) \wedge \xi_X(d) \times (E(N^o(b)nN^o(d))) = 0.4 \times max(0.7,0.4) = 0.28$ since $N^o(b)nN^o(d) = (c,0.9)$. In this way, DMS of other edges can be calculated. The FCG of this directed graph is depicted in Fig 7(b).

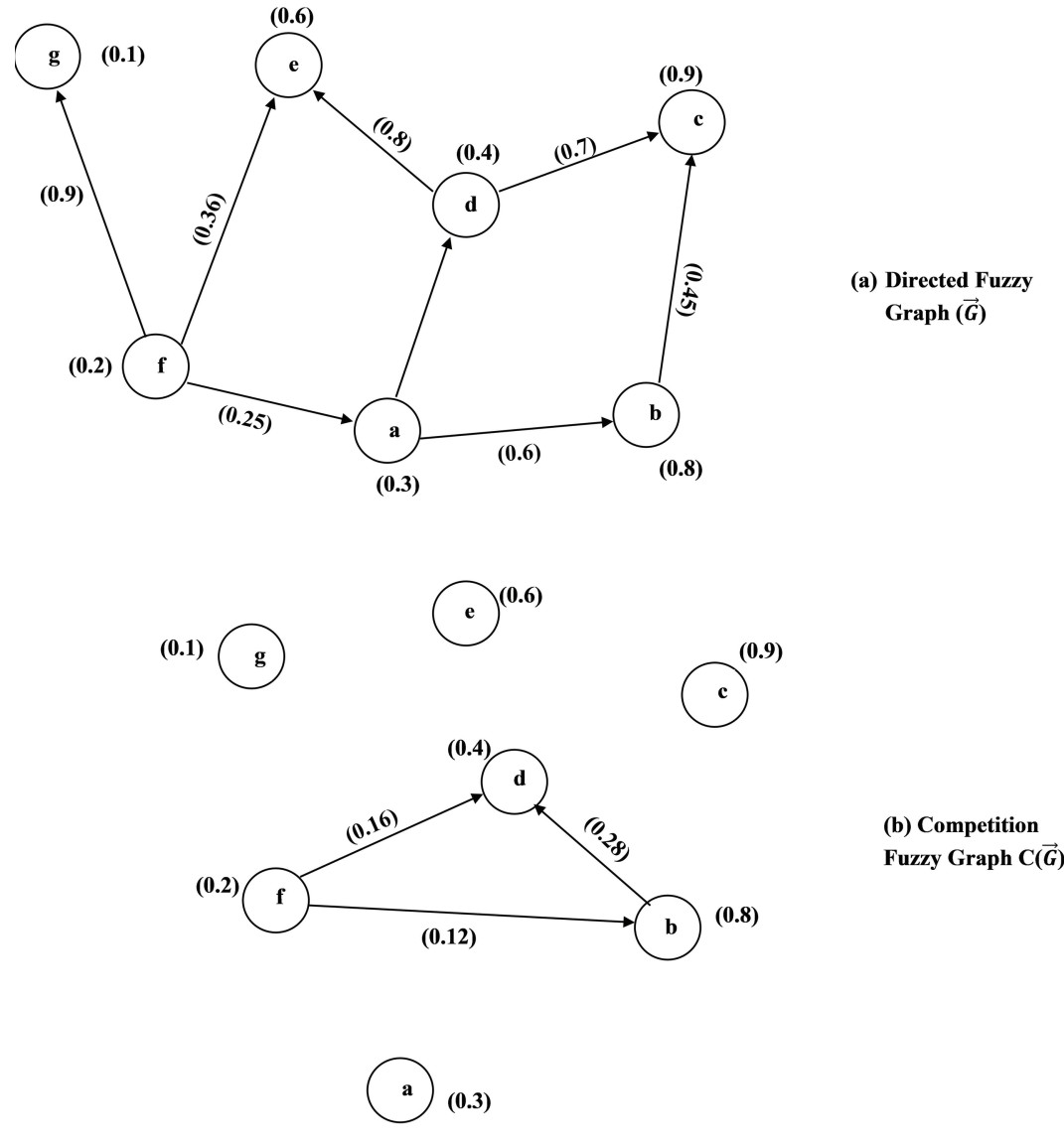

(a) Directed Fuzzy Graph ($\vec{G}$)

(b) Competition Fuzzy Graph C($\vec{G}$)

**Fig 7. Example of FCG.**

**Definition 15.** [19] The BFCG of BFDG $\vec{G} = (V, \xi_R, \vec{\xi_S})$ abbreviated as $C(\vec{G})$ is an undirected graph $G = (V, R, T)$ having similar set of vertices and $T$ is edge set of $C(\vec{G})$ in such a way that there is an edge between two vertices $p$ and $q$ $(p, q \in V)$ in $C(\vec{G})$ if and only if $N^o(p) \cap N^o(q) \neq \varnothing$. The DMS and DNMS values of the edge $(p, q)$ in $C(\vec{G})$ is determined by

$$\xi_T^+(p, q) = \{\xi_R^+(p) \wedge \xi_R^+(q)\} h_\xi(N^o(p)nN^o(q))$$
$$\xi_T^-(p, q) = \{\xi_R^-(p) \vee \xi_R^-(q)\} h_\xi(N^o(p)nN^o(q))$$

where $E(N_E^o(p)nN_E^o(q))$ is the arrangement of edges occurrence to the vertices of $N_E^o(p)nN_E^o(q)$.

**Definition 16.** [23] The BIFCG of BIFDG $\vec{G} = (V, \xi_R, \vec{\xi_S})$ represented by $\mathbf{C}(\vec{G})$ is a graph with undirected edges $G = (V, R, T)$ having same set of vertices and $T$ is edge set of $C(\vec{G})$ in such a way that there will be an edge between two vertices $p$ and $q$ $(p, q \in V)$ in $C(\vec{G})$ if and only if $N^o(p) \cap N^o(q) \neq \varnothing$. The DMS and DNMS values of the edge $(x, y)$ in $C(\vec{G})$ is determined by

$$\xi_T^+(p, q) = \{\xi_R^+(p) \wedge \xi_R^+(q)\} h_\xi(N^o(p) n N^o(q))$$
$$\xi_T^-(p, q) = \{\xi_R^-(p) \vee \xi_R^-(q)\} h_\xi(N^o(p) n N^o(q))$$
$$\chi_T^+(p, q) = (\chi_R^+(p) \vee \chi_R^+(q)) h_\chi(N^o(p) n N^o(q))$$
$$\chi_T^-(p, q) = (\chi_R^-(p) \wedge \chi_R^-(q)) h_\chi(N^o(p) n N^o(q))$$

where $E(N_E^o(p) n N_E^o(q))$ is the arrangement of edges occurrence to the vertices of $N_E^o(p) n N_E^o(yq)$.

**Definition 17.** [17] Let $k$ be a non-negative number. The k-FCG $C_k(\vec{G})$ of a FDG $\vec{G} = (V, R, \vec{S})$ is an undirected FG $G = (V, R, T)$ that has the same fuzzy vertex set as $\vec{G}$ and $T$ as a set of edges in $C_k(\vec{G})$. There is an edge between two vertices $p, q \in V in C_k(\vec{G})$ if and only if $|N^o(p) \cap N^o(q)| > k$. The edge $(p, q) in C_k(\vec{G})$ is $\xi_T(p, q) = \frac{l-k}{l}[\xi_R(p) \wedge \xi_R(q)] h(N^o + (p) \cap N^o(q))$, where $l = |N^o(p) \cap N^o(q)|$.

**Definition 18.** [17] For a given positive integer $p$ and a FDG $\vec{G} = (V, R, \vec{R})$, the p-FCG is a undinrected FG $C_p(\vec{G}) = (V, R, T)$ with same set of vertices and $T$ as a set of edges in $C_p(\vec{G})$. There is an edge between the vertices $p, q \in V$ in $C_p(\vec{G})$ if and only if $|supp(N^o(p) \cap N^o(q))| \geq p$ and the DMS of the edge $(p, q)$ in $C_p(\vec{G})$ is determined by $\xi_R(p, q) = \frac{(n-p)+1}{n}[\xi_X(p) \wedge \xi_X(q)] h(N^o(p) \cap N^o(q))$, where $n = |supp(N^o(p) \cap N^o(q))|$.

**Definition 19.** [16] Let $\vec{G} = (V, R, S)$ be a FDG. The m-SFCG of FDG is a collection $C_m(\vec{G})$, an undirected graph defined as $C_m(\vec{G}) = (V, R, S)$, where $S$ is set of edges in $C_m(\vec{G})$. The MS value of an edge $(p, q)$ can be determined by $\xi_S(p, q) = \xi_R(p) \wedge \xi_R(q) h(N_m^o(p) \cap N_m^o(q))$ for all $p, q \in V$.

## 3 Bipolar picture fuzzy Competition graphs (BPFCGs)

We commence this section by introducing the notion of bipolar picture fuzzy digraphs (BPFDGs). Following this, we initiate the concepts of bipolar picture fuzzy out-neighbourhood and bipolar picture fuzzy in-neighbourhood. Subsequently, we introduce the concepts of BPFCGs and discuss several important results and fascinating properties. Throughout, a triplet $\vec{G} = (V, R, \vec{S})$ represents a BPFDG defined on the vertex set $V$.

**Definition 20.** A BPFDG is a pair $\vec{G} = (V, R, \vec{S})$, where $\{\xi_R^+, \xi_R^-, \psi_R^+, \psi_R^-, \chi_R^+, \chi_R^-\}$ is a BPFS in $U$ and $\vec{S} = \{\xi_S^+, \xi_S^- \psi_S^+, \psi_S^-, \chi_S^+, \chi_S^-\}$ is a bipolar picture fuzzy relation (BPFR) on $U$ such that

$$\xi_S^+(\overrightarrow{pq}) \leq \xi_R^+(p) \wedge \xi_R^+(q)$$
$$\xi_S^-(\overrightarrow{pq}) \leq \xi_R^-(p) \vee \xi_R^-(q)$$
$$\psi_S^+(\overrightarrow{pq}) \leq \psi_R^+(p) \wedge \psi_R^+(q)$$
$$\psi_S^-(\overrightarrow{pq}) \leq \psi_R^-(p) \vee \psi_R^-(q)$$

$$\chi_{\vec{S}}^{+}(\overrightarrow{pq}) \leq \chi_{R}^{+}(p) \vee \chi_{R}^{+}(q)$$

$$\chi_{\vec{S}}^{-}(\overrightarrow{pq}) \leq \chi_{R}^{-}(p) \wedge \chi_{R}^{-}(q)$$

for all $p, q \in R$.

**Example 21.** Let $\vec{G} = (V, R, \vec{S})$ be a BPFDG in Fig 8, where $V = (p, q, r, s, t, u, v)$ is the classical set of vertices with $R = \{(p, (0.1, -0.7, 0.6, -0.3, 0.7, -0.2)), (q, (0.5, -0.7, 0.3, -0.1, 0.6, -0.4)),$ $(r, (0.19, -0.3, 0.7, -0.2, 0.6, -0.7)), (s, (0.4, -0.8, 0.1, -0.4, 0.6, -0.1)), (t, (0.1, -0.4, 0.6, -0.8,$ $0.4, -0.1)), (u, (0.5, -0.3, 0.4, -0.3, 0.9, -0.2)), (v, (0.3, -0.1, 0.4, -0.7, 0.1, -0.5))\}$ and $\vec{S} = \{((\overrightarrow{p, q}), 0.5, -0.8, 0.4, -0.2, 0.1, -0.7)), ((\overrightarrow{p, t}), (0.4, -0.1, 0.4, -0.9, 0.3, -0.7)), ((\overrightarrow{q, v}),$ $(0.3, -0.5, 0.7, -0.3, 0.3, -0.5)), ((\overrightarrow{r, p}), (0.3, -0.6, 0.2, -0.7, 0.1, -0.5)), ((\overrightarrow{r, u}), (0.5, -0.9, 0.1,$ $-0.6, 0.4, -0.7)), ((\overrightarrow{s, p}), (0.7, -0.3, 0.6, -0.1, 0.4, -0.2)), ((\overrightarrow{t, s}), (0.5, -0.9, 0.3, -0.2, 0.7, -0.1)),$ $((\overrightarrow{u, s}), (0.2, -0.4, 0.5, -0.7, 0.4, -0.6)), ((\overrightarrow{u, t}), (0.4, -0.8, 0.3, -0.5, 0.3, -0.1)),$ $((\overrightarrow{v, t}), (0.5, -0.9, 0.2, -0.6, 0.1, -0.7)).$

**Definition 22.** The order $O(G)$ of a BPFCG $G = (V, R, S)$ is described as $O(G) = \{O_{\xi}(G), O_{\psi}$ $(G), O_{\chi}(G)\}$, where

$$O_{\xi}(G) = (\sum_{p \in V} O_{\xi^{+}(G)}, \sum_{p \in V} O_{\xi^{-}(G)})$$

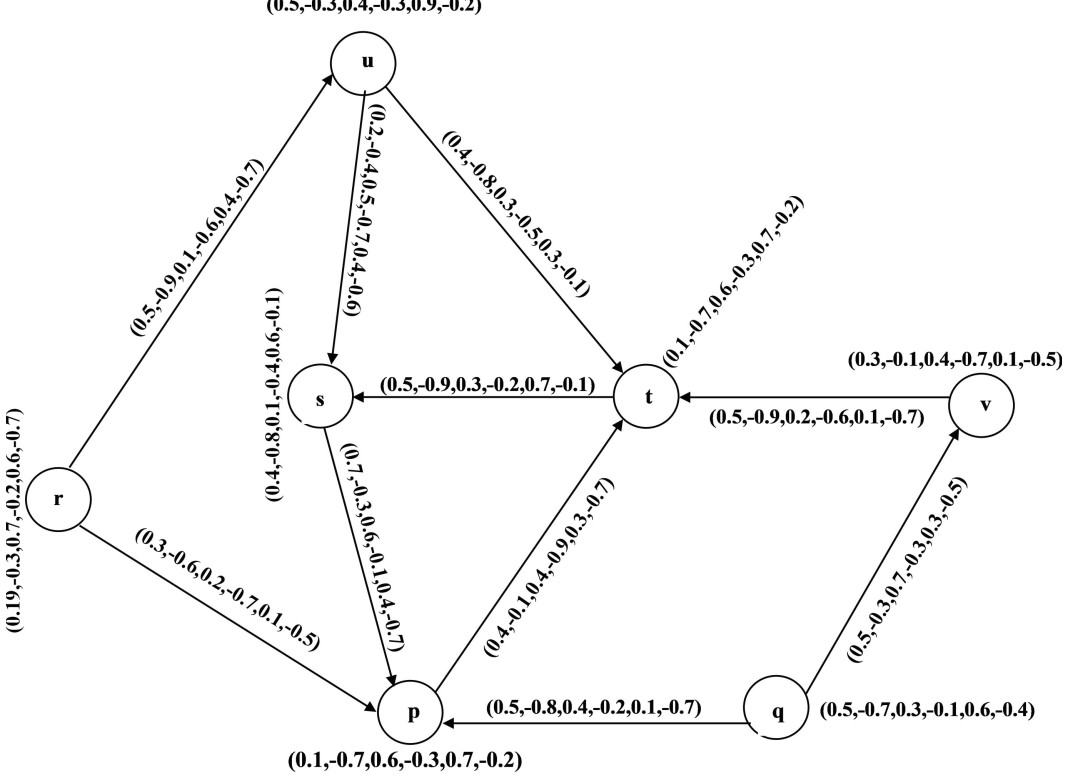

**Fig 8. Bipolar picture fuzzy diagraph.**

$$O_\psi(G) = \left(\sum_{p\in V} O_{\psi^+(G)}, \sum_{p\in V} O_{\psi^-(G)}\right)$$

$$O_\chi(G) = \left(\sum_{p\in V} O_{\chi^+(G)}, \sum_{p\in V} O_{\chi^-(G)}\right).$$

**Definition 23.** The size $S(G)$ of a BPFCG $G = (V, R, S)$ is described as $S(G) = \{S_\xi(G), S_\psi(G), S_\chi(G)\}$, where

$$S_\xi(G) = \left(\sum_{p\in V} S_{\xi^+(G)}, \sum_{p\in V} S_{\xi^-(G)}\right)$$

$$S_\psi(G) = \left(\sum_{p\in V} S_{\psi^+(G)}, \sum_{p\in V} S_{\psi^-(G)}\right)$$

$$S_\chi(G) = \left(\sum_{p\in V} S_{\chi^+(G)}, \sum_{p\in V} S_{\chi^-(G)}\right).$$

**Definition 24.** The support of BPFS $S = (\xi_S^+, \xi_S^-, \psi_S^+, \psi_S^-, \chi_S^+, \chi_S^-)$ is expressed as $Supp(S) = Supp^+(S) \cup Supp^-(S)$, where $Supp^+(S) = \{p \mid \xi_S^+(p) > 0, \psi_S^+(p) > 0, \chi_S^+(p) > 0\}$, $Supp^-(S) = \{x \mid \xi_S^-(p) < 0, \psi_S^-(p) < 0, \chi_S^-(p) < 0\}$. $Supp^+(S)$ is a positive value of support and $Supp^-(S)$ is the negative value of the support.

**Definition 25.** BPFON of a vertex $p$ of $\vec{G}$ is

$$N^o(p) = \left\{U_p^o, \xi_{N^o(p)}^+, \xi_{N^o(p)}^-, \psi_{N^o(p)}^+, \psi_{N^o(p)}^-, \chi_{N^o(p)}^+, \chi_{N^o(p)}^-\right\},$$

where $U_p^o = \{q : \xi^+\overrightarrow{(p,q)} > 0, \xi^-\overrightarrow{(p,q)} < 0, \psi^+\overrightarrow{(p,q)} > 0, \psi^-\overrightarrow{(p,q)} < 0,$ $\chi^+\overrightarrow{(p,q)} > 0, \chi^-\overrightarrow{(p,q)} < 0$ $\xi_{N^o(p)}^+, \psi_{N^o(p)}^+, \chi_{N^o(p)}^+ : U_p^o \to [0,1]$ and $\xi_{N^o(p)}^-, \psi_{N^o(p)}^-, \chi_{N^o(p)}^- : U_p^o \to [-1,0]$ are defined as $\xi_{N^o(p)}^+(q) = \xi_{\vec{B}}^+\overrightarrow{(p,q)}, \psi_{N^o(p)}^+(q) = \psi_{\vec{B}}^+\overrightarrow{(p,q)}, \chi_{N^o(p)}^+(q) = \chi_{\vec{B}}^+\overrightarrow{(p,q)},$ $\xi_{N^o(p)}^-(q) = \xi_{\vec{B}}^-\overrightarrow{(p,q)}, \psi_{N^o(p)}^-(q) = \psi_{\vec{B}}^-\overrightarrow{(p,q)}, \chi_{N^o(p)}^-(q) = \chi_{\vec{B}}^-\overrightarrow{(p,q)}.$

**Example 26.** BPFON of a vertex $u$ in Fig 8 is $N^o(u) = \{(s, (0.2, -0.4, 0.5, -0.7, 0.4, -0.6)), (t, (0.4, -0.8, 0.3, -0.5, 0.3, -0.1))\}$.

**Definition 27.** BPFIN of a vertex $p$ of $\vec{G}$ is

$$N^i(p) = \left\{U_p^i, \xi_{N^i(p)}^+, \xi_{N^i(p)}^-, \psi_{N^i(p)}^+, \psi_{N^i(p)}^-, \chi_{N^i(p)}^+, \chi_{N^i(p)}^-\right\},$$

where $U_p^i = \{q : \xi^+\overrightarrow{(q,p)} > 0, \xi^-\overrightarrow{(q,p)} < 0, \psi^+\overrightarrow{(q,p)} > 0, \psi^-\overrightarrow{(q,p)} < 0,$ $\chi^+\overrightarrow{(q,p)} > 0, \chi^-\overrightarrow{(q,p)} < 0$ $\xi_{N^i(p)}^+, \psi_{N^i(p)}^+, \chi_{N^i(p)}^+ : U_p^i \to [0,1]$ and $\xi_{N^i(p)}^-, \psi_{N^i(p)}^-, \chi_{N^i(p)}^- : U_p^i \to [-1,0]$ are defined as $\xi_{N^i(p)}^+(q) = \xi_{\vec{B}}^+\overrightarrow{(q,p)}, \psi_{N^i(p)}^+(q) = \psi_{\vec{B}}^+\overrightarrow{(q,p)}, \chi_{N^i(p)}^+(q) = \chi_{\vec{B}}^+\overrightarrow{(q,p)},$ $\xi_{N^i(p)}^-(q) = \xi_{\vec{B}}^-\overrightarrow{(q,p)}, \psi_{N^i(p)}^-(q) = \psi_{\vec{B}}^-\overrightarrow{(q,p)}, \chi_{N^i(p)}^-(q) = \chi_{\vec{B}}^-\overrightarrow{(q,p)}.$

**Example 28.** BPFIN of $u$ in Fig 8 is $N^i(u) = \{(r, (0.5, -0.9, 0.1, -0.6, 0.4, -0.7))\}$.

**Definition 29.** The BPFCG of $\vec{G} = (V, R, \vec{S})$ expressed as $C(\vec{G}) = (V, R, S)$ is an undirected BPFG $G = (V, A, S)$ having same set of vertices $V$ and $S$ be its set of edges. There is an edge between two vertices $p$ and $q$ $(p, q \in V)$ in $C(\vec{G})$ if and only if $N^o(p) \cap N^o(q) \neq \varnothing$. The DMS

and DNMS of the edge (p,q) in $C(\vec{G})$ is determined by

$$\xi_S^+(p,q) = \{\xi_A^+(p) \wedge \xi_A^+(q)\} h_m(N^o(p)nN^o(q))$$
$$\xi_S^-(p,q) = \{\xi_A^-(p) \vee \xi_A^-(q)\} h_m(N^o(p)nN^o(q))$$
$$\psi_S^+(p,q) = \{\psi_A^+(p) \wedge \psi_A^+(q)\} h_{neu}(N^o(p)nN^o(q))$$
$$\psi_S^-(p,q) = \{\psi_A^-(p) \vee \psi_A^-(q)\} h_{neu}(N^o(p)nN^o(q))$$
$$\chi_S^+(p,q) = \{\chi_A^+(p) \vee \chi_A^+(q)\} h_n(N^o(p)nN^o(q))$$
$$\chi_S^-(p,q) = \{\chi_A^-(p) \wedge \chi_A^-(q)\} h_n(N^o(p)nN^o(q))$$

for all $p, q \in V$.

**Computational method to generate BPFCGs:**
1. Get the BPFDG.
2. Find the out-neighbourhood of all the vertices in the BPFDG.
3. Find the common out-neighbourhood of all the vertices in the BPFDG.
4. Draw edges between those vertices which have common neighbourhood.
5. Determine positive and negative DMS, DNMS and RMS by using the formula in the Definition 29.

**Example 30.** Let $\vec{G} = (V, R, \vec{S})$ be a BPFDG in Fig 8, where $V = (p, q, r, s, t, u, v)$ is the classical set of vertices:

$R = \{(p, (0.1, -0.7, 0.6, -0.3, 0.7, -0.2)), (q, (0.5, -0.7, 0.3, -0.1, 0.6, -0.4)), (r, (0.19, -0.3, 0.7, -0.2, 0.6, -0.7)), (s, (0.4, -0.8, 0.1, -0.4, 0.6, -0.1)), (t, (0.1, -0.4, 0.6, -0.8, 0.4, -0.1)), (u, (0.5, -0.3, 0.4, -0.3, 0.9, -0.2)), (v, (0.3, -0.1, 0.4, -0.7, 0.1, -0.5))\}$ and $\vec{S} = \{(\overrightarrow{(p,q)}, (0.5, -0.8, 0.4, -0.2, 0.1, -0.7)), (\overrightarrow{(p,t)}, (0.4, -0.1, 0.4, -0.9, 0.3, -0.7)), (\overrightarrow{(q,v)}, (0.3, -0.5, 0.7, -0.3, 0.3, -0.5)), (\overrightarrow{(r,p)}, (0.3, -0.6, 0.2, -0.7, 0.1, -0.5)), (\overrightarrow{(r,u)}, (0.5, -0.9, 0.1, -0.6, 0.4, -0.7)), (\overrightarrow{(s,p)}, (0.7, -0.3, 0.6, -0.1, 0.4, -0.2)), (\overrightarrow{(t,s)}, (0.5, -0.9, 0.3, -0.2, 0.7, -0.1)), (\overrightarrow{(u,s)}, (0.2, -0.4, 0.5, -0.7, 0.4, -0.6)), (\overrightarrow{(u,t)}, (0.4, -0.8, 0.3, -0.5, 0.3, -0.1)), (\overrightarrow{(v,t)}, (0.5, -0.9, 0.2, -0.6, 0.1, -0.7))\}$. Then

$$N^o(r) = \{(u, (0.5, -0.9, 0.1, -0.6, 0.4, -0.7)), (p, (0.3, -0.6, 0.2, -0.7, 0.1, -0.5))\}$$
$$N^o(s) = \{(p, (0.7, -0.3, 0.6, -0.1, 0.4, -0.2))\}$$
$$N^o(t) = \{(s, (0.5, -0.9, 0.3, -0.2, 0.7, -0.1))\}$$
$$N^o(u) = \{(s, (0.2, -0.4, 0.5, -0.7, 0.4, -0.6)), (t, (0.4, -0.8, 0.3, -0.5, 0.3, -0.1))\}$$
$$N^o(v) = \{(t, (0.5, -0.9, 0.2, -0.6, 0.1, -0.7))\}$$
$$N^o(p) = \{(t, (0.4, -0.1, 0.4, -0.9, 0.3, -0.7))\}$$
$$N^o(q) = \{(p, (0.5, -0.8, 0.4, -0.2, 0.1, -0.7))), (v, (0.3, -0.5, 0.7, -0.3, 0.3, -0.5))\}$$

Therefore, $N^o(s) \cap N^o(t) = \{(p, (0.3, -0.3, 0.2, -0.1, 0.4, -0.7))\}$ $N^o(u) \cap N^o(p) = \{(t, (0.4, -0.1, 0.3, -0.5, 0.3, -0.7))\}$, $N^o(u) \cap N^o(t) = \{(s, (0.2, -0.4, 0.3, -0.2, 0.7, -0.6))\}$, $N^o(u) \cap N^o(v) = \{(t, (0.4, -0.9, 0.2, -0.6, 0.3, -0.7)\}$ and $N^o(v) \cap N^o(p) = \{(t, (0.4, -0.1, 0.2, -0.6, 0.3, -0.7))\}$

Therefore, the competition is among $(s,t)$, $(u,t)$, $(p,u)$, $(u,v)$ and $(p,v)$ in BPFCG, whose membership and non-membership values are given by

$$(u,v) = \{0.03, -0.03, 0.24, -0.24, 0.36, -0.05\}, (s,t) = \{0.019, -0.21, 0.06, -0.06, 0.42, -0.14\},$$
$$(u,t) = \{0.4, -0.24, 0.04, -0.12, 0.54, -0.02\}, (u,p) = \{0.01, -0.12, 0.24, -0.24, 0.36, -0.02\},$$
$$(v,p) = \{0.01, -0.04, 0.24, -0.24, 0.28, -0.05\}.$$

The corresponding BPFCG $C(\vec{G})$ is shown in Fig 9

**Competition strength of BPFCGs:**

In this study, we follow the steps outlined below to determine the competition strength of BPFCGs.

1. Get the BPFG.

2. Find the out-neighbourhood of all the vertices in the BPFCG.

3. Find the $C(\vec{G})$ by following the Definition 29.

If $N^o(u) \cap N^o(v) \neq \varnothing$, then the strength of $(u,v)$ of $C(\vec{G})$ is

$$S(u,v) = \frac{1}{18}\left[\xi_V^+(u) + \xi_V^+(v) + \xi_X^+(uv) + \psi_V^+(u) + \psi_V^+(v) + \psi_X^+(uv) + \chi_V^-(u) + \chi_V^-(v) + \chi_X^-(uv)\right.$$
$$\left. +(9 - \xi_V^-(u) + \xi_V^-(v) + \xi_X^-(uv) - \psi_V^-(u) + \psi_V^-(v) + \psi_X^-(uv) - \chi_V^+(u) + \chi_V^+(v) + \chi_X^+(uv))\right]$$

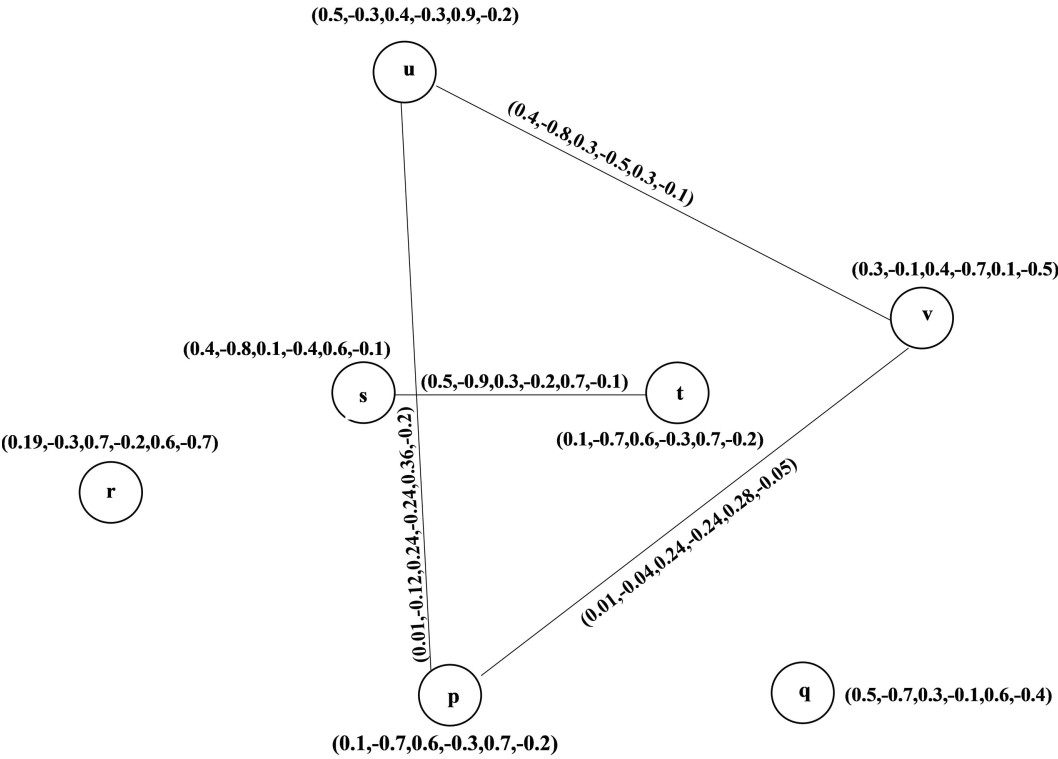

**Fig 9. BPFCG of G.**

Similarly, one can also find the competition strength of two vertices in BPFCG by following the above mentioned steps.

**Example 31.** The strength of competition of every edge in BPFCG of Fig 9 is shown in Table 4.

Strength among $p$ and $v$ is the highest strength and strength among $u$ and $v$ is the lowest strength.

Let $u$ and $v$ be the two vertices of a BPFCG $C(\vec{G}) = (V, R, S)$. If an edge $uv$ satisfies $\frac{1}{2}min\{\xi(u), \xi(v)\} \le \psi(uv)$, then it is called strong edge. Otherwise, we call it a weak edge. Strength of an edge $uv$ in FG can be calculated as $\frac{\psi(uv)}{\xi(u) \wedge \xi(v)}$ [59].

The formula to obtain the strength of an edge $uv$ in BPFDG is designed in this study as follows.

$(\frac{\xi_S^+(uv)}{\xi_R^+(u) \wedge \xi_R^+(v)}, \frac{\xi_S^-(uv)}{\xi_R^-(u) \vee \xi_R^-(v)}, \frac{\psi_S^+(uv)}{\psi_R^+(u) \wedge \psi_R^+(v)}, \frac{\psi_S^-(uv)}{\psi_R^-(u) \vee \psi_R^-(v)}, \frac{\chi_S^+(uv)}{\chi_R^+(u) \vee \chi_R^+(v)}, \frac{\chi_S^-(uv)}{\chi_R^-(u) \wedge \chi_R^-(v)})$. An edge is said to be strong in BPFDG, if

$\frac{\xi_B^+(uv)}{\xi_A^+(u) \wedge \xi_A^+(v)} \ge 0.5$, $\frac{\xi_B^-(uv)}{\xi_A^-(u) \vee \xi_A^-(v)} \le -0.5$, $\frac{\psi_B^+(uv)}{\psi_A^+(u) \wedge \psi_A^+(v)} \ge 0.5$, $\frac{\psi_B^-(uv)}{\psi_A^-(u) \vee \psi_A^-(v)} \le -0.5$,

$\frac{\chi_B^+(uv)}{\chi_A^+(u) \vee \chi_A^+(v)} \le 0.5$ and $\frac{\chi_B^-(uv)}{\chi_A^-(u) \wedge \chi_A^-(v)} \ge -0.5$.

**Theorem 32.** *In BPFDG $\vec{G} = (V, R, \vec{S})$, where $R = \{\xi_R^+, \xi_R^-, \psi_R^+, \psi_R^-, \chi_R^+, \chi_R^-\}$ and $\vec{S}=\{\xi_S^+, \xi_S^-, \psi_S^+,$*

*$\psi_S^-, \chi_B^+, \chi_B^-\}$, if every edge is strong, then $\frac{\xi_S^+(uv)}{\xi_R^+(u) \wedge \xi_R^+(v)} \ge 0.5$, $\frac{\xi_S^-(uv)}{\xi_R^-(u) \vee \xi_R^-(v)} \le -0.5$, $\frac{\psi_S^+(uv)}{\psi_R^+(u) \wedge \psi_R^+(v)} \ge$*

*$0.5$, $\frac{\psi_S^-(uv)}{\psi_R^-(u) \vee \psi_R^-(v)} \le -0.5$, $\frac{\chi_S^+(uv)}{\chi_R^+(u) \vee \chi_R^+(v)} \le 0.5$ and $\frac{\chi_S^-(uv)}{\chi_R^-(u) \wedge \chi_R^-(v)} \ge -0.5$, for each edge in $C(\vec{G})$.*

*Proof*: Let us consider a BPFCG $C(\vec{G})$ of $\vec{G}$ such that each edge of $\vec{G}$ is strong i.e.,

$\frac{\xi_S^+(xy)}{\xi_R^+(x) \wedge \xi_A^+(y)} \ge 0.5$, $\frac{\xi_B^-(xy)}{\xi_A^-(x) \vee \xi_A^-(y)} \le -0.5$, $\frac{\psi_S^+(xy)}{\psi_R^+(x) \wedge \psi_R^+(y)} \ge 0.5$, $\frac{\psi_S^-(xy)}{\psi_R^-(x) \vee \psi_R^-(y)} \le -0.5$,

$\frac{\chi_{SB}^+(xy)}{\chi_R^+(x) \vee \chi_R^+(y)} \le 0.5$ and $\frac{\chi_S^-(xy)}{\chi_R^-(x) \wedge \chi_R^-(y)} \ge -0.5$.

Case 1: Let $N^o(x) \cap N^o(y) = 0$. Then, $C(\vec{G})$ does not exist for $\vec{G}$. Therefore, the proof is not valid is this case.

Case 2: If $N^o(x) \cap N^o(y) \ne 0$, then

$S = \{(x_1, (m_1^+, m_1^-, neu_1^+, neu_1^-, nm_1^+, nm_1^-)), (x_2, (m_2^+, m_2^-, neu_2^+, neu_2^-, nm_2^+, nm_2^-)), ...,$

$(x_k, (m_k^+, m_k^-, neu_k^+, neu_k^-, nm_k^+, nm_k^-))\}$, where $m_i^+, neu_i^+, nm_i^+$ are the positive DMS, DRMS and DNMS and $m_i^-, neu_i^-, nm_i^-$ are the negative DMS, DRMS and DNMS, mutually i = \{1, 2, . . ., k\}.

So, $m_i^+ = min\{\xi_B^+(xy), \xi_B^+(zy)\}$, $m_i^- = max\{\xi_S^-(xy), \xi_S^-(zy)\}$, $neu_i^+ = min\{\xi_S^+(xy), \xi_S^+(zy)\}$, $neu_i^- = max\{\xi_S^-(xy), \xi_S^-(zy)\}$, $nm_i^+ = max\{\xi_S^+(xy), \xi_S^+(zy)\}$ and $nm_i^- = min\{\xi_S^-(xy), \xi_S^-(zy)\}$.

Let $h_m(N^o(x) \cap N^o(y)) = max(m_i^+) = m_{max}$.

**Table 4. Competition strength.**

| Edges | Strength of Competition |
| --- | --- |
| (r,s) | 0.492 |
| (u,v) | 0.475 |
| (p,v) | 0.566 |
| (u,t) | 0.513 |
| (p,u) | 0.541 |

$$\xi_R^+(xy) = (\xi^+(x) \wedge \xi^+(y))h_m(N^o(x) \cap N^o(y)) = (\xi^+(x) \wedge \xi^+(y)) \times m_{max}$$

$$\frac{\xi_R^+(xy)}{(\xi^+(x) \wedge \xi^+(y))} = m_{max}$$

$$\frac{\xi_R^+(xy)}{(\xi^+(x) \wedge \xi^+(y))^2} = \frac{m_{max}}{(\xi^+(x) \wedge \xi^+(y))} \geq 0.5 \qquad (1)$$

$$\xi_R^-(xy) = (\xi^-(x) \vee \xi^-(y))h_m(N^o(x) \cap N^o(y)) = (\xi^+(x) \vee \xi^+(y)) \times m_{max}$$

$$\frac{\xi_R^-(xy)}{(\xi^-(x) \vee \xi^-(y))} = m_{max}$$

$$\frac{\xi_R^-(xy)}{(\xi^-(x) \vee \xi^-(y))^2} = \frac{m_{max}}{(\xi^-(x) \vee \xi^-(y))} \leq -0.5 \qquad (2)$$

Let $h_{neu}(N^o(x) \cap N^o(y)) = \max(neu_i^+) = neu_{max}$.

$$\psi_R^+(xy) = (\psi^+(x) \wedge \psi^+(y))h_{neu}(N^o(xu) \cap N^o(y)) = (\psi^+(x) \wedge \psi^+(y)) \times neu_{max}$$

$$\frac{\psi_R^+(xy)}{(\psi^+(x) \wedge \psi^+(y))} = neu_{max}$$

$$\frac{\psi_R^+(xy)}{(\psi^+(x) \wedge \psi^+(y))^2} = \frac{neu_{max}}{(\psi^+(x) \wedge \psi^+(y))} \geq 0.5 \qquad (3)$$

$$\psi_R^-(xy) = (\psi^-(x) \vee \psi^-(y))h_{neu}(N^o(x) \cap N^o(y) = (\psi^+(x) \vee \psi^+(y)) \times neu_{max}$$

$$\frac{\psi_R^-(xy)}{(\psi^-(x) \vee \psi^-(y))} = neu_{max}$$

$$\frac{\psi_R^-(xy)}{(\psi^-(x) \vee \psi^-(y))^2} = \frac{neu_{max}}{(\psi^-(x) \vee \psi^-(y))} \leq -0.5 \qquad (4)$$

Let $h_{nm}(N^o(x) \cap N^o(y)) = \min\{nm_i^+, i = 1, 2, ..., k\} = nm_{min}$.

$$\chi_R^+(xy) = (\chi^+(x) \vee \chi^+(y))h_{nm}(N^o(x) \cap N^o(y)) = (\chi^+(x) \vee \chi^+(y)) \times nm_{min}$$

$$\frac{\chi_R^+(xy)}{(\chi^+(x) \vee \chi^+(y))} = nm_{min}$$

$$\frac{\chi_R^+(xy)}{(\chi^+(x) \vee \chi^+(y))^2} = \frac{nm_{min}}{(\chi^+(x) \vee \chi^+(y))} \geq 0.5 \qquad (5)$$

$$\chi_R^-(xy) = (\chi^-(x) \wedge \chi^-(y))h_{nm}(N^o(x) \cap N^o(y)) = (\chi^+(x) \wedge \chi^+(y)) \times nm_{min}$$

$$\frac{\chi_R^-(xy)}{(\chi^-(x) \wedge \chi^-(y))} = nm_{min}$$

$$\frac{\chi_R^-(xy)}{(\chi^-(x) \wedge \chi^-(vy)^2)} = \frac{nm_{min}}{(\chi^-(x) \wedge \chi^-(y))} \leq -0.5 \qquad (6)$$

Eqs (1), (2), (3), (4), (5) and (6) show that each edge $xy$ of $C(\vec{G})$ is strong. □

## 3.1 Bipolar picture fuzzy k-competition graphs

This is an extension of BPFCGs, where a new condition is applied to determine the existence of an edge between vertices, based on the concept of competitive power among rivals. Given a non-negative value 'k' and a BPFDG, a BPFCG is constructed, which considers the competitive interactions between vertices with an intensity of at least 'k'.

**Definition 33.** Let $K = \{k_1^+, k_1^-, k_2^+, k_2^-, k_3^+, k_3^-\}$ be the set of real numbers such that $k_1^+, k_2^+, k_3^+ \in [0, 1]$ and $k_1^-, k_2^-, k_3^- \in [-1, 0]$. The BPFk-CG of $\vec{G} = (V, R, \vec{S})$, abbreviated as $C_k(\vec{G}) = (R, T)$, is an undirected graph having the similar arrangement of vertices as in $\vec{G}$, $T$ be the edge set of $C_k(\vec{G})$ and there exists $(p, q)$ if and only if $|N^o(p) \cap N^o(q)|_m^+ > k_1^+$, $|N^o(p) \cap N^o(q)|_m^- < k_1^-$, $|N^o(p) \cap N^o(q)|_{neu}^+ > k_2^+$, $|N^o(p) \cap N^o(q)|_{neu}^- < k_2^-$, $|N^o(p) \cap N^o(q)|_n^+ > k_3^+$ and $|N^o(p) \cap N^o(q)|_n^- < k_3^-$. The DMS, DRMS and DNMS of edge $(p, q)$ can be determined as

$$\xi_T^+(p, q) = \frac{k_m^+ - k_1^+}{k_m^+} \{\xi_R^+(p) \wedge \xi_R^+(q)\} h_\xi(N^o(p) n N^o(q))$$

$$\xi_T^-(p, q) = \frac{k_m^- - k_1^-}{k_m^-} \{\xi_R^-(p) \vee \xi_R^-(q)\} h_\xi(N^o(p) n N^o(q))$$

$$\psi_T^+(p, q) = \frac{k_{neu}^+ - k_2^+}{k_{neu}^+} \{\psi_R^+(p) \wedge \psi_R^+(q)\} h_\psi(N^o(p) n N^o(q))$$

$$\psi_T^-(p, q) = \frac{k_{neu}^- - k_2^-}{k_{neu}^-} \{\psi_R^-(p) \vee \psi_R^-(q)\} h_\psi(N^o(p) n N^o(q))$$

$$\chi_T^+(p, q) = \frac{k_n^+ - k_3^+}{k_n^+} \{\chi_R^+(p) \vee \chi_R^+(q)\} h_\chi(N^o(p) n N^o(q))$$

$$\chi_T^-(p, q) = \frac{k_n^- - k_3^-}{k_n^-} \{\chi_R^-(p) \wedge \chi_R^-(q)\} h_\chi(N^o(p) n N^o(q))$$

for all $p, q \in R$, where $|N^o(p) \cap N^o(q)|_m^+ = k_m^+$, $|N^o(p) \cap N^o(q)|_m^- = k_m^-$, $|N^o(p) \cap N^o(q)|_{neu}^+ = k_{neu}^+$, $|N^o(p) \cap N^o(q)|_{neu}^- = k_{neu}^-$ and $|N^o(p) \cap N^o(q)|_n^+ = k_n^+$, $|N^o(p) \cap N^o(q)|_n^- = k_n^-$.

**Example 34.** Let us take a BPFG given in Fig 10. The set of edges with MS values of this graph are as follows.

$$\vec{su} = (0.3, -0.3, 0.4, -0.4, 0.2, -0.2)$$
$$\vec{sv} = (0.4, -0.3, 0.3, -0.2, 0.8, -0.5)$$
$$\vec{sw} = (0.5, -0.4, 0.4, -0.4, 0.6, -0.2)$$
$$\vec{tu} = (0.4, -0.3, 0.5, -0.3, 0.8, -0.2)$$
$$\vec{tv} = (0.5, -0.4, 0.2, -0.2, 0.7, -0.6)$$
$$\vec{tw} = (0.6, -0.1, 0.3, -0.4, 0.4, -0.8).$$

Thus $N^o(s) = \{(u, (0.3, -0.3, 0.4, -0.4, 0.2, -0.2)), (v, (0.4, -0.3, 0.3, -0.2, 0.8, -0.5)), (w, (0.5, -0.4, 0.4, -0.4, 0.6, -0.2))\}$

$N^o(t) = \{(u, (0.4, -0.3, 0.5, -0.3, 0.8, -0.2)), (v, (0.5, -0.4, 0.2, -0.2, 0.7, -0.6)), (w, (0.6, -0.1, 0.3, -0.4, 0.7, -0.8))\}$

$N^o(s) \cap N^o(t) = \{(u, (0.3, -0.3, 0.4, -0.3, 0.8, -0.2)), (v, (0.4, -0.4, 0.2, -0.2, 0.8, -0.6)), (w, (0.5, -0.4, 0.3, -0.4, 0.7, -0.8))\}$.

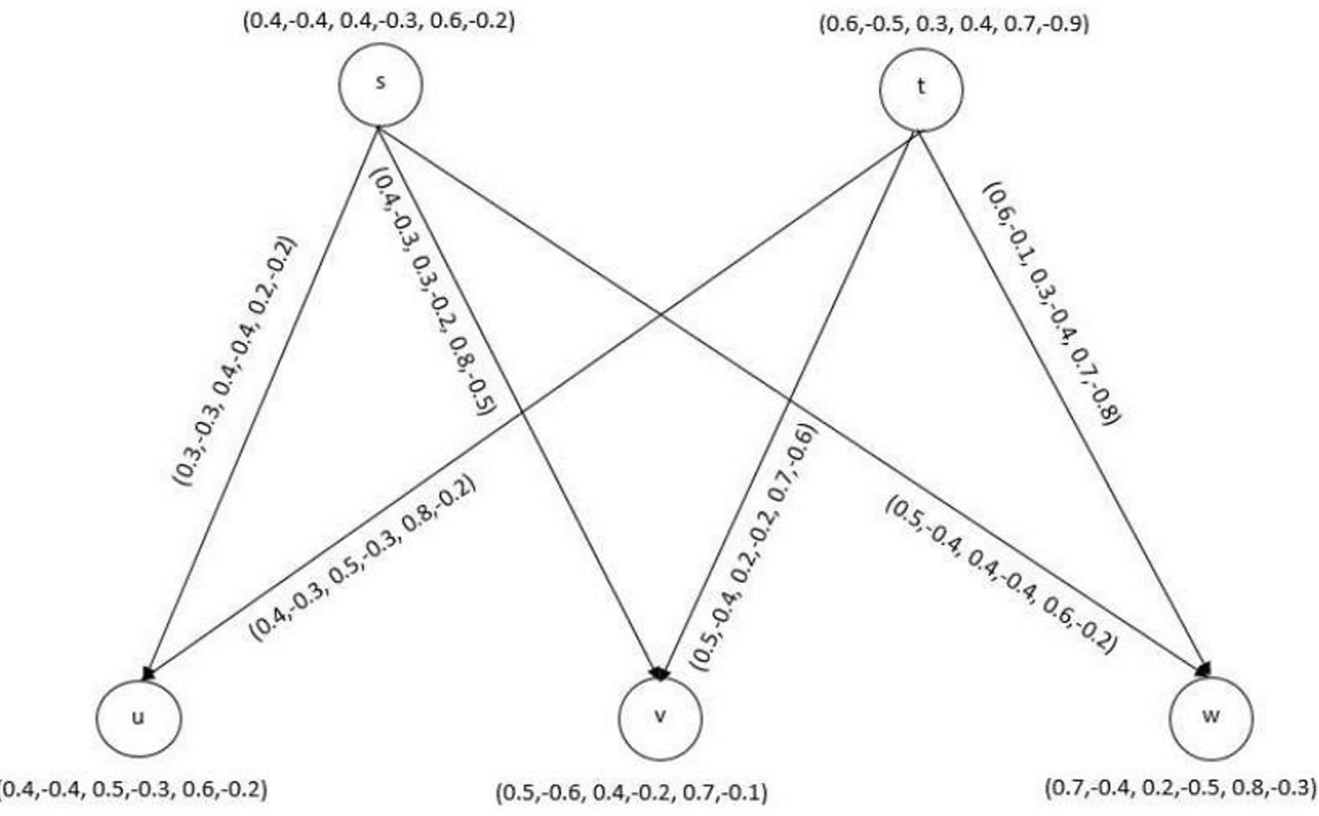

**Fig 10. Bipolar picture fuzzy diagraph.**

Hence

$$k_m^+ = 0.3 + 0.4 + 0.5 = 1.2, k_m^- = -0.3 - 0.4 - 0.4 = -1.1$$

$$k_{neu}^+ = 0.4 + 0.2 + 0.3 = 0.9, k_{neu}^- = -0.3 - 0.2 - 0.4 = -0.9$$

$$k_n^+ = 0.8 + 0.8 + 0.7 = 2.3, k_n^- = -0.2 - 0.6 - 0.8 = -1.6$$

Let $k_1^+ = 0.6$, $k_1^- = -0.4$, $k_2^+ = 0.4$, $k_2^- = -0.5$, $k_3^+ = 0.2$, $k_3^- = -0.5$. Then $\xi_m^+ = 0.13$, $\xi_m^- = -0.13$, $\psi_{neu}^+ = 0.06$, $\psi_{neu}^- = -0.053$, $\chi_n^+ = 0.511$ and $\chi_n^- = -0.49$. Finally, the corresponding BPFk-CG is depicted in Fig 11.

**Theorem 35.** *Let $\vec{G} = (V, R, \vec{S})$ is a BPFDG, where $R = (\xi_R^+, \xi_R^-, \psi_R^+, \psi_R^-, \chi_R^+, \chi_R^-)$ and $\vec{S} = (\xi_{\vec{S}}^+, \xi_{\vec{S}}^-, \psi_{\vec{S}}^+, \psi_{\vec{S}}^-, \chi_{\vec{S}}^+, \chi_{\vec{S}}^-)$. If $h_m(N^o(p) \cap h_o(q)) = 1, h_{neu}(N^o(p) \cap N^o(q)) = 0, h_{nm}(N^o(p) \cap N^o(q)) = 0, |\xi_{N^o(p) \cap N^o(q)}^+| > 2k_1^+, |\xi_{N^o(p) \cap N^o(q)}^-| < 2k_1^-, |\psi_{N^o(p) \cap N^o(q)}^+| > 2k_2^+, |\psi_{N^o(p) \cap N^o(q)}^-| < 2k_2^-, |\chi_{N^o(p) \cap N^o(q)}^+| < 2k_3^+$ and $|\chi_{N^o(p) \cap N^o(q)}^-| > 2k_3^-$, then an edge $(p, q)$ is strong in $C_k(\vec{G})$.*

*Proof*: Assume $C_k\vec{G} = (V, R, T)$ be a BPFK-CG of $\vec{G}$. Suppose $p, q \in R$. Then

$$\xi_T^+(pq) = \frac{k_m^+ - k_1^+}{k_m^+} [\xi_R^+(p) \wedge \xi_R^+(q)] h_m(E(N^o(p) \cap N^o(q)))$$

**Fig 11. Bipolar picture fuzzy k-competition graph.**

Let $h_m(N^o(p) \cap N^o(q)) = 1$

$$(k_m^+)\xi_T^+(pq) = (k_m^+ - k_1^+)[\xi_R^+(p) \wedge \xi_R^+(q)](1) \geq k_1^+[\xi_R^+(p) \wedge \xi_R^+(q)]$$

$$2k_1^+\xi_T^+(pq) \geq (k_1^+)[\xi_R^+(p) \wedge \xi_R^+(q)]$$

$$\xi_m^+(pq) \geq \frac{1}{2}[\xi_R^+(p) \wedge \xi_R^+(q)]$$

$$\frac{\xi_S^+(pq)}{[\xi_R^+(p) \wedge \xi_R^+(q)]} \geq 0.5 \tag{1}$$

$$\xi_T^-(pq) = \frac{k_m^- - k_1^-}{k_m^-}[\xi_R^-(p) \vee \xi_R^-(q)]h_m(E(N^o(p) \cap N^o(q)))$$

$$(k_m^-)\xi_T^-(pq) = (k_m^- - k_1^-)[\xi_R^-(p) \vee \xi_R^-(q)](1) \leq k_1^-[\xi_R^-(p) \vee \xi_R^-(q)]$$

$$2k_1^-\xi_T^-(pq) \leq (k_1^-)[\xi_R^-(p) \vee \xi_R^-(q)]$$

$$\xi_T^-(pq) \leq \frac{1}{2}[\xi_R^-(p) \vee \xi_R^-(q)]$$

$$\frac{\xi_T^-(pq)}{[\xi_R^-(p) \vee \xi_R^-(q)]} \leq 0.5 \tag{2}$$

$$\psi_T^+(pq) = \frac{k_{neu}^+ - k_1^+}{k_{neu}^+}[\psi_R^+(p) \wedge \psi_R^+(q)]h_{neu}(E(N^o(p) \cap N^o(q)))$$
Let $h_{neu}(N^o(p) \cap N^o(q)) = 0$

$$(k_{neu}^+)\psi_T^+(pq) = (k_{neu}^+ - k_2^+)[\psi_R^+(p) \wedge \psi_R^+(q)](0) \geq k_2^+[\psi_R^+(p) \wedge \psi_R^+(q)]$$

$$2k_2^+\psi_T^+(pq) \geq (k_2^+)[\psi_R^+(p) \wedge \psi_R^+(q)]$$

$$\psi_T^+(pq) \geq \frac{1}{2}[\psi_R^+(p) \wedge \psi_R^+(q)]$$

$$\frac{\psi_T^+(pq)}{[\psi_R^+(p) \wedge \psi_R^+(q)]} \geq 0.5 \tag{3}$$

$$\psi_T^-(pq) = \frac{k_{neu}^- - k_2^-}{k_{neu}^-}[\psi_R^-(p) \vee \psi_R^-(q)]h_{neu}(E(N^o(p) \cap N^o(q)))$$

$$(k_{neu}^-)\psi_T^-(pq) = (k_{neu}^- - k_2^-)[\psi_R^-(p) \vee \psi_R^-(q)](0) \leq k_2^-[\psi_R^-(p) \vee \psi_R^-(q)]$$

$$2k_2^-\psi_T^-(pq) \leq (k_2^-)[\psi_R^-(p) \vee \psi_R^-(q)]$$

$$\psi_T^-(pq) \leq \frac{1}{2}[\psi_R^-(p) \vee \psi_R^-(q)]$$

$$\frac{\psi_T^-(pq)}{[\psi_R^-(p) \vee \psi_R^-(q)]} \leq 0.5 \tag{4}$$

$$\chi_T^+(pq) = \frac{k_{nm}^+ - k_3^-}{k_{nm}^-}[\chi_R^+(p) \vee \chi_R^+(q)]h_{nm}(E(N^o(p) \cap N^o(q)))$$

Let $h_{nm}(N^o(p) \cap N^o(q)) = 0$

$$(k_{nm}^+)\chi_T^+(pq) = (k_{nm}^+ - k_3^+)[\chi_R^+(p) \vee \chi_R^+(q)](0) \geq k_3^+[\chi_R^+(p) \vee \chi_R^+(q)]$$

$$2k_3^+\chi_T^+(pq) \geq (k_3^+)[\chi_R^+(p) \vee \chi_R^+(q)]$$

$$\chi_T^+(pq) \geq \frac{1}{2}[\chi_R^+(p) \vee \chi_R^+(q)]$$

$$\frac{\chi_T^+(pq)}{[\chi_R^+(p) \vee \chi_R^+(q)]} \geq 0.5 \tag{5}$$

$$\chi_T^-(pq) = \frac{k_{nm}^- - k_3^-}{k_{nm}^-}[\chi_R^-(p) \wedge \chi_R^-(q)]h_{nm}(E(N^o(p) \cap N^o(q)))$$

$$(k_{nm}^-)\chi_T^-(pq) = (k_{nm}^- - k_3^-)[\chi_R^-(p) \wedge \chi_R^-(q)](0) \leq k_3^-[\chi_R^-(p) \wedge \chi_R^-(q)]$$

$$2k_3^-\chi_T^-(pq) \leq (k_3^-)[\chi_R^-(p) \wedge \chi_R^-(q)]$$

$$\chi_T^-(pq) \leq \frac{1}{2}[\chi_R^-(p) \wedge \chi_R^-(q)]$$

$$\frac{\chi_T^-(pq)}{[\chi_R^-(p) \wedge \chi_R^-(q]} \leq 0.5 \tag{6}$$

Eqs (1), (2), (3), (4), (5), and (6) show that the edge $(pq)$ of $C_k(\vec{G})$ is strong. □

### 3.2 Bipolar Picture Fuzzy p-Competition Graphs (BPFp-CGs)

Another extension of BPFCG is a BPFp-CG which shares similar concepts to BPFk-CGs. The main difference between these two graphs is the value of the parameters: in the BPFk-CGs, $k$ can be any real number which shows the power of competitors, whereas in the BPFp-CG, $p$ is a positive integer which shows the number of competitors. This means that $p$ is a special case of $k$ with more restricted possibilities. If there is a directed walk from $a$ to $b$ in a digraph $\vec{D}$ of length $m$, then $b$ is called an m-step neighbor of $a$. Again, if a vertex $c$ is an m-step neighbor of both vertices $a$ and $b$, then $c$ is called an m-step common neighbor of the vertices $a$ and $b$.

**Definition 36.**     BPFp-CG of $\vec{G} = (V, R, \vec{S})$, abbreviated as $C_p(\vec{G}) = (V, R, X)$ is an undirected graph having the similar arrangement of vertices as in $\vec{G}$, and $X$ is the edges set of $C_p(\vec{G})$ and there exists an edge $(p, q)$ if and only if $|supp(N^o(p) \cap N^o(q)| \geq p$, where $p$ is a positive integer. The DMS of edge $(p, q)$ can be determined as

$$\xi_X^+(p,q) = \frac{(k-p)+1}{k}\{\xi_R^+(p) \wedge \xi_R^+(q)\}h_\xi(N^o(p)nN^o(q))$$

$$\xi_X^-(p,q) = \frac{(k-p)+1}{k}\{\xi_R^-(p) \vee \xi_R^-(q)\}h_\xi(N^o(p)nN^o(q))$$

$$\psi_X^+(p,q) = \frac{(k-p)+1}{k}\{\psi_R^+(p) \wedge \psi_R^+(q)\}h_\psi(N^o(p)nN^o(q))$$

$$\psi_X^-(p,q) = \frac{(k-p)+1}{k}\{\psi_R^-(p) \vee \psi_R^-(q)\}h_\psi(N^o(p)nN^o(q))$$

$$\chi_X^+(p,q) = \frac{(k-p)+1}{k}\{\chi_R^+(p) \vee \chi_R^+(q)\}h_\chi(N^o(p)nN^o(q))$$

$$\chi_X^-(p,q) = \frac{(k-p)+1}{k}\{\chi_R^-(p) \wedge \chi_R^-(q)\}h_\chi(N^o(p)nN^o(q))$$

where $k = |supp(N^o(p) \cap N^o(q))|$.

**Example 37.**     One can easily verify that the graph given in Fig 12 is a BPFp-CG of BPFDG $\vec{G}$ shown in Fig 10.

**Theorem 38.**  Let $\vec{G} = (V, R, \vec{S})$ is a BPFDG, where $R = (\xi_R^+, \xi_R^-, \psi_R^+, \psi_R^-, \chi_R^+, \chi_R^-)$ and $\vec{S} = (\xi_{\vec{S}}^+, \xi_{\vec{S}}^-, \psi_{\vec{S}}^+, \psi_{\vec{S}}^-, \chi_{\vec{S}}^+, \chi_{\vec{S}}^-)$. If $h_m(N^o(p) \cap N^o(q)) = 1, h_{neu}(N^o(p) \cap N^o(q) = 0$ and $h_{nm}(N^o(p) \cap N^o(q) = 0$ in $C_{[\frac{k}{2}]}(\vec{G})$, then the edge $(p, q)$ is strong. Here $k = |supp(N^o(p) \cap N^o(q))|$.

*Proof*: Consider a bipolar picture fuzzy $[\frac{k}{2}]$-competition graph $C_{[\frac{k}{2}]}(\vec{G}) = (V, R, X)$ of $\vec{G}$, where $k = |supp(N^o(p) \cap N^o(q))|$.

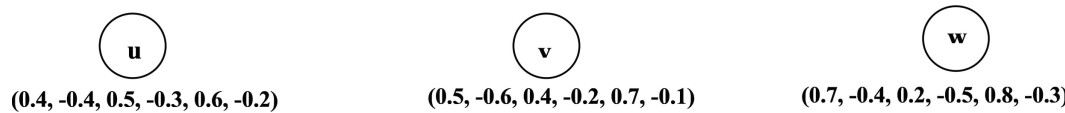

**Fig 12. Bipolar picture fuzzy p-competition graph.**

Assume that $h_m(E(N^o(p) \cap N^o(q))) = 1$, $h_{neu}(E(N^o(p) \cap N^o(q))) = 0$ and $h_{nm}(E(N^o(p) \cap N^o(q))) = 0$, $p, q \in R$. Then

$$\xi_X^+(p,q) = \frac{(k - [\frac{k}{2}]) + 1}{k}[\xi_R^+(p) \wedge \xi_R^+(q)]h_m(E(N^o(p) \cap N^o(q)))$$

$$\xi_X^-(p,q) = \frac{(k - [\frac{k}{2}]) + 1}{k}[\xi_R^-(p) \vee \xi_R^-(q)]h_m(E(N^o(p) \cap N^o(q)))$$

$$\psi_X^+(p,q) = \frac{(k - [\frac{k}{2}]) + 1}{k}[\psi_R^+(p) \wedge \psi_R^+(q)]h_{neu}(E(N^o(p) \cap N^o(q)))$$

$$\psi_X^-(p,q) = \frac{(k - [\frac{k}{2}]) + 1}{k}[\psi_R^-(p) \vee \psi_R^-(q)]h_{neu}(E(N^o(p) \cap N^o(q)))$$

$$\chi_X^+(p,q) = \frac{(k - [\frac{k}{2}]) + 1}{k}[\chi_R^+(p) \vee \chi_R^+(q)]h_{nm}(E(N^o(p) \cap N^o(q)))$$

$$\chi_X^-(p,q) = \frac{(k - \frac{k}{2}) + 1}{k}[\chi_R^-(p) \wedge \chi_R^-(q)]h_{nm}(E(N^o(p) \cap N^o(q)))$$

We have $h_m(N^o(p) \cap N^o(q)) = 1$. Therefore

$$\xi_X^+(p,q) = \frac{(k - [\frac{k}{2}]) + 1}{k}[\xi_R^+(p) \wedge \xi_R^+(q)](1)$$

$$\xi_X^-(p,q) = \frac{(k - [\frac{k}{2}]) + 1}{k}[\xi_R^-(p) \vee \xi_R^-(q)](1)$$

Also, we have $h_{neu}(N^o(p) \cap N^o(q)) = 0$. Hence

$$\psi_X^+(p,q) = \frac{(k - [\frac{k}{2}]) + 1}{k}[\psi_R^+(p) \wedge \psi_R^+(q)](0)$$

$$\psi_X^-(p,q) = \frac{(k - [\frac{k}{2}]) + 1}{k}[\psi_R^-(p) \vee \psi_R^-(q)](0)$$

Also, we have $h_{nm}(N^o(p) \cap N^o(q)) = 0$. Thus

$$\chi_X^+(p,q) = \frac{(k - [\frac{k}{2}]) + 1}{k}[\chi_R^+(p) \vee \chi_R^+(q)](0)$$

$$\chi_X^-(p,q) = \frac{(k - \frac{k}{2}) + 1}{k}[\chi_R^-(p) \wedge \chi_R^-(q)](0)$$

Hence $\frac{\xi_X^+(p,v)}{(\xi_R^+(p) \wedge \xi_R^+(v))} = \frac{(k - [\frac{k}{2}]) + 1}{k} \geq 0.5$, $\frac{\xi_X^-(p,q)}{(\xi_R^-(p) \wedge \xi_R^-(q))} = \frac{(k - [\frac{k}{2}]) + 1}{k} \leq -0.5$, $\frac{\psi_X^+(p,q)}{(\psi_R^+(p) \wedge \psi_R^+(q))} = \frac{(k - [\frac{k}{2}]) + 1}{k} \geq 0.5$, $\frac{\psi_X^-(p,q)}{(\psi_R^-(p) \wedge \psi_R^-(q))} = \frac{(k - [\frac{k}{2}]) + 1}{k} \leq -0.5$, $\frac{\chi_X^+(p,q)}{(\chi_R^+(p) \wedge \chi_R^+(q))} = \frac{(k - [\frac{k}{2}]) + 1}{k} \leq 0.5$, and $\frac{\chi_X^-(p,q)}{(\chi_R^-(p) \wedge \chi_R^-(q))} = \frac{(k - [\frac{k}{2}]) + 1}{k} \geq -0.5$.

This shows that the edge $(p,q)$ of $C_{[\frac{k}{2}]}(\vec{G})$ is strong. $\square$

## 3.3 Bipolar Picture Fuzzy m-step Competition Graphs (m-SBPFCGs)

In this part, we present the idea of another extension of BPFCGs, termed m-SBPFCG. In a CG, there is an edge between any two vertices $p$ and $q$ if there exists a common neighbor. However, in an m-SBPFCG there exists an edge between any of two vertices $p$ and $q$, if these

vertices have a common neighbor whose distance is $m$ from them. Similarly, we call a vertex $q$ an m-step neighbor of $p$, if there exists directed walk of length $m$ from $p$ to $q$ in a digraph $\vec{D}$. Furthermore, if a vertex $r$ is an m-step neighbor of both vertices $p$ and $q$, then $r$ is called an m-step common neighbor of the vertices $p$ and $q$. Before proceeding our discussions, we first explain a few useful notations.

$P^m_{(p,q)}$: Edge between $p$ and $q$ in FG.

$\vec{P}^m_{(p,q)}$: A directed edge from $p$ to $q$ in fuzzy graph.

$N^o_p$: m-step fuzzy out-neighbourhood of vertex $p$.

$N^i_p$: m-step fuzzy in-neighbourhood of vertex $q$.

$C_m(\vec{G})$: m-SFCG of fuzzy digraph $\vec{G}$.

**Definition 39.** An m-SBPFDG of a BPFDG $\vec{G} = (V, R, \vec{S})$ can be described as $\vec{G_m} = (V, R, \vec{U})$, where $R = (\xi^+_R, \xi^-_R, \psi^+_R, \psi^-_R, \chi^+_R, \chi^-_R)$ and $\vec{U} = (\xi^+_{\vec{U}}, \xi^-_{\vec{U}}, \psi^+_{\vec{U}}, \psi^-_{\vec{U}}, \chi^+_{\vec{U}}, \chi^-_{\vec{U}})$ and having a directed edge $(\overrightarrow{p,q})$ in $\vec{G_m}$, if there is a directed BPFP $\vec{P}^m_{(p,q)} in \vec{G}$.

**Example 40.** A 2-SBPFDG with a set of vertices $V = \{r, s, t, u, v, w\}$ is shown in Fig 13.

**Definition 41.** An m-SBPFON of a vertex $p$ of $\vec{G}$ is a BPFS

$N^o_m(p) = (X^o_p, (\xi^+_{N^o(p)}, \xi^-_{N^o(p)}, \psi^+_{N^o(p)}, \psi^-_{N^o(p)}, \chi^+_{N^o(p)}, \chi^-_{N^o(p)}))$, where $X^o_p$={q: there is a directed BPFP $\vec{P}^m_{(p,q)}$ of length $m$ from $p$ to $q$ }, $\xi^+_{N^o(p)}, \psi^+_{N^o(p)}, \chi^+_{N^o(p)}: X^o_p \to [0,1]$ can be defined as

$\xi^+_{N^o(p)}(q) = \min\{\xi^+_{\vec{U}}\overrightarrow{(r,s)}: \overrightarrow{(r,s)}$ is an edge of $\vec{P}^m_{(p,q)}\}, \psi^+_{N^o(p)}(q) = \min\{\psi^+_{\vec{U}}\overrightarrow{(r,s)}: \overrightarrow{(r,s)}$ is an edge of $\vec{P}^m_{(p,q)}\}$ and

$\chi^+_{N^o(p)} = \max\{\chi^+_{\vec{U}}\overrightarrow{(r,s)}: \overrightarrow{(r,s)}$ is an edge of $\vec{P}^m_{(p,q)}\}$ and
$\xi^-_{N^o(p)}, \psi^-_{N^o(p)}, \chi^-_{N^o(p)}: X^o_p \to [-1,0]$ are defined as

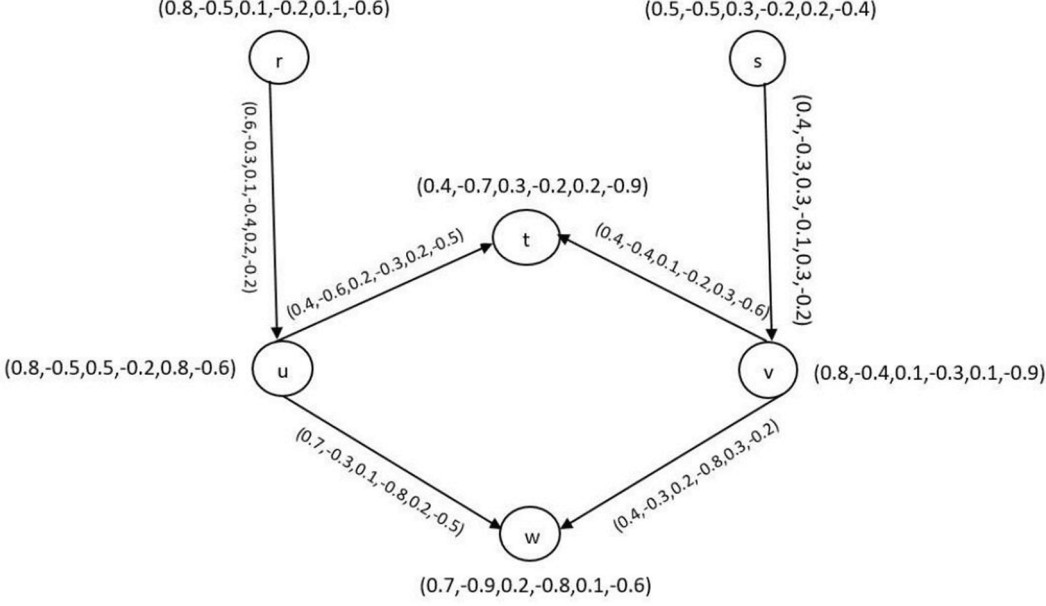

**Fig 13. 2-SBPFDG.**

$$\xi_{N^o(p)}^-(q) = \max\{\xi_{\vec{U}}^- \overrightarrow{(r,s)} : \overrightarrow{(r,s)} \text{ is an edge of } \vec{P}_{(p,q)}^m\}, \psi_{N^o(p)}^-(q) = \max\{\psi_{\vec{U}}^- \overrightarrow{(r,s)} : \overrightarrow{(r,s)} \text{ is an}$$
edge of $\vec{P}_{(p,q)}^m\}$ and $\chi_{N^o(p)}^-(q) = \min\{\chi_{\vec{U}}^- \overrightarrow{(r,s)} : \overrightarrow{(r,s)} \text{ is an edge of } \vec{P}_{(p,q)}^m\}$.

**Example 42.** Consider the 2-SBPFDG shown in Fig 13. Then 2-SBPFON of $r$ and $s$ are

$$N_2^o(r) = \{(t,(0.4,-0.3,0.1,-0.3,0.2,-0.5)),(w,(0.6,-0.3,0.1,-0.4,0.2,-0.6))\}$$

$$N_2^o(s) = \{(t,(0.4,-0.3,0.1,-0.1,0.3,-0.5)),(w,(0.4,-0.3,0.2,-0.1,0.3,-0.2))\}.$$

**Definition 43.** An m-SBPFIN of a vertex $p$ of $\vec{G}$ is a BPFS
$N_m^i(p) = (X_p^i,(\xi_{N^i(p)}^+,\xi_{N^i(p)}^-,\psi_{N^i(p)}^+,\psi_{N^i(p)}^-,\chi_{N^i(p)}^+,\chi_{N^i(p)}^-))$, where $X_p^i = \{q$:there is a directed
BPFP $\vec{P}_{(p,q)}^m$ of length $m$ from $q$ to $p\}$, $\xi_{N^i(p)}^+,\psi_{N^i(p)}^+,\chi_{N^i(p)}^+: X_p^i \to [0,1]$ are defined as
$\xi_{N^i(p)}^+(q) = \min\{\xi_{\vec{U}}^+ \overrightarrow{(r,s)} : \overrightarrow{(r,s)} \text{ is an edge of } \vec{P}_{(q,p)}^m\}, \psi_{N^i(p)}^+(q) = \min\{\psi_{\vec{U}}^+ \overrightarrow{(r,s)} : \overrightarrow{(r,s)} \text{ is an}$
edge of $\vec{P}_{(q,p)}^m\}$ and $\chi_{N^i(p)}^+(q) = \max\{\chi_{\vec{U}}^+ \overrightarrow{(r,s)} : \overrightarrow{(r,s)} \text{ is an edge of } \vec{P}_{(q,p)}^m\}, \xi_{N^i(p)}^-,\psi_{N^i(p)}^-,\chi_{N^i(p)}^-:$
$X_p^i \to [-1,0]$ are defined as
$\xi_{N^i(p)}^-(q) = \max\{\xi_{\vec{U}}^- \overrightarrow{(r,s)} : \overrightarrow{(r,s)} \text{ is an edge of } \vec{P}_{(q,p)}^m\}, \psi_{N^i(p)}^-(q) = \max\{\psi_{\vec{U}}^- \overrightarrow{(r,s)} : \overrightarrow{(r,s)} \text{ is an}$
edge of $\vec{P}_{(q,p)}^m\}$ and $\chi_{N^i(p)}^-(q) = \min\{\chi_{\vec{U}}^- \overrightarrow{(r,s)} : \overrightarrow{(r,s)} \text{ is an edge of } \vec{P}_{(q,p)}^m\}$.

**Example 44.** Consider the 2-SBPFDG given in Fig 13. Then the 2-SBPFIN of $t$ is $N_2^i(t) = \{t,(0.4,-0.3,0.1,-0.4,0.2,-0.5)\}$.

**Definition 45.** An m-SBPFCG of $\vec{G}$ is denoted by $C_m(\vec{G}) = (V,R,U)$, where $V$ is a vertex set and there is an edge $(p,q)$ in $C_m(\vec{G})$ if and only if $N_m^o(p) \cap N_m^o(q) \neq \varnothing$ in $\vec{G}$. The DMS, DRMS and DNMS of an edge $(p,q)$ are given as

$$\xi_U^+(p,q) = [\xi_R^+(p) \wedge \xi_R^+(q)]h_\xi(N_m^o(p) \cap N_m^o(q))$$
$$\xi_U^-(p,q) = [\xi_R^-(p) \vee \xi_R^-(q)]h_\xi(N_m^o(p) \cap N_m^o(q))$$
$$\psi_U^+(p,q) = [\psi_R^+(p) \wedge \psi_R^+(q)]h_\psi(N_m^o(p) \cap N_m^o(q))$$
$$\psi_U^-(p,q) = [\psi_R^-(p) \vee \psi_R^-(q)]h_\psi(N_m^o(p) \cap N_m^o(q))$$
$$\chi_U^+(p,q) = [\chi_R^+(p) \vee \chi_R^+(q)]h_\chi(N_m^o(p) \cap N_m^o(q))$$
$$\chi_U^-(p,q) = [\chi_R^-(p) \wedge \chi_R^-(q)]h_\chi(N_m^o(p) \cap N_m^o(q))$$

respectively.

In Example 15, we explore 2-SBPFCG.

**Example 46.** Consider a 2-SBPFD $\vec{G}$ given in Fig 14(a) and $C_2(\vec{G})$ shown in Fig 14(b). We have
$N_2^o(r) = \{(t,(0.4,-0.3,0.1,-0.3,0.2,-0.5)),(w,(0.6,-0.3,0.1,-0.4,0.2,-0.6))\}$
$N_2^o(s) = \{(t,(0.4,-0.3,0.1,-0.1,0.3,-0.5)),(w,(0.4,-0.3,0.2,-0.1,0.3,-0.2))\}$
$N_2^o(r) \cap N_2^o(s) = \{(t,(0.4,-0.3,0.1,-0.1,0.3,-0.2)),(w,(0.4,-0.3,0.1,-0.1,0.3,-0.2))\} \neq 0$.
Then there is an edge between $r$ and $s$ in $C_2(\vec{G})$ with DMS, DRMS and DNMS, respectively are $(0.2,0.2,0.01,-0.02,0.06,-0.18)$.

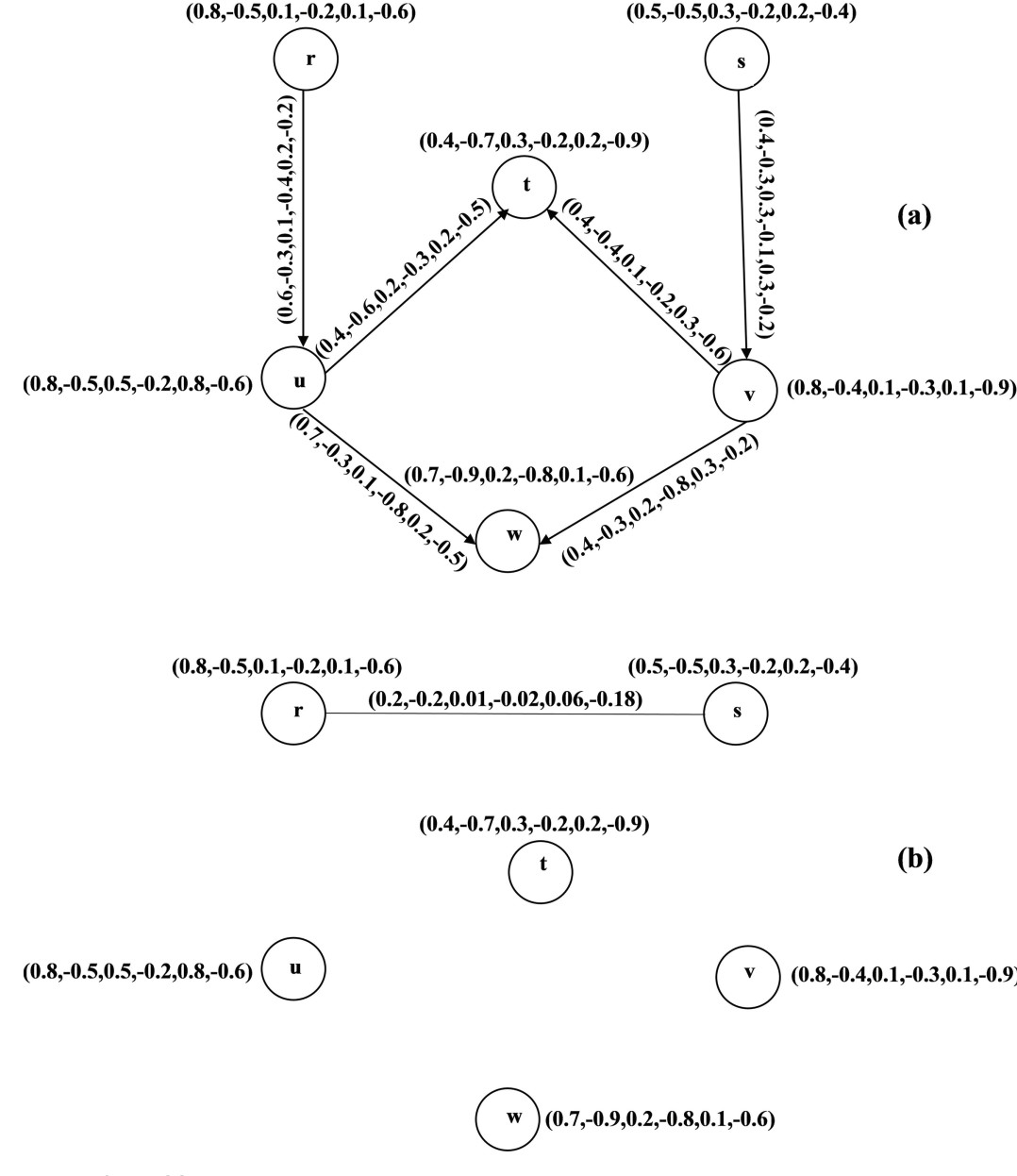

**Fig 14. 2-SBPPFCG.**

## 4 Applications

BPFCGs are the combination of both BPFSs and CGs enabling us to demonstrate complex relationships with uncertainties. In this section, we utilize BPPFCGs to analyze competition in various domains. Overall, we provide four applications corresponding to BPFCGs, BPFp-CGs, BPFk-CGs and BPFm-SCGs, respectively. In each case, we first develop the frameworks and then mathematical computations for each application. These applications serve as strong evidence of the practical utility of BPFCGs.

## 4.1 Analysis of competition among different cell phones companies through BPFCGs

**4.1.1 Framework development.** The concept of BPFCGs has great importance in our real-life scenarios containing uncertainties. In this application, we study competition in cell phone companies through BPFCGs. Consider a BPFDG depicted in Fig 16 which shows a competition among four different mobile phone companies - Apple($P1$), Samsung($P2$), Hawaii($P3$) and Techno($P4$) in the global industry. To remain competitive in the global market, companies struggle to build their products using cutting-edge technologies like data security, applications and unique designs using modern technologies and different processors. In a competitive business era companies attract customers by using their unique designs and modern facilities. Fig 15 illustrates the factors behind the competition among these companies while Table 5 depicts data refer to BPF-environment.

We consider all these companies as nodes (vertices) of the directed graph. Assume the percentage of products of $P1$ maintaining these facilities are categorized as: total facilities offer is 60 percent with 40 percent positive facilities and 20 percent negative facilities, the indeterminacy of facilities is 30 percent with 20 percent positive facilities and 10 percent negative

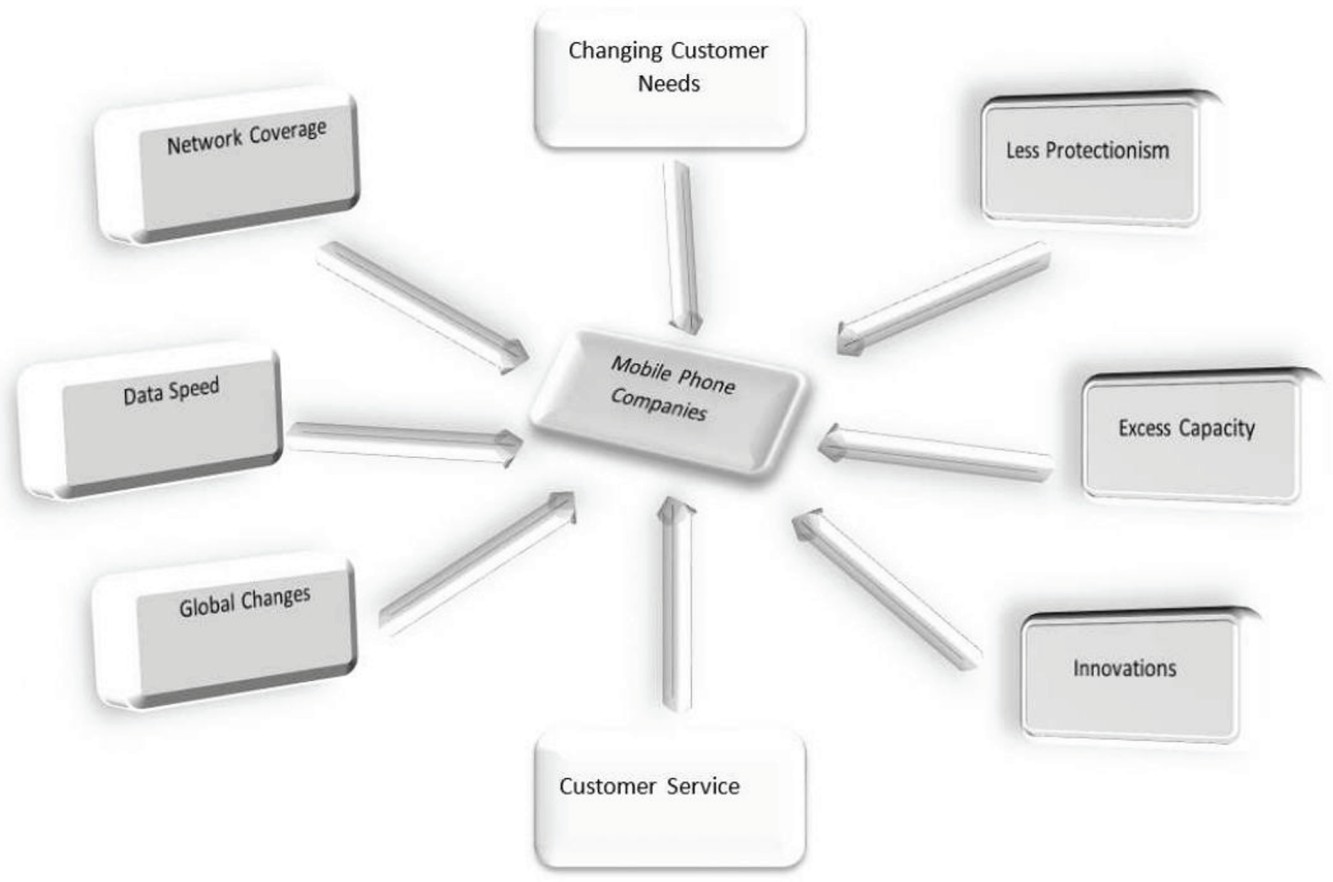

**Fig 15. Factors of competition among companies.**

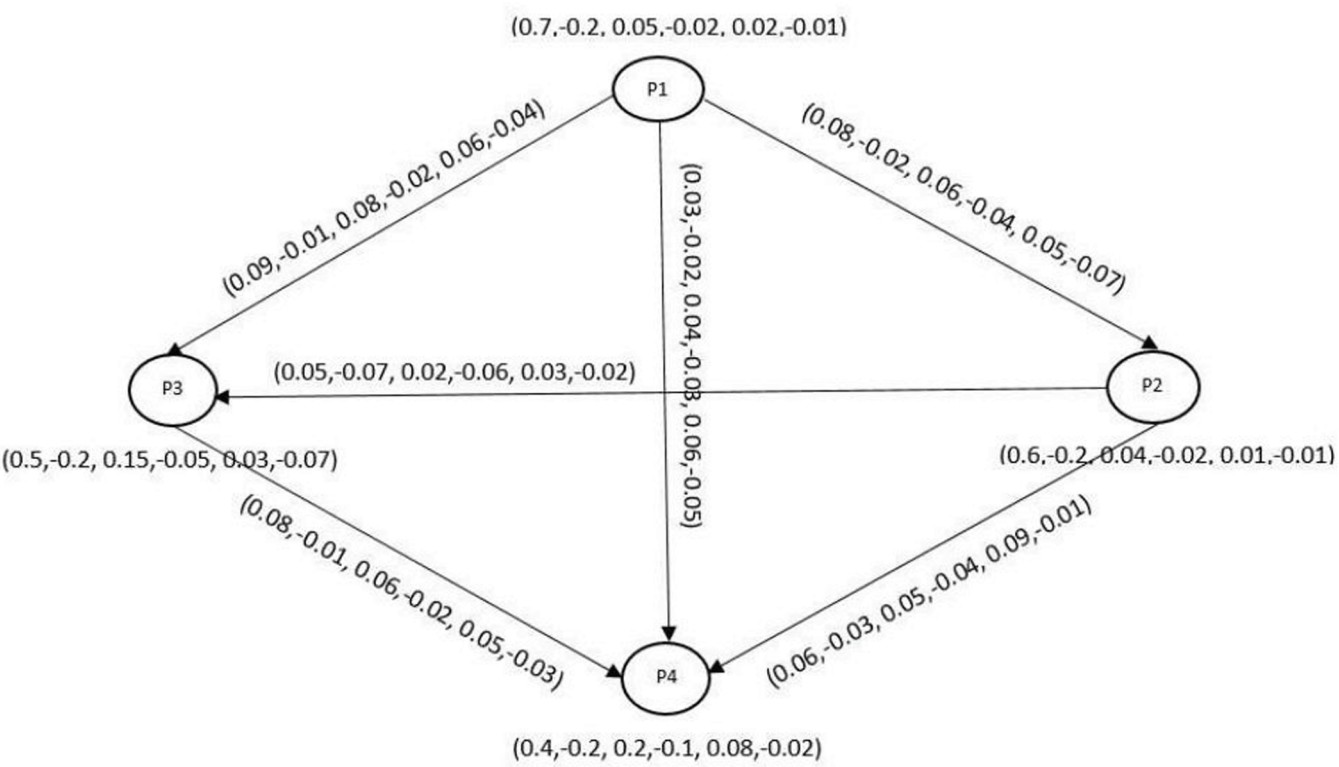

**Fig 16. BPFG of mobile phone companies.**

**Table 5**. **Data set.**

| Vertices | Companies | Per.Of facilities | Per. of indeterminacy of facilities | Per.of less facilities |
|---|---|---|---|---|
| P1 | Apple | (70,−20) | (05,−02) | (02,−01) |
| P2 | Samsung | (60,−20) | (08,−02) | (03,−07) |
| P3 | Hawaii | (50,−20) | (15,−05) | (03,−07) |
| P4 | Techno | (40,−20) | (20,−10) | (08,−02) |

facilities and lacking of facilities is total 10 percent with 8 percent positive facilities and 2 percent negative facilities i.e., DMS, DRMS and DNMS of $P1$ are $\{0.4, -0.2, 0.2, -0.1, 0.08, -0.02\}$. Similarly, we provide the data for all the other nodes (vertices) in Table 6.

**Table 6**. **BPFG Corresponding to each company.**

| Vertices | Companies | BPFG of mobile companies |
|---|---|---|
| P1 | Apple | $\langle(0.7, -0.2), (0.05, -0.02), (0.02, -0.01)\rangle$ |
| P2 | Samsung | $\langle(0.6, -0.2), (0.08, -0.02), (0.03, -0.07)\rangle$ |
| P3 | Hawaii | $\langle(0.5, -0.2), (0.15, -0.05), (0.03, -0.07)\rangle$ |
| P4 | Techno | $\langle(0.4, -0.2), (0.2, -0.1), (0.08, -0.02)\rangle$ |

The directed edge $\overrightarrow{(P1, P2)}$ indicates that the products of $P1$ offer more advanced facilities compared to the products of $P2$. Assume the percentage of products of $P1$ having more facilities is 70 percent with 30 percent positive facilities and 40 percent negative facilities, the indeterminacy of facilities is 10 percent with 6 percent positive facilities and 4 percent negative facilities and less facilities 5 percent with 3 percent positive facilities and 2 percent negative facilities than the products of $P2$ {0.3,-0.4,0.06,-0.04,0.03,-0.02}. Similarly, the data for all the other edges is depicted through Table 7.

## 4.2 An application of bipolar picture fuzzy k-competition graphs in Airline system

**4.2.1 Framework development.** The application of BPFk-CG can be highly beneficial in real-life scenarios particularly within the airline system. Consider a BPFD in Fig 18, which illustrates the competitive dynamics among different airlines in Pakistan such as PIA(A1), Blue Airline(A2), Pakistan Airways(A3) and Shaheen Air(A4). To remain competitive, airlines struggle to improve their facilities by investing in staff training and customer services, enhancing airport lounges and upgrading aircraft with modern amenities and technology. In a competitive business era, airlines attract customers by providing the best customer services and improving their pricing strategies. Fig 17 illustrates the factors behind the competition among different airlines.

In this model, we consider an airline system with four airlines, all of these are represented as nodes of the directed graph. To elaborate, let's take PIA ($A1$) as an example. $A1$ demonstrates different percentages indicating the standard of infrastructure: percentage of maintaining above facilities is 70,–30, the percent for the indeterminacy of facilities is 30,–25 and the percent of inadequate facilities is 90,–75 i.e., DMS, DRMS and DNMS of $A1$ are $\{0.7, -0.3, 0.3, -0.25, 0.9, -0.75\}$. Similarly, for all the other airlines as depicted in Table 8.

The directed edge $\overrightarrow{(A1, A2)}$ in the graph Fig 18 represents that $A1$ offers more advanced facilities as compared to $A2$. For instance, assume a percentage of $A1$ having more facilities is 90,-10, the indeterminacy of facilities is 70,-30 and less facilities 60,-40, i.e., DMS, DRMS and DNMS are $\{0.4, -0.6, 0.8, -0.2, 0.3, -0.7\}$. Similarly, for all the other edges as depicted in Table 9.

**4.2.2 Mathematical computation of framework development.** Here we analyze the competition among these airlines by using BPF4-CG. We have

$N_4^o(A1) \cap N_4^o(A2) = \{(A4, (0.1, -0.1, 0.2, -0.3, 0.6, -0.5))\}$ and $N_4^o(A2) \cap N_4^o(A4) = \{(A3, (0.3, -0.2, 0.5, -0.1, 0.6, -0.7))\}$ (See Table 10. Therefore competition edges of BPF4-CG are $E1 = (A1, A2)$ and $E2 = (A2, A4)$. The DMS, DRMS and DNMS of these edges are $\{0.06, -0.03, 0.23, -0.03, 0.06, -0.12\}$ and $\{0.12, -0.03, 0.46, -0.11, 0.06, -0.12\}$ as depicted in Fig 19. Hence there is a competition between pair of airlines $(A2, A4)$ and $(A1, A4)$.

**Table 7. Relation among companies.**

| Edges | BPF relation among different companies |
|---|---|
| $\overrightarrow{(P1, P2)}$ | $\langle(0.08, -0.02), (0.06, -0.04), (0.05, -0.07)\rangle$ |
| $\overrightarrow{(P1, P3)}$ | $\langle(0.09, -0.01), (0.08, -0.02), (0.06, -0.04)\rangle$ |
| $\overrightarrow{(P1, P4)}$ | $\langle(0.03, -0.02), (0.04, -0.03), (0.06, -0.05)\rangle$ |
| $\overrightarrow{(P2, P3)}$ | $\langle(0.03, -0.02), (0.04, -0.03), (0.06, -0.05)\rangle$ |
| $\overrightarrow{(P2, P4)}$ | $\langle(0.06, -0.03), (0.05, -0.04), (0.09, -0.01)\rangle$ |
| $\overrightarrow{(P3, P4)}$ | $\langle(0.08, -0.01), (0.06, -0.02), (0.05, -0.03)\rangle$ |

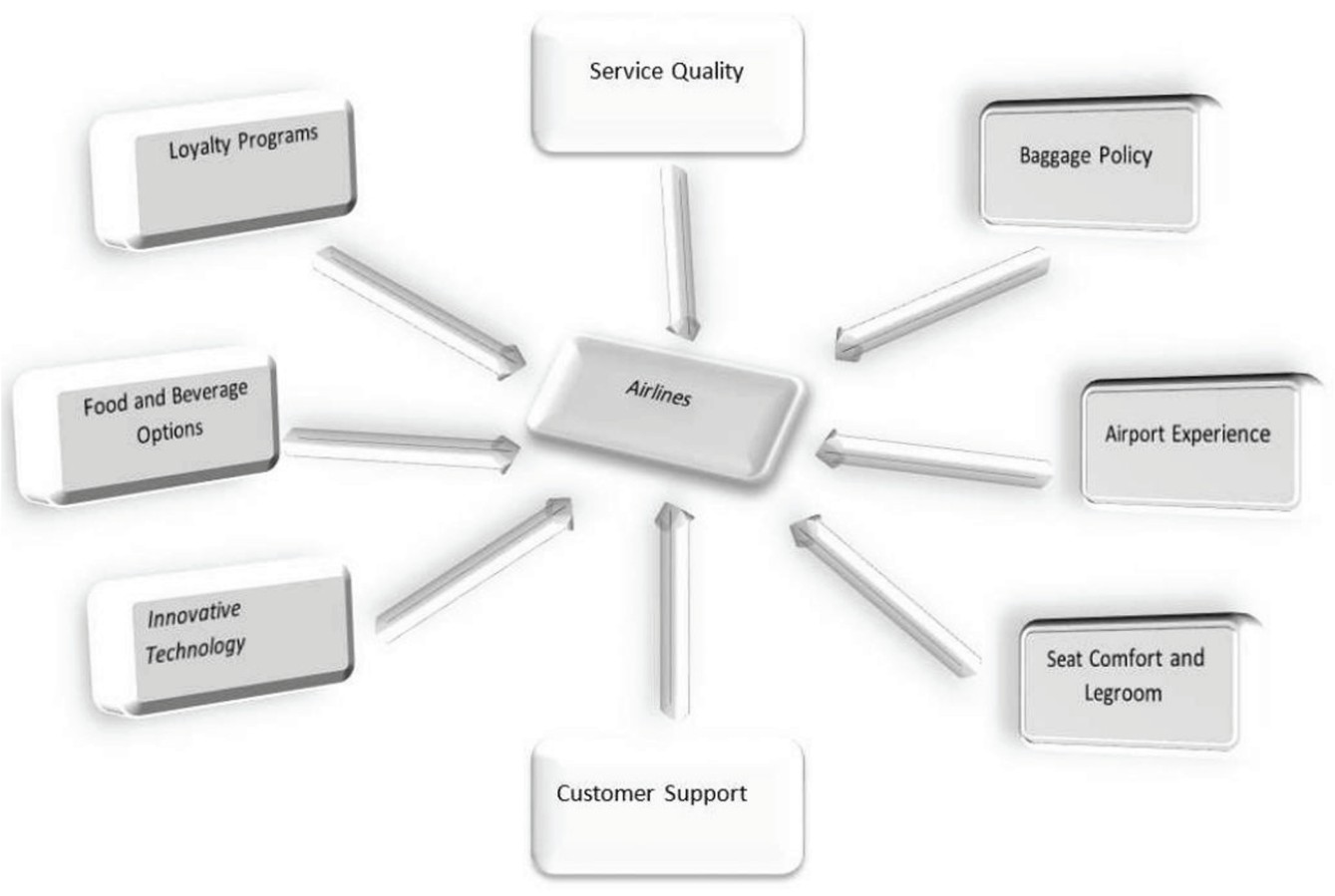

**Fig 17. Factors of the competition among Airlines.**

**Table 8**. BPFG corresponding to each company.

| Vertices | Airlines | BPFG Of Airlines |
|---|---|---|
| A1 | PIA | $\langle(0.7,-0.3),(0.3,-0.25),(0.9,-0.75)\rangle$ |
| A2 | BA | $\langle(0.5,-0.3),(0.7,-0.02),(0.8,-0.5)\rangle$ |
| A3 | PA | $\langle(0.8,-0.2),(0.7,-0.3),(0.4,-0.6)\rangle$ |
| A4 | SA | $\langle(0.6,-0.2),(0.7,-0.3),(0.8,-0.1)\rangle$ |

## 4.3 An application of bipolar picture fuzzy p-competition graph in networking system

**4.3.1 Framework development.** The application of BPFp-CG proves highly beneficial in real-life scenarios. An important application is present in networking systems. Consider a BPFDG depicted in Fig 21, which shows the competition among four different communication networks, i.e., PAN(N1), LAN(N2), MAN(N3) and WAN(N4) based on coverage area, cost, security, connectivity, speed and bandwidth. Fig 20 illustrates the factors behind the competition among different communication networks (Table 11).

In this model, all of these networks are represented as nodes of the directed graph shown in Fig 21. To elaborate, let's take PAN (N1) as an example. Assume the percentage of maintaining above facilities is 80, −20, indeterminacy of facilities is 70, −30 and less facilities

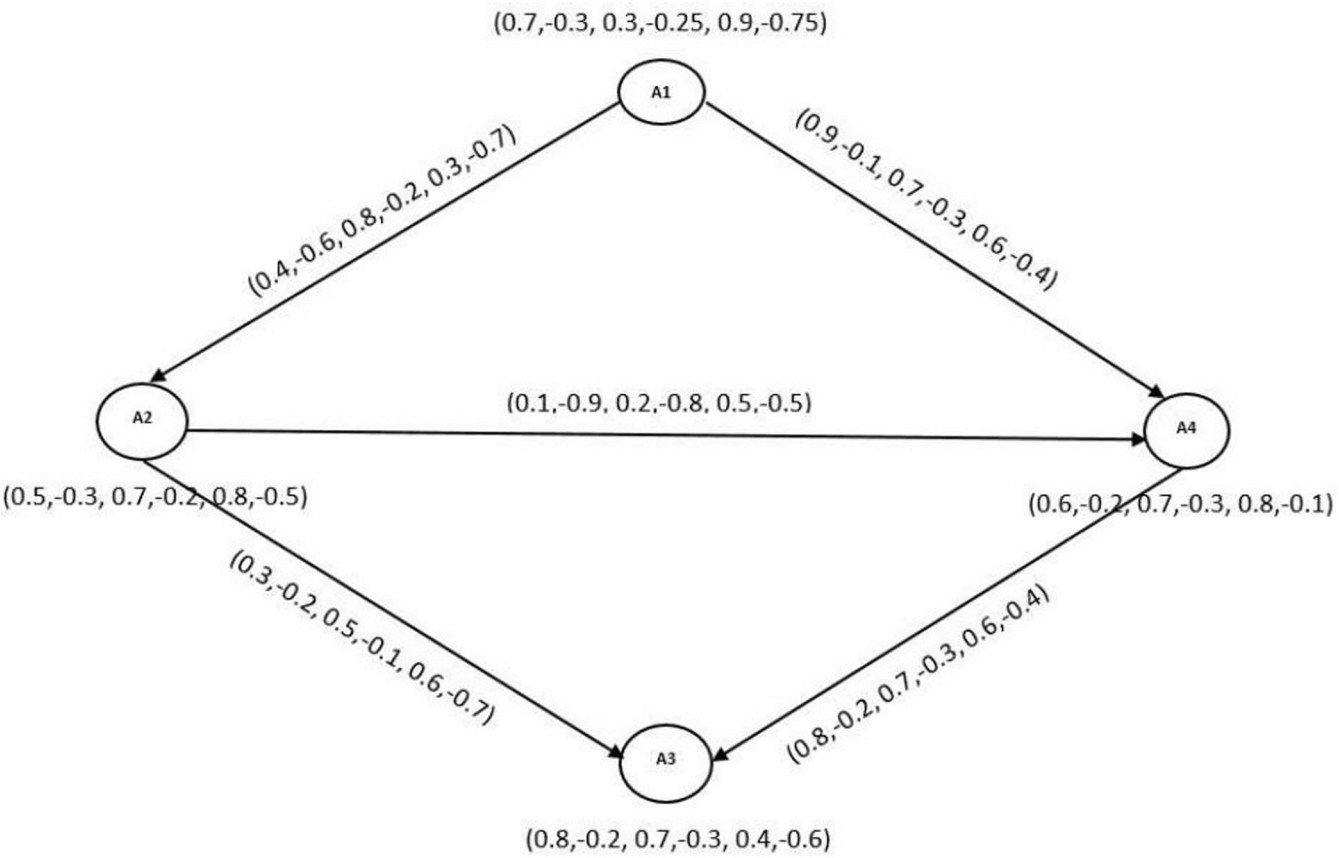

**Fig 18. BPFDG of Pakistan's Airline System.**

**Table 9**. BPFG.

| Edges | BPFG |
|---|---|
| $\overrightarrow{(A1, A2)}$ | $\langle (0.4, -0.6), (0.8, -0.2), (0.3, -0.7) \rangle$ |
| $\overrightarrow{(A2, A3)}$ | $\langle (0.3, -0.2), (0.5, -0.1), (0.6, -0.7) \rangle$ |
| $\overrightarrow{(A4, A3)}$ | $\langle (0.8, -0.2), (0.7, -0.3), (0.6, -0.4) \rangle$ |
| $\overrightarrow{(A2, A4)}$ | $\langle (0.1, -0.9), (0.2, -0.8), (0.5, -0.5) \rangle$ |
| $\overrightarrow{(A1, A4)}$ | $\langle (0.9, -0.1), (0.7, -0.3), (0.6, -0.4) \rangle$ |

**Table 10**. Picture fuzzy out-neighborhood of Fig 18.

| $v \in V$ | $N^o(v)$ |
|---|---|
| $N_4^o(A1)$ | $\{(A2,(0.4,-0.6,0.8,-0.2,0.3,-0.7)),(A4,(0.9,-0.1,0.7,-0.3,0.6,-0.4))\}$ |
| $N_4^o(A2)$ | $\{(A3,(0.3,-0.2,0.5,-0.1,0.6,-0.7)),(A4,(0.1,-0.9,0.2,-0.8,0.5,-0.5))\}$ |
| $N_4^o(A3)$ | $\phi$ |
| $N_4^o(A4)$ | $\{A3,(0.8,-0.2,0.7,-0.3,0.6,-0.4)\}$ |

$40, -60$. This breakdown, denoted as DMS, DRMS, and DNMS $\{0.8, -0.2, 0.7, -0.3, 0.4, -0.6\}$ respectively, informs decisions about the infrastructure of other networks in a similar way as depicted in Table 12. Initially, people prefer PAN (N1), then they shift to another network according to their need and requirements. The directed edge $\overrightarrow{(N1, N2)}$ indicates

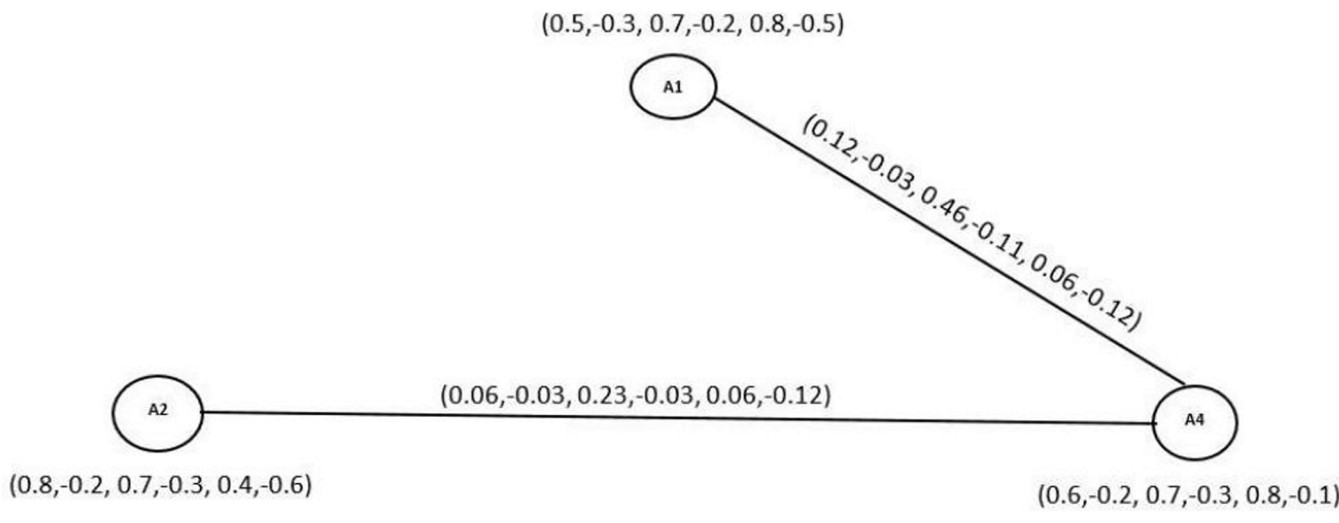

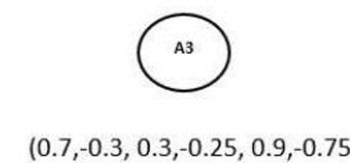

**Fig 19. BPF4-CG.**

that $N1$ offer more advanced facilities as compared to $N2$, i.e., DMS, DRMS and DNMS $\{0.4, -0.6, 0.8, -0.2, 0.3, -0.7\}$. Similarly, for all other edges as depicted in Table 13.

**4.3.2 Mathematical computation of framework development.** We have $N^o(LAN) \cap N^o$ $(MAN) = \{(PAN, (0.4, -0.6, 0.4, -0.2, 0.3, -0.7)), (WAN, (0.8, -0.2, 0.4, -0.3, 0.6, -0.4))\}$ $N^o(LAN) \cap N^o(PAN) = \{(WAN, (0.3, -0.2, 0.4, -0.1, 0.6, -0.7))\}$ and $N^o(PAN) \cap N^o(MAN) = \{(WAN, (0.3, -0.2, 0.5, -0.1, 0.6, -0.7))\}$ (See Table 14). Subsequently, competition edges of BPFp-CG are $E1 = (P1, P2)$, $E2 = (P1, P3)$ and $E3 = (P2, P3)$. The DMS, DRMS and DNMS of the paths are $\{0.2, -0.12, 0.14, -0.04, 0.24, -0.03\}$, $\{0.75, -0.03, 0.14, -0.04, 0.024, -0.18\}$ and $\{0.09, -0.03, 0.18, -0.08, 0.24, -0.18\}$ as shown in Fig 22. Hence there is a competition among LAN and MAN, LAN and PAN, PAN and MAN in the BPFp-CG.

## 4.4 An Application of Bipolar picture fuzzy m-step competition graph in ecosystem

**4.4.1 Framework development.** The m-SBPFCG is also applicable in real life scenarios where competition is indirect. The m-SBPFCG can also be useful within the ecosystem. In this model, let us assume an ecosystem with seven species, all of these are represented as nodes of the directed graph presented in Fig 23. In this ecosystem, feeding interactions of species define relationships between them. Hawk prey on snake and grasshopper, snake eats

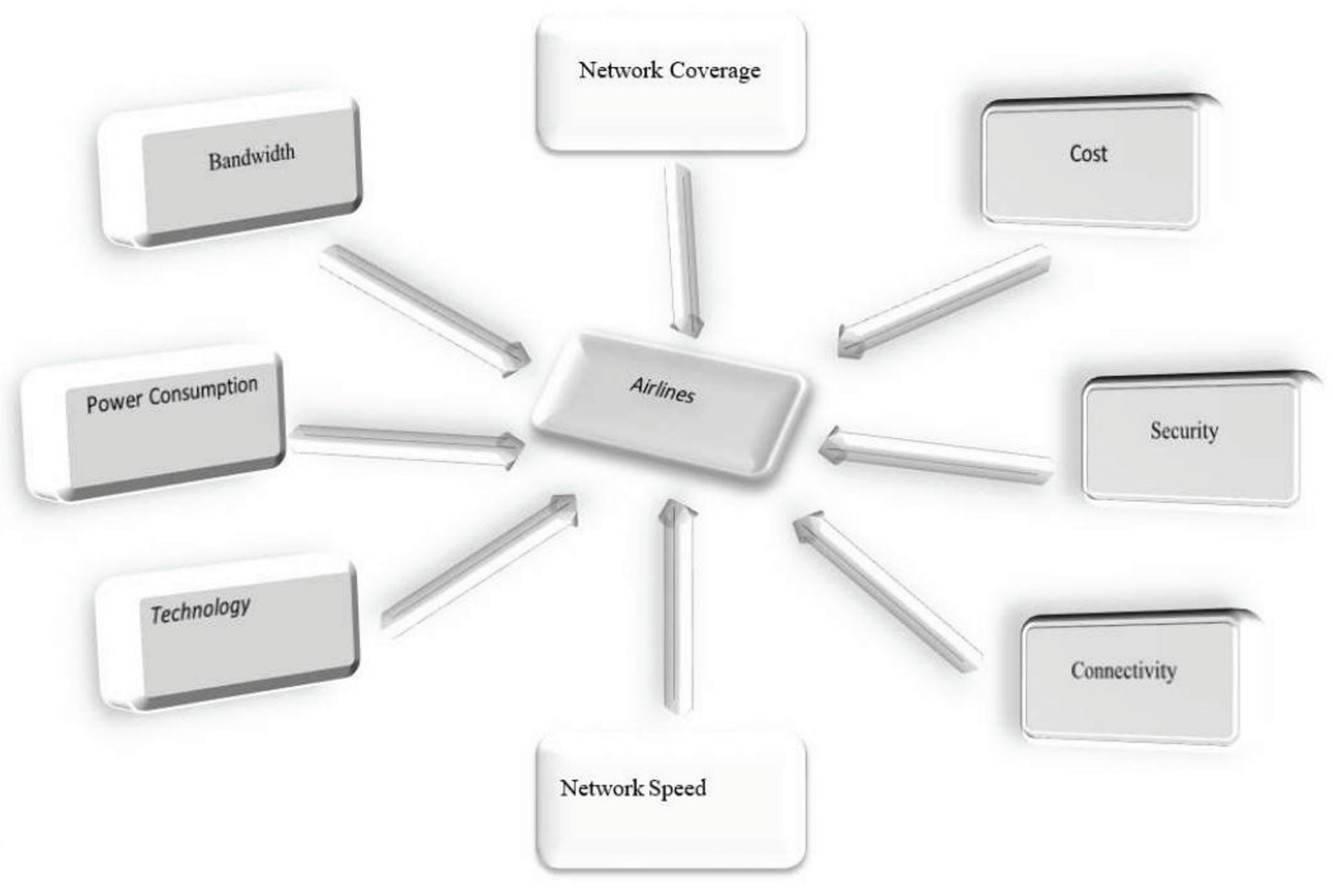

**Fig 20. Factors of the competition among communication networks.**

frogs and grasshopper, owl prey on birds and mice, birds feed on grains and grasshopper, mice feed on grains and grasshopper prey on grain and grass. Consider the probability of presence of the hawk in the environment is 40 and -30 percent, the probability of indeterminacy in existence is 5 and -19 percent, and probability of non-existence is 30 and -50 percent. DMS, DRMS and DNMS of the hawk can be expressed as $\{0.4, -0.3, 0.05, -0.19, 0.3, -0.5\}$. DMS, DRMS, and DNMS for other species in the ecosystem can be defined in the similar way as shown in Table 15.

 **4.4.2 Mathematical computation of framework development.** We have $N_2^o(hawk) \cap N_2^o$ $(owl) = \{(grain, (0.35, 0.05, 0.25))\}$ (See Table 16). Subsequently, there is a path among hawk and owl in the 2-SBPPFCG. This shoes there is a 2-step connection in the BPPFCG. The DMS, DRMS and DNMS of the path is $\{0.08, -0.02, 0.05, -0.01, 0.63, -0.45\}$. Subsequently, based on take care of in environment, there is a 2-step competition among hawk and owl as shown in Fig 24.

## 5 Comparative study and prevalence of our presented model

The idea of CGs is of great importance and has a quite large number of applications in different fields. In real-life scenarios, competition among individuals or organizations containing uncertainties is being solved by using CGs in FGs. The idea of FGs was first introduced by Rosenfeld and has been generalized in different ways. FGs use DMS of vertices and edges in

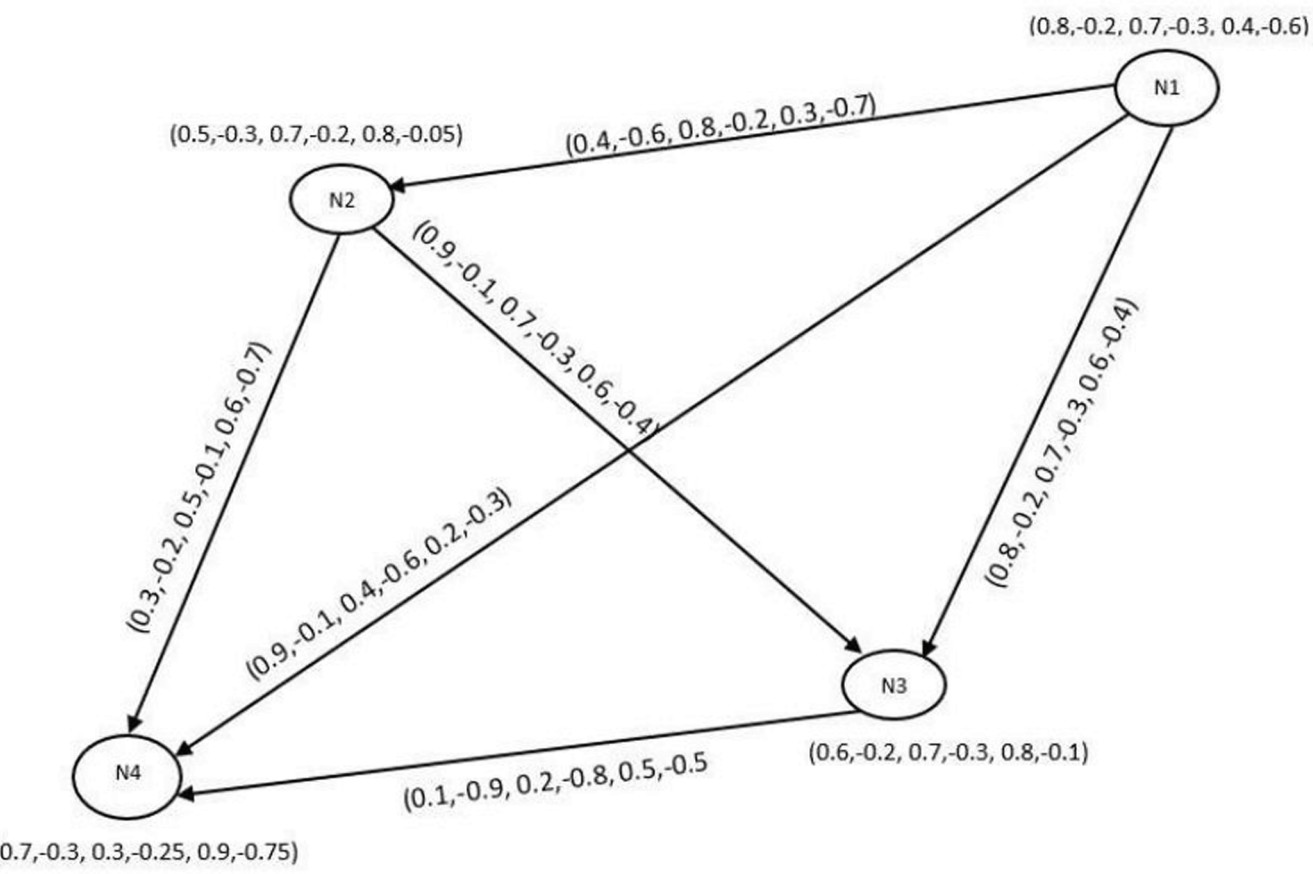

**Fig 21. Networking system.**

**Table 11. Data set.**

| Vertices | Networks | Per.Of facilities | Per. of indeterminacy of facilities | Per.of less facilities |
|---|---|---|---|---|
| N1 | PAN | $(80, -20)$ | $(70, -30)$ | $(40, -60)$ |
| N2 | LAN | $(50, -30)$ | $(70, -20)$ | $(80, -05)$ |
| N3 | MAN | $(60, -20)$ | $(70, -30)$ | $(80, -10)$ |
| N4 | WAN | $(70, -30)$ | $(30, -25)$ | $(90, -75)$ |

**Table 12. BPFG.**

| Vertices | Networks | BPFG corresponding to each network |
|---|---|---|
| N1 | PAN | $\langle (0.8, -0.2)(0.7, -0.3)(0.4, -0.6) \rangle$ |
| N2 | LAN | $\langle (0.5, -0.3)(0.7, -0.2)(0.8, -0.05) \rangle$ |
| N3 | MAN | $\langle (0.6, -0.2)(0.7, -0.3)(0.8, -0.1) \rangle$ |
| N4 | WAN | $\langle (0.7, -0.3)(0.3, -0.25)(0.9, -0.75) \rangle$ |

specific ways. FGs were used to solve many real-world problems. As like as FSs, FGs were generalized in different ways to deal with complex issues with uncertainties. In our daily lives, human judgments typically involve both positive and negative aspects of events and hence

**Table 13**. BPFG.

| Edges | BPFG corresponding to each network |
|---|---|
| $\overrightarrow{(N1,N2)}$ | $\langle(0.4,-0.6),(0.8,-0.2),(0.3,-0.7)\rangle$ |
| $\overrightarrow{(N1,N3)}$ | $\langle(0.8,-0.2),(0.7,-0.3),(0.6,-0.4)\rangle$ |
| $\overrightarrow{(N1,N4)}$ | $\langle(0.9,-0.1),(0.4,-0.6),(0.2,-0.3)\rangle$ |
| $\overrightarrow{(N2,N3)}$ | $\langle(0.9,-0.1),(0.7,-0.3),(0.6,-0.4)\rangle$ |
| $\overrightarrow{(N2,N4)}$ | $\langle(0.3,-0.2),(0.5,-0.1),(0.6,-0.7)\rangle$ |
| $\overrightarrow{(N3,N4)}$ | $\langle(0.1,-0.9),(0.2,-0.8),(0.5,-0.5)\rangle$ |

**Table 14**. Picture fuzzy out-neighborhood of Fig 21.

| $v \in V$ | $N^o(v)$ |
|---|---|
| $N1$ | {(N2,(0.4, −0.6, 0.8, −0.2, 0.3, −0.7)),(N3,(0.8, −0.2, 0.7, −0.3, 0.6, −0.4), (N4,(0.9, −0.1, 0.4, −0.6, 0.2, −0.3))} |
| $N2$ | {(N3,(0.9, −0.1, 0.7, −0.3, 0.6, −0.4)),(N4,(0.3, −0.2, 0.5, −0.1, 0.6, −0.7))} |
| $N3$ | {N4,(0.1, −0.9, 0.2, −0.8, 0.5, −0.5)} |
| $N4$ | $\phi$ |

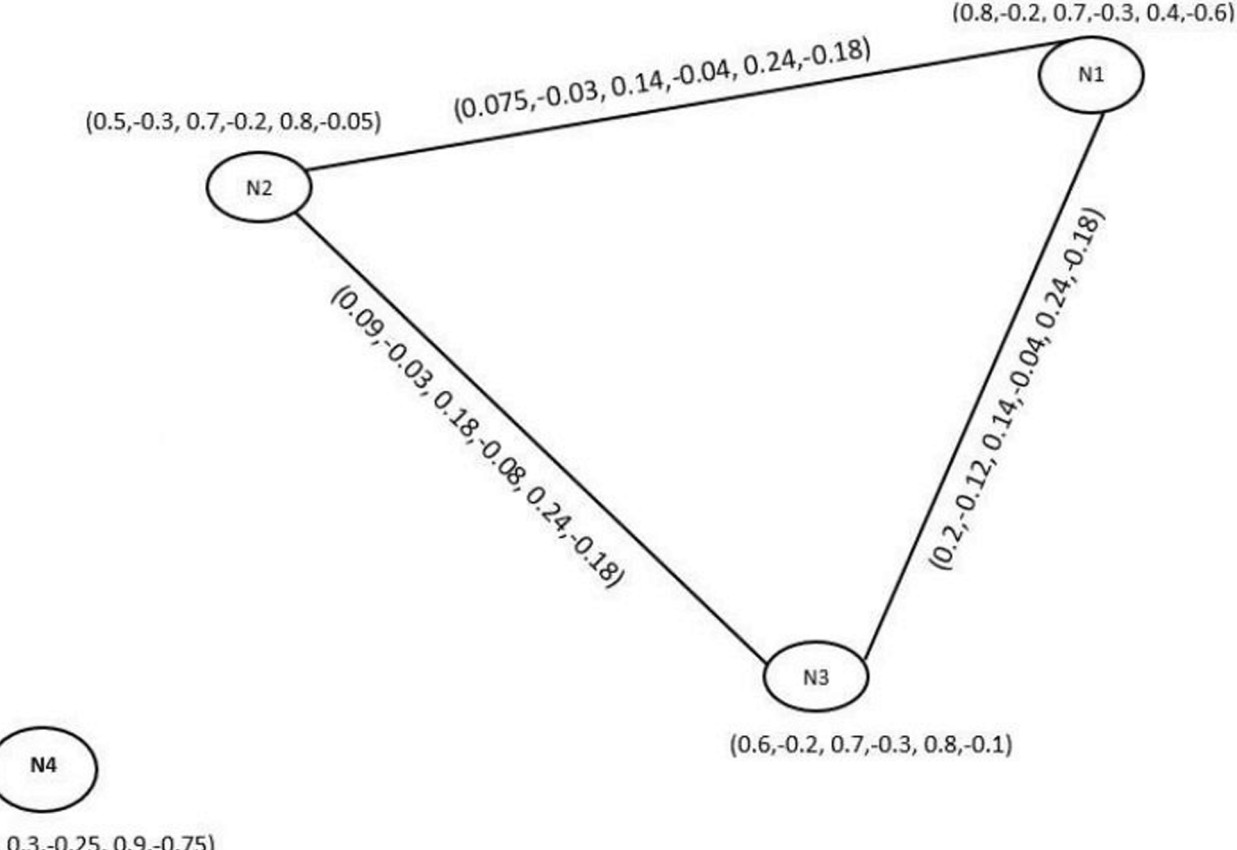

**Fig 22. Competition graph.**

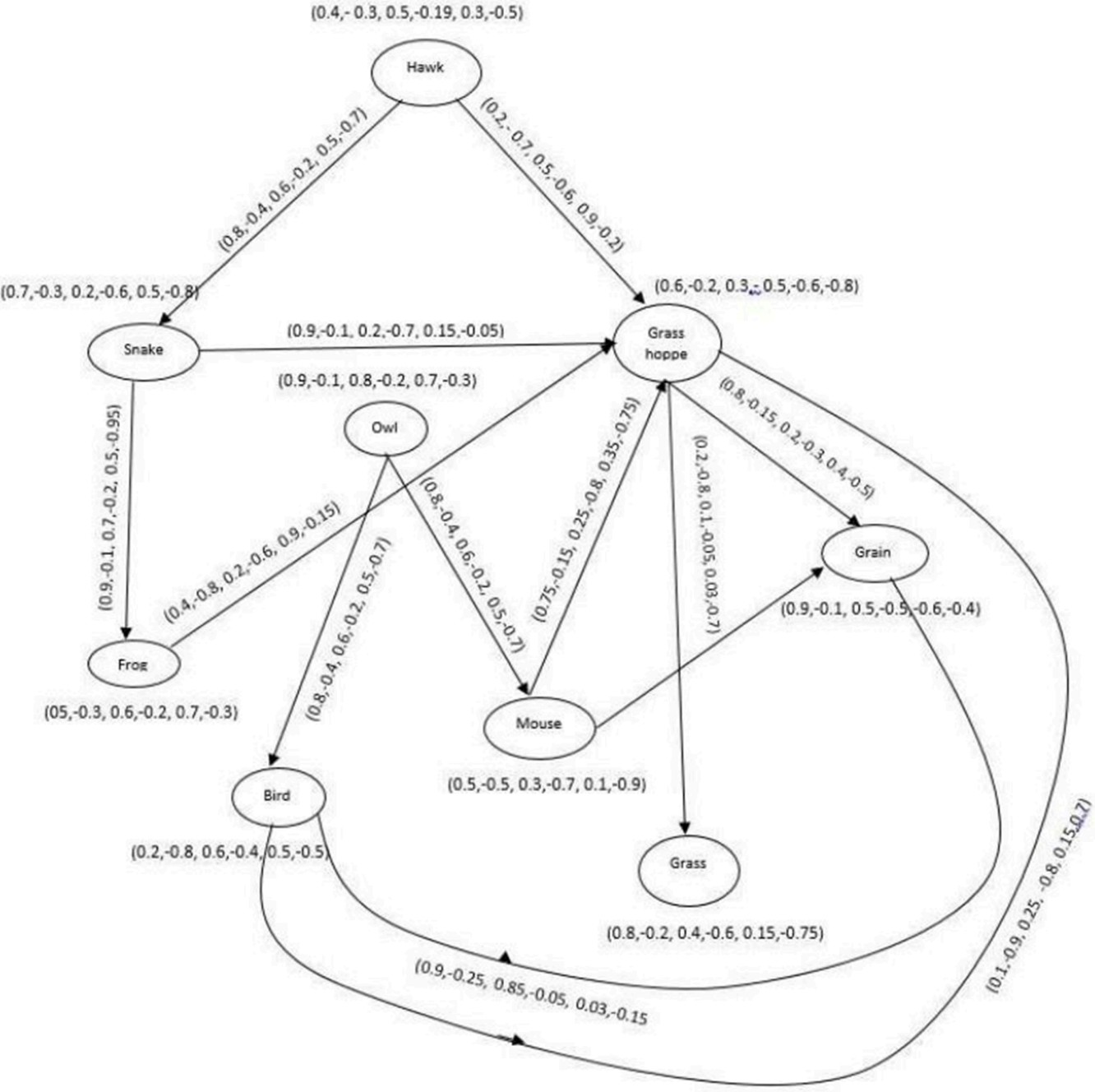

**Fig 23. BPFDG of ecosystem.**

FGs were unable to deal with such scenarios. To overcome these limitations of FSs and FGs, BFSs and BFGs were introduced. BFGs was the generalization of FGs and it uses BFSs to express values for vertices and edges. However, with respect to time uncertainties in problems are getting more complex. There are many aspects of human life where some things have both membership and non-membership values simultaneously. However, FSs and FGs only discuss about MS value of a situation or problem. Hence the concept of IFGs was introduced

**Table 15**. BPFG.

| Species | DMS | DRMS | DNMS |
|---------|-----|------|------|
| Hwak | (0.4, −0.3) | (0.5, −0.19) | (0.3, −0.5) |
| Snake | (0.7, −0.3) | (0.2, −0.6) | (0.5, −0.8) |
| Grasshopper | (0.6, −0.2) | (0.3, −0.5) | (0.6, −0.8) |
| Owl | (0.9, −0.1) | (0.8, −0.2) | (0.7, −0.3) |
| Frog | (0.5, −0.3) | (0.6, −0.4) | (0.5, −0.5) |
| Bird | (0.2, −0.8) | (0.6, −0.4) | (0.5, −0.5) |
| Mouse | (0.5, −0.5) | (0.3, −0.7) | (0.1, −0.9) |

**Table 16**. 2-step Picture fuzzy out-neighbourhood of Fig 23.

| $v \in V$ | $N_2^o(v)$ |
|-----------|------------|
| Hawk | {(Frog, (0.8, −0.1, 0.6, −0.2, 0.5, −0.7)), (grasshopper, (0.8, −0.1, 0.2, −0.2, 0.5, −0.05)), (grass,(0.2, −0.7, 0.1, −0.05, 0.03, −0.2)), (grain,(0.2, −0.15, 0.2, −0.3, 0.4, −0.2))} |
| Snake | {(grasshopper, (0.4, −0.1, 0.2, −0.2, 0.9, −0.95)), (grass, (0.2, −0.1, 0.1, −0.05, 0.15, −0.7)), (grain, (0.8, −0.1, 0.2, −0.3, 0.15, −0.5))} |
| Grasshopper | $\phi$ |
| Owl | {(grasshopper, (0.1, −0.75, 0.05, −0.8, 0.15, −0.5)), (grasshopper, (0.7, −0.15, 0.25, −0.3, 0.35, −0.75)), (grain, (0.9, −0.25, 0.05, −0.05, 0.15, −0.75)), (grain,(0.7, −0.9, 0.1, −0.2, 0.5, −0.6))} |
| Frog | {(grass,(0.2, −0.8, 0.1, −0.05, 0.9, −0.15)), (grain,(0.4, −0.8, 0.2, −0.3, , 0.9, −0.15))} |
| Mouse | {(grass,(0.2, −0.8, 0.1, −0.05, 0.35, −0.75)), (grain,(0.75, −0.15, 0.2, −0.3, 0.4, −0.75))} |
| Bird | {(grass,(0.1, −0.8, 0.1, −0.05, 0.15, −0.7)), (grain,(0.1, −0.9, 0.2, −0.3, 0.15, −0.7))} |
| Grass | $\phi$ |
| Grain | $\phi$ |

using IFSs to overcome these complex problems. The concept of CGs was also introduced for IFGs to deal with the competition different fields. To deal with double-sided opinions in the field of IFSs and IFGs, the concept of BIFGs was introduced. BIFGs are a more advanced and bigger structure than FGs, BFGs and IFGs. The concept of CGs for BIFGs is also introduced to deal with competition in various real-life scenarios having uncertainties and bipolarity. There are many aspects of human life where the concept of "neutrality" arises, but IFSs and IFGs do not accommodate this concept. To overcome these limitations of IFSs and IFGs, PFSs and PFGs were introduced. PFSs are a generalization of FSs and IFSs and having DMS, DRMS and DNMS. The model of PFGs was introduced by using picture fuzzy relations. Since BFGs, BIFGs and BPFGs were introduced to deal with bipolarity in different fields. But BPFGs were more generalized and summed up than BFGs and BIFGs as depicted in Table 17.

Moreover, the notions of competition graphs were introduced in the domains of various types of FGs in literature, as depicted in Table 18.

However, we have noticed that there is a gap in the existing competition analysis i.e., the concept of CGs has not yet been introduced for BPFGs. To fill this gap, we apply the concept of CGs for BPFGs and develop its further classification i.e., BPFk-CGs, BPFp-CGs and m-SBPFCGs. Since BPFGs deal the uncertain real-life scenarios with great accuracy. In this context, we have presented few applications in daily life scenarios with demonstrative examples by using these terms that prove the importance of our presented concepts. Moreover, our presented idea of CGs for BPFGs is comparatively more beneficial than BFGs, BIFGs and PFGs. The reason is that it deals with membership, neutrality and non-membership with bipolarity.

For the sake of convincing, if we deal the above presented application of cell phone companies through BFCGs, then we can't properly describe the competition between companies presented in the analysis because it only focuses on upper and lower MS values. Hence we can only deal with the percentage of facilities provided by the companies and their positive

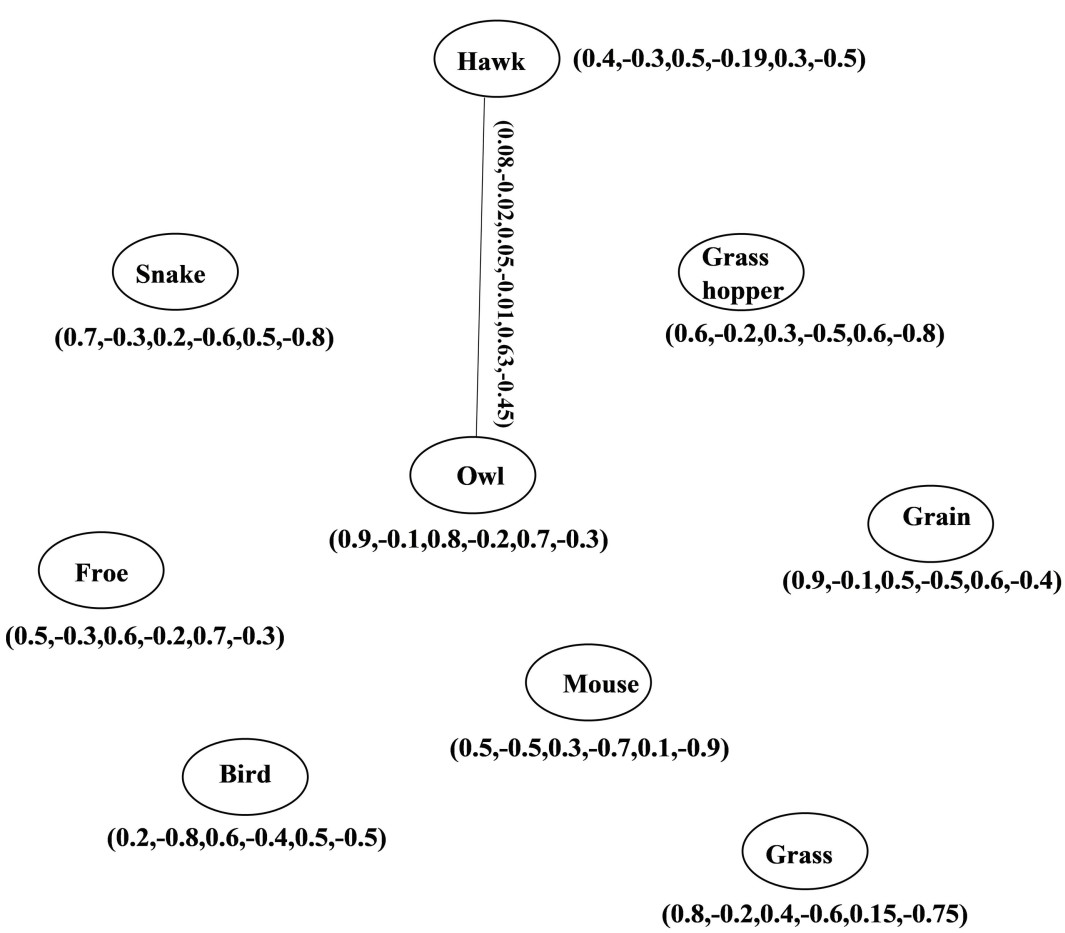

**Fig 24. 2-SBPFCG of ecosystem.**

**Table 17**. Generalization of fuzzy graphs.

| Authors | Reference | Notion introduced |
|---|---|---|
| A.Rosenfeld | [36] | Fuzzy graphs |
| M.Akram | [19] | Bipolar fuzzy graphs |
| Paravathi and Karunambigai | [42] | Intuitionistic fuzzy graph |
| Ezhilmaran and Sankar | [34] | Bipolar intuitionistic fuzzy graph |
| Zuo et al. | [47] | Picture fuzzy graph |
| Khan et al. | [58] | Bipolar picture fuzzy graph |

**Table 18**. Competition graph for different notations.

| Authors | Reference | Notion introduced |
|---|---|---|
| Cohen | [10] | Fuzzy Competition graphs |
| Al-shehri and Akram | [19] | Bipolar fuzzy Competition graphs |
| Sahoo and Pal | [22] | Intuitionistic fuzzy Competition graph |
| Deva and Felix | [23] | Bipolar intuitionistic fuzzy Competition graph |
| Das et al. | [24] | Picture fuzzy Competition graph |

and negative effects but it can't provide a comprehensive analysis of the companies' strengths, weaknesses and competitive strategies, as they ignore the indeterminacy and less facilities. Thus, the analysis becomes limited when indeterminacy and absence of facilities are ignored. Similarly, if we deal with our above application of mobile phone companies through BIFCGs, again we can't properly describe the competition between companies presented in the analysis because it focuses only on upper and lower MS values and non-membership values. It also only deals with the presence and absence of facilities provided by the companies and their positive and negative effects but it can't provide a comprehensive analysis as they also ignore the concept of neutrality. In this way, we can only analyze the percentage of the facilities as well as less facilities of the products provided by the companies. Consequently, BPFCGs is the only model to cover all three aspects (MS,NMS and neutral) of the competition with polarity.

## 5.1 Prevalence of our presented model for mobile companies through BPFCGs

In the analysis of competition graph in different cell phone companies through BPFCGs, the presence of facilities ranges from 90 percent representing the lower and upper MS values to 3 percent less facilities representing the lower and upper non-membership values. Along with MS values reflecting the facilities of products, and non-membership values representing less facilities of products, further considerations may be required i.e., PFCGs which include a "neutrality degree" that shows indeterminacy of facilities. This may help companies to build more advanced products to compete with other companies and attract customers. For example, the presence of facilities ranges from 90 percent representing MS values and 3 percent less facilities representing non-membership values and also 7 percent neutrality or indeterminacy. Assessing the impact of indeterminate advanced facilities, such as foldable displays and augmented reality capabilities, is crucial in the competitive landscape of mobile phone companies. Considering these uncertain technologies offers a more comprehensive understanding of their effectiveness and customer engagement, identifies potential obstacles to adoption, and uncovers areas for improvement in developing, marketing, and delivering these innovative features to gain customers' attention and stay ahead in the market. PFCGs have limitations as it does not deal with the concept of bipolarity. So we can't discuss the positive and negative effects of facilities, which play a vital role in competition among companies and in attracting customers. To address this, a more comprehensive structure is needed to represent the overall criteria of competition among companies. Hence, the introduced notation of bipolar picture fuzzy competition graphs in PFCGs provides a solution, enabling a more precise evaluation of factors of competition among companies. For example, the presence of facilities is 90 percent, indeterminate facilities are 7 percent and 3 percent less facilities. As we have illustrated in the previous example of competition among mobile phone companies, the concept of bipolarity is very useful. Hence our model is more generalized as it covers all aspects regarding competition between companies. Some characteristics comparison of BPFGs with existing models are depicted in Table 19. Moreover, the attribute comparison of competitions between our proposed model with the BFCGs and BIFCGs is also presented in Tables 20 and 21.

## 6 Real-world application: Modeling product competition using BPFCGs

**Data set description and relevance.** We utilize a simplified and representative subset of the data-set from SNAP [3] which captures relationships between products that are frequently

**Table 19.** Characteristic comparison of BPFGs with BFGs and BIFGs.

| Characteristics | Bipolar Fuzzy Graphs (BFGs) | Bipolar Intuitionistic Fuzzy Graphs (BIFGs) | Bipolar Picture Fuzzy Graphs (BPFGs) |
|---|---|---|---|
| Representation | Uses membership values for the presence of advanced facilities with positive and negative values | Uses membership values for the presence of advanced facilities and non-membership values for less facilities with positive and negative values | Uses membership values for the presence of advanced facilities and neutral values for indeterminacy and non-membership values for less facilities with positive and negative values |
| Membership Functions | Uses two membership functions one for positive membership and one for negative membership | Uses four membership functions one for positive membership, one for negative membership, one for positive non-membership and one for negative non-membership | Uses six membership functions one for positive membership, one for negative membership, one for positive neutral membership, one for negative neutral membership, one for positive non-membership and one for negative non-membership |
| Proficiency in dealing with real world problems | Less proficient in handling complex, world problems with uncertainty. | More proficient in handling complex real-world problem | Most proficient in handling complex, real-world problems with multiple aspects and uncertainty. |
| flexibility representation of competition among companies. | Least flexible,as they only allow for simple positive and negative relationship | more flexible and provide a balance between flexibility and simplicity as they only allow for positive and negative membership and non-membership | BPFGs offer the highest flexibility, making them suitable for complex, dynamic competitive environments |

bought together. Each product is represented as a node, and a co-purchase connection is represented as an edge. This data set is well-suited for modeling competitive and complementary interactions among products, especially when viewed through the lens of bipolar picture fuzzy logic, which allows the integration of uncertainty and contradiction in customer preference data. Selected products from the Amazon Co-Purchasing Network data set used for constructing the BPFCG. These items are highly co-purchased and serve as a representative subset for modeling competition under uncertainty, shown in Table 22.

## 6.1 Methodology: BPFCG construction steps

1. **Data Preprocessing:** A representative set of five products was selected: `P1`, `P2`, `P3`, `P4`, and `P5`, with co-purchase links among them forming the edge set.

2. **Fuzzy Membership Assignment:** Each product was assigned three values based on synthetic criteria reflecting real-world analogs. In the construction of the BPFCG as depicted in the Table 23, where each edge connecting two products is assigned a six-tuple of fuzzy values, corresponding to the degrees of membership, non-membership, and neutrality, each measured in both positive and negative perspectives. These values are interpreted as follows:

   - $\xi^+$: The positive membership degree, indicating the strength of competition from the perspective of product preference overlap or substitutability.

   - $\xi^-$: The negative membership degree, representing inverse competition or dissimilar appeal between the two products.

**Table 20.** Comparison of BPFCGs with existing models.

| Model | Key Features | Advantages | Limitations |
|---|---|---|---|
| Bipolar Fuzzy Competition Graphs(BFCGs) | Both good and negative associations are modeled, Fuzzy memberships are used to manage ambiguity and uncertainty, Adapting dynamically to shifting market conditions | Captures both positive and negative influences makes interpretation easy, can handle uncertainty and incomplete data, makes computing relationships or applying graph algorithms is easy, use minimum memory consumption, allow for a nuanced analysis of competition | It ignores neutral and non-membership values,not effective in complex and multi-dimensional relationships. |
| Bipolar Intuitionistic Fuzzy Competition Graphs(BIFCGs) | Both good and negative associations are modeled, Fuzzy memberships are used to manage ambiguity and uncertainty, Adapting dynamically to shifting market conditions | Captures both positive and negative influences, can handle uncertainty interpretation and incomplete data, deals with membership as well as non-membership values simultaneously | Makes interpretation a bit difficult, makes computing relationships or graph algorithms a bit difficult, it ignores neutral membership |
| Bipolar Picture Fuzzy Competition Graphs(BPFCGs) | Both good and negative associations are modeled, Fuzzy memberships are used to manage ambiguity and uncertainty, Adapting dynamically to shifting market conditions, Can model complex, multi-dimensional relationships | Captures both positive and negative influences, can handle uncertainty and incomplete data, effective in complex, uncertain, and ambiguous environments, deals with membership, neutral as well as non-membership simultaneously | Makes interpretation complex, computing relationships or applying graph algorithms is complex, use high memory values consumption |

- $\psi^+$: The positive non-membership degree, expressing the level of coexistence or complementary nature between products (e.g., often bought together).
- $\psi^-$: The negative non-membership degree, capturing opposing traits or lack of compatibility in consumer perception.
- $\chi^+$: The positive neutrality degree, reflecting uncertainty or partial indifference in consumer choice due to weak or indirect associations.
- $\chi^-$: The negative neutrality degree, indicating contradicting ambiguity, such as polarized preferences or inconsistent patterns of co-purchase.
- **Graph Modeling:** A BPFCG is created, where nodes are annotated with the above values and the competition strength will be computed using weighted aggregation or score function.

These six values provide a comprehensive structure for modeling nuanced, uncertain, and context-dependent competition dynamics among products in the Amazon co-purchasing network. The assignment of values is informed by product meta-data, genre similarity, co-purchase frequency, and substitutability indicators.

## 6.2 Competition strength analysis: Computation of competition strength in BPFCGs

In a BPFCG, each edge connecting two products $P_i$ and $P_j$ is described by a six-tuple of fuzzy values:

**Table 21.** **Comparison of BPFCGs with a broader range of fuzzy and competition graph models.**

| Model | Types Of Membership | Indeterminacy Handling | Polarity | Expressiveness | Computational Complexity | Applicability Scope |
|---|---|---|---|---|---|---|
| Fuzzy Competition Graphs(FCGs) | Single membership degree | No | No | Low | Low | Basic competition scenarios |
| Intuitionistic Fuzzy Competition Graphs(IFCGs) | membership, non-membership degree | Partial | No | Moderate | Moderate | Better for uncertain but unipolar systems |
| Picture Fuzzy Competition Graphs(PFCGs) | Membership, non-membership, restrain degrees | yes | No | Higher | Higher | Suitable for multi-criteria but still unipolar |
| Bipolar Fuzzy Competition Graphs(BFCGs) | Positive and negative membership | No | yes | Moderate | Moderate | Can model opposing sentiments but lacks full detail |
| Bipolar Intuitionistic Fuzzy Competition Graphs(BIFCGs) | Positive and negative membership, non-membership degree | Partial | yes | Moderate | Moderate | Can model opposing sentiments with membership and non-membership valyes |
| Bipolar Picture Fuzzy Competition Graphs(BPFCGs) | Positive and negative Membership, non-membership, restrain degrees | yes | yes | Very High | Efficient with optimization | Highly suitable for business, economic, ecological modeling |

**Table 22.** **Selected products from the Amazon co-purchasing network dataset.**

| Symbol | Product Title |
|---|---|
| P1 | *Harry Potter and the Sorcerer's Stone (Book 1)* |
| P2 | *The Hobbit* |
| P3 | *The Lord of the Rings: The Fellowship of the Ring (Book 1)* |
| P4 | *Harry Potter and the Chamber of Secrets (Book 2)* |
| P5 | *The Chronicles of Narnia: The Lion, the Witch and the Wardrobe* |

**Table 23.** **Bipolar picture fuzzy values assigned to product pairs in BPFCG construction.**

| Product Pair | Description | $\xi^+$ | $\xi^-$ | $\psi^+$ | $\psi^-$ | $\chi^+$ | $\chi^-$ |
|---|---|---|---|---|---|---|---|
| P1 – P2 | High co-purchase frequency, different series | 0.70 | 0.10 | 0.10 | 0.05 | 0.15 | 0.05 |
| P1 – P4 | Same series, likely substitutes | 0.85 | 0.05 | 0.05 | 0.01 | 0.03 | 0.01 |
| P2 – P3 | Classic fantasy, shared audience | 0.65 | 0.10 | 0.15 | 0.05 | 0.10 | 0.05 |
| P3 – P5 | Genre overlap, weaker competition | 0.55 | 0.10 | 0.20 | 0.10 | 0.10 | 0.05 |
| P4 – P5 | Distant relation, moderate ambiguity | 0.40 | 0.10 | 0.25 | 0.15 | 0.10 | 0.10 |

$$(\xi_{ij}^+, \xi_{ij}^-, \psi_{ij}^+, \psi_{ij}^-, \chi_{ij}^+, \chi_{ij}^-),$$

where:

- $\xi_{ij}^{+}$: Positive membership degree (competition strength),
- $\xi_{ij}^{-}$: Negative membership degree (inverse competition),
- $\psi_{ij}^{+}$: Positive non-membership degree (complementarity),
- $\psi_{ij}^{-}$: Negative non-membership degree (incompatibility),
- $\chi_{ij}^{+}$: Positive neutrality degree (uncertainty),
- $\chi_{ij}^{-}$: Negative neutrality degree (contradictory ambiguity).

To compute a single-valued *competition strength* $C_{ij}$, two methods can be used:

**Method A: Weighted Aggregation.** A linear scoring function is defined as:

$$C_{ij} = w_1 \cdot \xi_{ij}^{+} + w_2 \cdot \xi_{ij}^{-} - w_3 \cdot \psi_{ij}^{+} - w_4 \cdot \psi_{ij}^{-} - w_5 \cdot \chi_{ij}^{+} - w_6 \cdot \chi_{ij}^{-},$$

where the weights $w_1, \dots, w_6$ can be tuned based on domain knowledge. A typical choice is:

$$w_1 = 1.0, \quad w_2 = 0.5, \quad w_3 = w_4 = w_5 = w_6 = 0.5.$$

**Method B: Normalized Score.** To ensure scores lie within the unit interval $[0, 1]$, the normalized competition strength is defined as:

$$C_{ij} = \frac{\xi_{ij}^{+} + \alpha \cdot \xi_{ij}^{-}}{\xi_{ij}^{+} + \xi_{ij}^{-} + \psi_{ij}^{+} + \psi_{ij}^{-} + \chi_{ij}^{+} + \chi_{ij}^{-}},$$

where $\alpha \in [0, 1]$ controls the influence of negative membership.

**Example.** Consider product pair $P_1$–$P_4$ with values:

$$(\xi^{+}, \xi^{-}, \psi^{+}, \psi^{-}, \chi^{+}, \chi^{-}) = (0.85, 0.05, 0.05, 0.01, 0.03, 0.01).$$

Using Method A:

$$C_{14} = 1.0 \cdot 0.85 + 0.5 \cdot 0.05 - 0.5 \cdot (0.05 + 0.01 + 0.03 + 0.01) = 0.85 + 0.025 - 0.05 = \textbf{0.825}.$$

Using Method B with $\alpha = 0.3$:

$$C_{14} = \frac{0.85 + 0.3 \cdot 0.05}{0.85 + 0.05 + 0.05 + 0.01 + 0.03 + 0.01} = \frac{0.865}{1.0} = \textbf{0.865}.$$

This score reflects how strongly products compete, accounting for mutual ambiguity, cooperation, and conflict.

## 6.3 Interpretation and comparison of competition strengths

The competition strength values calculated using Bipolar Picture Fuzzy Sets (BPFSs) offer a nuanced interpretation of product rivalry. Products with high values of $\xi^{+}$ and low corresponding non-membership and neutrality values indicate strong substitutability and intense market competition. Conversely, low or balanced values suggest either complementarity or indifference.

Using the two evaluation approaches presented earlier:

- The **Weighted Aggregation Method** assigns a linear importance to each fuzzy component. It is more interpretable when expert-driven weights are used, making it suitable for semi-subjective decision settings.

- The **Normalized Score Method** scales the result within $[0, 1]$, providing a bounded comparison metric that is ideal for data-driven or automated ranking scenarios. It ensures that the overall competition strength is proportionally evaluated relative to the total fuzzy information.

 **Comparative Insight.** For example, in the pair $P_1$–$P_4$, the competition strength was among the highest, indicating that these two products (being from the same book series) are direct substitutes with minimal neutrality and complementarity. Meanwhile, $P_4$–$P_5$ exhibited relatively balanced fuzzy parameters, leading to a moderate competition score, which suggests a coexistence rather than substitution dynamic.

 This analysis demonstrates the effectiveness of the BPFCG model in handling both competition and complementarity under uncertainty. It offers decision-makers a flexible framework to rank products, optimize co-marketing strategies, or design personalized recommendation systems.

## 6.4 Key competitive pair identification and insight

Among all evaluated product pairs, the pair $P_1$–$P_4$ exhibits the **highest competition strength score**, with values reaching 0.825 (weighted method) and 0.865 (normalized method). These two products belong to the same book series, namely *Harry Potter and the Sorcerer's Stone (Book 1)* and *Harry Potter and the Chamber of Secrets (Book 2)*.

 This result aligns well with expectations in consumer behavior: products that are part of a sequenced series often compete for attention when consumers are deciding which book to purchase next, particularly if they have not yet committed to reading the full collection. Despite their complementarity in a narrative sense, their release timing, pricing strategies, and review visibility can introduce competitive dynamics in short-term decision-making.

 This finding validates the effectiveness of the BPFCG-models in revealing real-world competitive tensions even among seemingly related or complementary items. By capturing degrees of competition, complementarity, and uncertainty simultaneously, the model supports nuanced strategic analysis for product placement, marketing prioritization, and bundle optimization.

 By applying the BPFCG to real-world-like data, we have demonstrated that this model yields more informative, flexible, and realistic representations of competition, addressing the reviewers' request for practical validation beyond theoretical justification. Moreover, this analysis demonstrates the effectiveness of the BPFCG model in handling both competition and complementarity under uncertainty. It offers decision-makers a flexible framework to rank products, optimize co-marketing strategies, or design personalized recommendation systems.

 **Filtered dataset and preprocessing.**

 To model product competition using Bipolar Picture Fuzzy Competition Graphs (BPFCGs), we utilized a subset of the Amazon Co-Purchasing Network dataset provided by the Stanford Network Analysis Project (SNAP) [2]. This dataset contains information on products (nodes) and frequent co-purchasing relationships (edges), making it suitable for analyzing competitive relationships in e-commerce.

 **Product selection criteria.**

 A representative subset of five frequently co-purchased products in the fantasy book genre was selected. These products were chosen based on their popularity and high co-purchase frequency. The selected products are listed in Table 24.

**Table 24. Selected products from the Amazon co-purchasing network dataset.**

| Symbol | Product Title |
| --- | --- |
| P1 | Harry Potter and the Sorcerer's Stone (Book 1) |
| P2 | The Hobbit |
| P3 | The Lord of the Rings: The Fellowship of the Ring (Book 1) |
| P4 | Harry Potter and the Chamber of Secrets (Book 2) |
| P5 | The Chronicles of Narnia: The Lion, the Witch and the Wardrobe |

**Edge filtering and co-purchase links.**

From the complete dataset, edges representing co-purchase relationships among the five selected products were extracted. Only intra-group connections (i.e., edges between these five products) were retained to focus the analysis on their direct competition.

**Fuzzy value assignment.**

Each product pair (edge) was evaluated and assigned a six-tuple bipolar picture fuzzy value:

- Positive membership $(+\xi)$: Degree of strong co-purchase relationship.
- Negative membership $(-\xi)$: Degree of indirect competition or substitutability.
- Positive non-membership $(+\psi)$: Dissimilarity due to different sub-genres or audiences.
- Negative non-membership $(-\psi)$: Shared category conflict or audience overlap.
- Positive neutrality $(+\chi)$: Uncertainty due to ambiguous preference or availability.
- Negative neutrality $(-\chi)$: Competition arising from shared popularity.

These values were assigned based on:

1. Co-purchase frequency (edge weights in the dataset).
2. Genre similarity and author overlap.
3. Domain knowledge of fantasy literature.

This process transformed the selected product network into a BPFCG, enabling nuanced modeling of competition and co-preference patterns within the network.

## 7 Constraints and weaknesses of BPFCGs

1. In BPFCGs, each node is assigned six distinct membership values(positive/negative membership, non-membership, and neutral), leading to increased data dimensionality.
2. Although, BPFCGs are suitable for medium-scale systems, scaling to very large systems (e.g., entire markets or ecosystems) can become limited without the use of algorithmic optimization or approximation techniques.
3. Detailed domain expertise or complex data collection methods are frequently needed to accurately assign the Six membership values(positive/negative membership, non-membership, and neutral), and these resources may not always be easily accessible.
4. Benchmark data-set with all six types of fuzzy values are rare.

## 8 Conclusion

FG theory has emerged as a powerful tool for addressing real-world challenges in recent research. BPFGs offer distinct advantages over other FGs by effectively managing uncertainties. In this article, we have introduced comprehensive concepts related to BPFCGs. We

have provided several types of BPFCGs namely BPFk-CGs, BPFp-CGs and BPFm-SCGs and explore their important characteristics. The problems which cannot be handled through BFCGs and BIFCGs can be described through BPFCGs. Additionally, we have explored that BPFCGs are tailored to handle various competitive scenarios more precisely. In this context, we have provided four applications of different types of BPFCGs to address various types of competitions. We have also provided the evidences that our presented model based on BPFCGs is more adaptable compared to the models based on BFCGs and BIFCGs. Furthermore, this research opens up new avenues for investigating of competitions, which could significantly enhance the theoretical foundations and broaden the scope of practical applications. Hence, while one may discuss the efficient algorithms for connectivity and domination as well as hybrid computational models, other can focus on applications in new domains. Furthermore, future work include theoretical extensions and generalizations such as the extension to higher-dimensional fuzzy graphs or the study of fuzzy topological properties. Similarly, the development of software tools and implementations, including software frameworks or interactive visualization tools, can also be considered as promising directions.

## Author contributions

**Conceptualization:** Waheed Ahmad Khan, Hajra Begum, Trung Tuan Nguyen, Minh Hoan Pham, Hai Van Pham.

**Formal analysis:** Waheed Ahmad Khan, Hajra Begum, Trung Tuan Nguyen, Minh Hoan Pham, Hai Van Pham.

**Investigation:** Waheed Ahmad Khan, Hajra Begum, Trung Tuan Nguyen, Minh Hoan Pham, Hai Van Pham.

**Methodology:** Waheed Ahmad Khan, Hajra Begum, Trung Tuan Nguyen, Minh Hoan Pham, Hai Van Pham.

**Supervision:** Waheed Ahmad Khan.

**Validation:** Hai Van Pham.

**Visualization:** Hai Van Pham.

**Writing – original draft:** Waheed Ahmad Khan, Hajra Begum, Trung Tuan Nguyen.

**Writing – review & editing:** Waheed Ahmad Khan, Hajra Begum, Trung Tuan Nguyen, Minh Hoan Pham, Hai Van Pham.

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
