## [Decision Letter · Decision Letter 0]

17 Apr 2025

PONE-D-25-05783A study of competitions in different fields through graphs under bipolar picture fuzzy environmentPLOS ONE

Dear Dr. Khan,

Thank you for submitting your manuscript to PLOS ONE. After careful consideration, we feel that it has merit but does not fully meet PLOS ONE’s publication criteria as it currently stands. Therefore, we invite you to submit a revised version of the manuscript that addresses the points raised during the review process.

We look forward to receiving your revised manuscript.

Kind regards,

Le Hoang Son, Ph.D

Academic Editor

PLOS ONE

 [This research is funded by National Economics University, Vietnam]. 

**Comments to the Author**

1. Is the manuscript technically sound, and do the data support the conclusions?

Reviewer #1: Yes

2. Has the statistical analysis been performed appropriately and rigorously? 

Reviewer #1: No

3. Have the authors made all data underlying the findings in their manuscript fully available?

Reviewer #1: Yes

4. Is the manuscript presented in an intelligible fashion and written in standard English?

Reviewer #1: No

5. Review Comments to the Author

**Reviewer #1**: 

The paper introduces the concept of bipolar picture fuzzy competition graphs (BPFCGs) as an extension of fuzzy competition graphs. It claims to enhance the modeling of real-world competitive systems through a more comprehensive representation of uncertainty. The authors provide mathematical formulations and comparative analyses to support their argument.

While the study contributes to the field of fuzzy graph theory, its practical implications remain ambiguous. The paper lacks explicit real-world case studies that illustrate the necessity of this extended model over traditional competition graphs. Providing concrete applications in business, economics, or ecological modeling would strengthen the paper's impact. To enhance the quality and impact of the paper, the authors should consider the following:

Here are 20 suggestions for improving the paper:

- Make the abstract more concise and the introduction more engaging by highlighting key contributions and applications.

- Reduce redundancy and simplify mathematical expressions to improve readability, especially for those unfamiliar with fuzzy graph theory.

- Provide specific real-world case studies to demonstrate how BPFCGs improve competition modeling in business, economics, or ecology.

- Clearly differentiate BPFCGs from prior models in terms of computational efficiency, predictive accuracy, or applicability.

- Conduct experiments using real-world datasets instead of relying solely on theoretical justifications.

- Compare BPFCGs with existing competition graph models using concrete measures such as accuracy, computational complexity, and scalability.

- Describe in detail the computational methods used to generate BPFCGs and explain their efficiency.

- Compare BPFCGs with a broader range of fuzzy and competition graph models to justify their necessity.

- Discuss the constraints and weaknesses of BPFCGs, such as computational overhead or specific application constraints.

- Remove redundant definitions and explanations that do not contribute to new insights.

- Streamline the introduction, definitions, and results sections to enhance logical flow.

- Explain the real-world significance of each theorem and derivation to make the findings more accessible.

- Provide a deeper discussion on the results, including how BPFCGs can be integrated into various research areas.

- Use visual representations, graphs, or real-world examples to clarify complex concepts.

- Ensure that all terms and abbreviations are consistently used throughout the paper.

- Provide a clear step-by-step explanation of how BPFCGs are constructed and analyzed.

- Suggest specific future work, such as algorithm optimization, application in new domains, or improvements in computational efficiency.

- Apply BPFCGs to a specific industry problem (e.g., market competition) and analyze their effectiveness compared to traditional models.

- Present a table comparing BPFCGs with existing models in terms of advantages, limitations, and key features.

- Ensure grammatical accuracy, proper citation formatting, and adherence to journal guidelines.

While the paper presents an interesting extension of fuzzy graph theory, its claims of superiority over existing models are not convincingly substantiated. The research would benefit from empirical validation, performance benchmarks, and clearer practical applications. Addressing these issues would significantly enhance the paper’s contribution to the field.

6. PLOS authors have the option to publish the peer review history of their article (what does this mean?). If published, this will include your full peer review and any attached files.

Reviewer #1: No

---

## [Author Response · Author response to Decision Letter 1]

16 Jun 2025

Reviewer-1 Comments with Responses:

Manuscript ID: PONE-D-25-05783

Title: A study of competitions in different fields through graphs under bipolar picture fuzzy environment

We appreciate the reviewer for their time and precious comments. No doubt, after endorsing these comments our article would become more effective. The point by point response of the reviewers comments are as follows.

Reviewer #1:

The paper introduces the concept of bipolar picture fuzzy competition graphs (BPFCGs) as an extension of fuzzy competition graphs. It claims to enhance the modeling of real-world competitive systems through a more comprehensive representation of uncertainty. The authors provide mathematical formulations and comparative analyses to support their argument.

While the study contributes to the field of fuzzy graph theory, its practical implications remain ambiguous. The paper lacks explicit real-world case studies that illustrate the necessity of this extended model over traditional competition graphs. Providing concrete applications in business, economics, or ecological modeling would strengthen the paper's impact. To enhance the quality and impact of the paper, the authors should consider the following:

Here are 20 suggestions for improving the paper:

Comment#1: Make the abstract more concise and the introduction more engaging by highlighting key contributions and applications.

Response. In accordance with the reviewer's suggestion, we have revised the abstract to enhance clarity and conciseness. Furthermore, we have also highlighted key contributions in the introduction section.

Comment#2: Reduce redundancy and simplify mathematical expressions to improve readability, especially for those unfamiliar with fuzzy graph theory.

1. Eliminate Redundant Explanations, like

A fuzzy set assigns a degree of membership to each element. This degree of membership is a number between 0 and 1.” Simplify to: “A fuzzy set assigns a membership value in [0, 1] to each element.”

2. Use Simpler Notation and Language, Add Brief Intuitive Descriptions

After formal definitions, add a simple sentence for readers unfamiliar with fuzzy theory.

Example: “The fuzzy degree reflects how strongly a node or connection is present in the system.”

3. Streamline Complex Expressions

Break long equations or conditions into smaller, readable components.

Use bullets or numbered steps for algorithmic parts or long logical statements.

Response. As per reviewer’s suggestions, we have removed redundancies wherever found, added sentences before definitions, further elaborated the terms through geometrical representations. Adjusted long equations and equation numbers are also allocated.

Comment#3: Provide specific real-world case studies to demonstrate how BPFCGs improve competition modeling in business, economics, or ecology.

Response. We have added specific real-world case studies to demonstrate how BPFCGs improve competition modeling in section 6: Real-World Application: Modeling Product

Competition Using BPFCGs. Moreover, in sections 4.1, 4.2, 4.4, we have also highlighted that how BPFCGs improve competition modeling in business, economics, or ecology.

Comment#4: Clearly differentiate BPFCGs from prior models in terms of computational efficiency, predictive accuracy, or applicability.

Response. Thank you for the insightful comment. We have expanded our discussion to clearly highlight the advantages of Bipolar Picture Fuzzy Competition Graphs (BPFCGs) over traditional Picture Fuzzy Competition Graphs (PFCGs) through Tables 20 & 21.

Comment#5: Conduct experiments using real-world datasets instead of relying solely on theoretical justifications.

Response. We have added specific real-world case studies to demonstrate how BPFCGs improve competition modeling in section 6: Real-World Application: Modeling Product

Competition Using BPFCGs.

Comment#6: Compare BPFCGs with existing competition graph models using concrete measures such as accuracy, computational complexity, and scalability.

Response. We have compared BPFCGs with existing competition graph models using concrete measures such as accuracy, computational complexity, and scalability in each subsection of the section 4. We constructed several tables in the section 5: Comparative study and prevalence of our presented model.

Comment#7: Describe in detail the computational methods used to generate BPFCGs and explain their efficiency.

Response. We have described the computational methods used to generate BPFCGs after defining BPFCGs, and explain their efficiency in detail in different sections like section 3 (theoretical aspect), section 4 (general application point of view) and section 6 (real-word data perspective).

Comment#8: Compare BPFCGs with a broader range of fuzzy and competition graph models to justify their necessity.

Response. We have compared BPFCGs with a broader range of fuzzy and competition graph models to justify their necessity in Section 5.1, “Prevalence of our presented model for mobile companies through BPFCGs (Tables 19, 20 & 21).”

Comment#9: Discuss the constraints and weaknesses of BPFCGs, such as computational overhead or specific application constraints.

Response. We have highlighted the constraints and weaknesses of BPFCGs, such as computational overhead or specific application constraints in section 7.

Comment#10: Remove redundant definitions and explanations that do not contribute to new insights.

Response. As per reviewer’s suggestion, we have removed redundant definitions and explanations that do not contribute to new insights.

Comment#11: Streamline the introduction, definitions, and results sections to enhance logical flow.

Response. As per reviewer’s suggestion, we have streamlined introduction, definitions, and results sections to enhance logical flow.

Comment#12: Explain the real-world significance of each theorem and derivation to make the findings more accessible.

Response. We thank the reviewer for their thoughtful suggestion. We respectfully note that the manuscript already includes detailed applications that illustrate the real-world relevance of the theoretical framework of BPFCGs. These applications are directly supported by the presented theorems and derivations. Nonetheless, we have reviewed the manuscript carefully and added brief clarifying remarks where needed to further emphasize the real-world significance of key results.

Comment#13: Provide a deeper discussion on the results, including how BPFCGs can be integrated into various research areas.

Response. We have added more information and also highlighted how BPFCGs can be integrated into various research areas in Section 6, “Conclusion”.

Comment#14: Use visual representations, graphs, or real-world examples to clarify complex concepts.

Response. We have added more visual representations in section 2: Preliminaries, and also highlighted graphs and real-world examples to clarify complex concepts in the whole article.

Comment#15: Ensure that all terms and abbreviations are consistently used throughout the paper.

Response. As per the reviewer’s suggestion, we have revised and checked that all terms and abbreviations are consistently used throughout the paper.

Comment#16: Provide a clear step-by-step explanation of how BPFCGs are constructed and analyzed.

Response. We have provided and highlighted a clear step-by-step explanation of how BPFCGs are constructed and analyzed in Section 3, Example 3.1. Moreover, in application section, we have also elaborated this point.

Comment#17: Suggest specific future work, such as algorithm optimization, application in new domains, or improvements in computational efficiency.

Response. We have provided and highlighted specific future work, such as algorithm optimization, application in new domains, or improvements in computational efficiency in Section 8 “Conclusion”.

Comment#18: Apply BPFCGs to a specific industry problem (e.g., market competition) and analyze their effectiveness compared to traditional models.

Response. We highlighted the application of BPFCGs in the market competition in telephonic companies in section, “Analysis of competition among different cell phone companies through BPFCGs,” and analyzed the effectiveness of the application of BPFCGs compared to traditional models in section 5.1, “Prevalence of our presented model for mobile companies through BPFCGs.”

Comment#19: Present a table comparing BPFCGs with existing models in terms of advantages, limitations, and key features.

Response. We have presented a table comparing BPFCGs with existing models in terms of advantages, limitations, and key features in Section 5.1, “Prevalence of our presented model for mobile companies through BPFCGs (Tables. 19, 20 & 21)”.

Comment#20: Ensure grammatical accuracy, proper citation formatting, and adherence to journal guidelines.

Response. As per reviewer’s suggestion, we have ensured grammatical accuracy, proper citation formatting, and adherence to journal guidelines.

---

## [Decision Letter · Decision Letter 1]

1 Jul 2025

PONE-D-25-05783R1A study of competitions in different fields through graphs under bipolar picture fuzzy environmentPLOS ONE

Dear Dr. Khan,

Thank you for submitting your manuscript to PLOS ONE. After careful consideration, we feel that it has merit but does not fully meet PLOS ONE’s publication criteria as it currently stands. Therefore, we invite you to submit a revised version of the manuscript that addresses the points raised during the review process.

We look forward to receiving your revised manuscript.

Kind regards,

Le Hoang Son, Ph.D

Academic Editor

PLOS ONE

Journal Requirements:

Reviewers' comments:

Reviewer's Responses to Questions

**Comments to the Author**

1. If the authors have adequately addressed your comments raised in a previous round of review and you feel that this manuscript is now acceptable for publication, you may indicate that here to bypass the “Comments to the Author” section, enter your conflict of interest statement in the “Confidential to Editor” section, and submit your "Accept" recommendation.

Reviewer #1: All comments have been addressed

2. Is the manuscript technically sound, and do the data support the conclusions?

Reviewer #1: Yes

3. Has the statistical analysis been performed appropriately and rigorously? 

Reviewer #1: Yes

4. Have the authors made all data underlying the findings in their manuscript fully available?

Reviewer #1: Yes

5. Is the manuscript presented in an intelligible fashion and written in standard English?

Reviewer #1: No

6. Review Comments to the Author

Reviewer #1: 1. Check all the citations, for example: Cuong [27] should be Cuong and Kreinovich [27]

2. Improve the quality of all figures, for example Figure 1,7,....

3. Improve the presentation follow the format of PLOS One.

4. Check and improve the quality of English

5. Page 38,39 N. Deva should be Deva

7. PLOS authors have the option to publish the peer review history of their article (what does this mean?). If published, this will include your full peer review and any attached files.

Reviewer #1: No

---

## [Author Response · Author response to Decision Letter 2]

3 Jul 2025

Manuscript ID: PONE-D-25-05783

Title: A study of competitions in different fields through graphs under bipolar picture fuzzy environment

We sincerely appreciate the reviewer for their time and valuable comments. There is no doubt that incorporating these suggestions has enhanced the overall quality of our article. A point-by-point response to the reviewer’s comments is provided below.

Comment#1: Check all the citations, for example: Cuong [27] should be Cuong and Kreinovich [27]

Response. Thank you very much for your suggestion. We have corrected and replaced Cuong by Cuong and Kreinovich. Moreover, we have revised citations and corrected wherever we have found such mistakes.

Comment#2: Improve the quality of all figures, for example Figure 1,7,....”

Response. As per reviewer suggestion, we have improved the quality of almost all the figures.

Comment#3: Improve the presentation follow the format of PLOS One.

Response. As per reviewer’s suggestion, we have followed the format of PLOS ONE, properly.

Comment: Check and improve the quality of English.

Response. Following the reviewers’ suggestions, we have revised the manuscript and improved the quality of English—from the abstract to the conclusion—wherever necessary.

Comment: Page 38,39 N. Deva should be Deva.

Response. As per the reviewer’s suggestion, we have replaced “N. Deva” with “Deva.” Moreover, we have made similar corrections wherever applicable throughout the manuscript.

---

## [Decision Letter · Decision Letter 2]

13 Jul 2025

A study of competitions in different fields through graphs under bipolar picture fuzzy environment

PONE-D-25-05783R2

Dear Dr. Khan,

We’re pleased to inform you that your manuscript has been judged scientifically suitable for publication and will be formally accepted for publication once it meets all outstanding technical requirements.

Kind regards,

Le Hoang Son, Ph.D

Academic Editor

PLOS ONE

Additional Editor Comments (optional):

- The authors are requested to remove sub-sections like 1.1 Background, 1.2 Need of fuzzy generalizations,... 

- Section "1.3 Literature Review" should move to Section 2 and onwards.

- The relevance to PLOS ONE should be discussed in the Abstract and in the Introduction. 

**Comments to the Author**

1. If the authors have adequately addressed your comments raised in a previous round of review and you feel that this manuscript is now acceptable for publication, you may indicate that here to bypass the “Comments to the Author” section, enter your conflict of interest statement in the “Confidential to Editor” section, and submit your "Accept" recommendation.

Reviewer #1: All comments have been addressed

2. Is the manuscript technically sound, and do the data support the conclusions?

Reviewer #1: Yes

3. Has the statistical analysis been performed appropriately and rigorously? 

Reviewer #1: Yes

4. Have the authors made all data underlying the findings in their manuscript fully available?

Reviewer #1: Yes

5. Is the manuscript presented in an intelligible fashion and written in standard English?

Reviewer #1: Yes

6. Review Comments to the Author

**Reviewer #1**: I introduce this paper for publication. Please check again carefully the English, the presentation and reference of the final proof.

---

## [Editor Report · Acceptance letter]

PONE-D-25-05783R2

PLOS ONE

Dear Dr. Khan,

I'm pleased to inform you that your manuscript has been deemed suitable for publication in PLOS ONE. Congratulations! Your manuscript is now being handed over to our production team.

Kind regards,

on behalf of

Prof. Le Hoang Son

Academic Editor

PLOS ONE